# Beyond Black Box Densities: Parameter Learning for the Deviated Components

**Dat Do**[⋆]
Department of Statistics
University of Michigan at Ann Arbor
Ann Arbor, MI 48109
dodat@umich.edu

**Nhat Ho**[⋆]
Department of Statistics and Data Sciences
University of Texas at Austin
Austin, TX 78712
minhnhat@utexas.edu

**XuanLong Nguyen**
Department of Statistics
University of Michigan at Ann Arbor
Ann Arbor, MI 48109
xuanlong@umich.edu

## Abstract

As we collect additional samples from a data population for which a known density function estimate may have been previously obtained by a black box method, the increased complexity of the data set may result in the true density being deviated from the known estimate by a mixture distribution. To learn about this phenomenon, we consider the *deviating mixture model* $(1-\lambda^*)h_0 + \lambda^*(\sum_{i=1}^{k} p_i^* f(x|\theta_i^*))$, where $h_0$ is a known density function, while the deviated proportion $\lambda^*$ and latent mixing measure $G_* = \sum_{i=1}^{k} p_i^* \delta_{\theta_i^*}$ associated with the mixture distribution are unknown. Using a novel notion of distinguishability between the known density $h_0$ and the deviated mixture distribution, we establish rates of convergence for the maximum likelihood estimates of $\lambda^*$ and $G^*$ under Wasserstein metrics. Simulation studies are carried out to illustrate the theory.

## 1 Introduction

Most data-driven learning processes typically consist of an iteration of steps that involve model training and fine-tuning, with more data in-take leading to further model re-training and refinement. As more samples become available and exhibit more complex patterns, the initial model may be obsolete, risks being discarded, or absorbed into a richer class of models that adapt better to the increased complexity. It takes considerable resources to train complex models on a rich data population. Moreover, many successful models in modern real-world applications have become so complex that make them hard to properly evaluate and interpret; aside from the predictive performance they may as well be considered as black boxes. Nonetheless, as data populations evolve and so must the learning models, several desiderata remain worthy: the ability to adapt to new complexity while retaining aspects of old "wise" model, and the ability to interpret the changes.

In this paper we will investigate a class of complex models for density estimation that are receptive to *adaptation*, *reuse* and *interpretablity*: we posit that there is an existing distribution $h_0$ which may have been obtained a priori by some means for the data population of interest, e.g., via kernel density estimation (KDE) [22] or mixture models [20] or some modern black box methods, such as generative adversarial networks (GANs) [13, 1] or normalizing flows [9]. Nonetheless, as more

---

[⋆]: Dat Do and Nhat Ho contributed equally to this work.

36th Conference on Neural Information Processing Systems (NeurIPS 2022).

samples become available and/or as the data population changes, it is possible that the true density may deviate from $h_0$. While $h_0$ is potentially difficult to explicate, it is the deviation from the known $h_0$ that we wish to learn and interpret. We will use a mixture distribution to represent this deviation, leading to what we call a *deviating mixture model* for the underlying data population:

$$p_{\lambda^* G_*}(x) := (1 - \lambda^*)h_0(x) + \lambda^* F(x, G_*), \tag{1}$$

for $x \in \mathbb{R}^d$, where $F(x, G_*) := \sum_{i=1}^{k_*} p_i^* f(x|\theta_i^*))$ represents a mixture distribution for the density components deviating from $h_0$. Such deviating components are from a known family of density function $f$. The unknown parameters for this model are the mixing proportion $\lambda^* \in [0, 1]$, and the mixing measure $G_* = \sum_{i=1}^{k_*} p_i^* \delta_{\theta_i^*}$, where $k_* \geq 1$ number of *deviated* components. The choice of mixture distribution $F(x, G_*)$ allows us to express complex deviation from $h_0$, yet the overall model remains amenable to the interpretation of its parameters: $\lambda_*$ represents the amount of deviation from the existing candidate $h_0$, while the mixing measure $G_*$ represents heterogeneous patterns of the deviation. Because $h_0$ might be complex and trained with great computational resource to estimate the density of prior data population, it is reasonable to assume $h_0$ be known in the model (1). The primary contribution of this paper is a rigorous investigation into the rather challenging questions of identifiability and parameter learning rates that arise from a standard maximum likelihood estimation procedure.

**Relations to existing works.** This modeling framework owes its roots to several significant bodies of work in both statistics and machine learning literature. In classical statistics, a dominant approach to address the increased complexity of data populations is via hypothesis testing: one can test an alternative (possibly composite) hypothesis represented by a class of distributions against the null hypothesis represented by $h_0$. Due to the constraint for obtaining simple and theoretically valid test statistics in order to accept or reject the null hypothesis, the testing approaches were mostly restricted to simple choices of distribution for the null and alternative hypotheses [6, 10, 7, 4, 8]. More similar to (1) is the class of *contaminated mixture models* for density estimation: in this framework, the data are assumed to be sampled from a mixture of $P_0$ and $Q$ where either $P_0$ or $Q$ can be an unknown distribution that needs to be estimated. While this approach offers more flexibility in terms of modeling, it does not always guarantee the identifiability of the mixing weight or mixture components $P_0, Q$ [25, 23, 19]. Without identifiability, it is virtually impossible to interpret the model parameters for the data domains. To avoid the identifiability issue, several researchers added the semi-parametric or parametric structures on $P_0$ and $Q$, such as $P_0$ and $Q$ are mixture distributions [2, 12]. However, to the best of our knowledge, the convergence theory of these models remains poorly understood, except for some simple settings (see also [3, 5]). The main distinction between our modeling framework of deviating mixture models and the existing research on contaminated mixtures lies in our assumption that one of the mixture components, namely $h_0$ is known, allowing us to focus on the inference of the deviation from $h_0$, for which a considerable learning theory for the parameters of interest can be established and will be presented in this paper. Finally, estimating parameters of mixture distributions is an essential problem in mixture models. The convergence properties have been studied using identifiability notions and Wasserstein distances [21, 18, 15]. Our technical approach requires a generalization of the identifiability notion to take account of structural property of the existing component $h_0$, which helps to shed light on a considerably more complex convergence behavior of the deviated components.

**Contributions.** The primary contribution of this paper is a rich theory of *identifiability* and rates of convergence for *parameters* and density estimation that arise in the *deviating mixture model* (1), under various settings of the existing component $h_0$, and that of the deviating components (via $f$ and $G_*$). Because the convergence of density estimation in Hellinger distance under the MLE procedure is well studied in [26], the bulk of our technical innovation lies in establishing a collection of *inverse bounds* which relate the Hellinger distance of densities in model (1) in terms of that of their parameters. To do that, we introduce a novel notion of *distinguishability* between $h_0$ and family of density $f$. The inverse bounds will be characterized under such distinguishability conditions (or the lack thereof). Our proof technique allows us to characterize different convergence rates of parameters in the deviating mixture model under distinguishable settings. It also gives rise to several new types of inverse bounds in partially distinguishable settings, where we may not have identifiability in our model. To the best of our knowledge, this is the first work in which such bounds are obtained in mixture modeling literature. Moreover, we will provide many examples to demonstrate the broad applicability of our theory, including cases where the existing component $h_0$ is obtained by a black box method (e.g., deep learning model) and a more traditional method (e.g., via KDE's or mixture

models). By doing so, we are able to push the boundary of identifiability and learning theory of mixture models toward a larger class of modern machine learning models.

**Organization.** The remainder part of this paper is structured as follows. In Section 2, we review the MLE method and the identifiability conditions, where the notion of distinguishability is presented. In Section 3, the main results of inverse bounds and convergence rates for parameters estimation of model (1) are shown. In Section 4, multiple simulation experiments are carried out to support the theory. Finally, Section 5 is used to discuss and conclude. Proofs of all the results in the main text are deferred to the Supplementary Material.

**Notation.** We denote by $\mathcal{E}_k(\Theta) = \{\sum_{i=1}^{k} p_i f(x|\theta_i) : \sum_{i=1}^{k} p_i = 1, p_i > 0, \theta_i \in \Theta \ \forall 1 \leq i \leq k\}$ the family of mixtures with exactly $k$ components and $\mathcal{O}_K(\Theta) = \{\sum_{i=1}^{K} p_i f(x|\theta_i) : \sum_{i=1}^{K} p_i = 1, p_i \geq 0, \theta_i \in \Theta \ \forall 1 \leq i \leq K\}$ the family of mixtures with no more than $K$ components. $\mathcal{E}_{k,c_0}(\Theta) = \{\sum_{i=1}^{k} p_i f(x|\theta_i) : \sum_{i=1}^{k} p_i = 1, p_i \geq c_0, \theta_i \in \Theta \ \forall 1 \leq i \leq k, k \leq K\}$ is the family of mixtures with exactly $K$ components and mixing proportions being bounded below by $c_0$, and $\mathcal{O}_{K,c_0}(\Theta) = \{\sum_{i=1}^{k'} p_i f(x|\theta_i) : \sum_{i=1}^{k'} p_i = 1, p_i \geq c_0, \theta_i \in \Theta \ \forall 1 \leq i \leq k', k' \leq K\}$. $\|\cdot\|_2$ is the usual $l^2$ norm for vectors in $\mathbb{R}^d$ and matrices in $\mathbb{R}^{d \times d}$. We write $g(x) \gtrsim h(x)$ if $g(x) > ch(x)$ for all $x$, where $c$ is a constant does not depend on $x$ (similar for $g(x) \lesssim h(x)$). For any $\lambda \in \mathbb{R}$ and $B \subset \mathbb{R}$, denote by $1_{\{\lambda \in B\}}$ the function that takes value 1 if $\lambda \in B$, and 0 otherwise. For any two densities $p$ and $q$, we denote $h(p, q)$ by the Hellinger distance and $V(p, q)$ by the Total Variation distance between them.

## 2 Identifiability and distinguishability theory

The principal goal of the paper is to establish the efficiency of parameter learning for the deviating mixture model (1) via the standard maximum likelihood estimation (MLE) method. To achieve this goal, the parameters have to be identifiable to begin with. Thus, our theory builds on and extends a standard notion of identifiability of families of density $\{f(x|\theta) : \theta \in \Theta\}$ that has been considered in previous work [21, 15].

**Definition 2.1.** The family $\{f(x|\theta), \theta \in \Theta\}$ (or in short, $f$) is identifiable in the order $r$, for some $r \geq 1$, if $f(x|\theta)$ is differentiable up to the order $r$ in $\theta$ and the following holds:

A1. For any $k \geq 1$, given $k$ different elements $\theta_1, \ldots, \theta_k \in \Theta$, if we have $\alpha_\eta^{(i)}$ such that for almost all $x$

$$\sum_{l=0}^{r} \sum_{|\eta|=l} \sum_{i=1}^{k} \alpha_\eta^{(i)} \frac{\partial^{|\eta|} f}{\partial \theta^\eta}(x|\theta_i) = 0$$

then $\alpha_\eta^{(i)} = 0$ for all $1 \leq i \leq k$ and $|\eta| \leq r$.

Many commonly used families $f$ for mixture modeling satisfy the first order identifiability condition, including location-scale Gaussian distributions, e.g., $f(x|\theta) = N(x|\mu, \sigma^2)$ where $\mu$ and $\sigma^2$ represent the mean (location) and variance (scale) parameters, and location-scale Student's t-distributions. In model (1), however, due to the presence of the existing component $h_0$, the deviated mixture components need to be *distinguishable* from $h_0$. This motivates a more general notion of identifiability, namely, *distinguishability* that we now define. This condition specifies a property jointly for both the existing component $h_0$ and the family of density functions $f$ that make up the deviated components.

**Definition 2.2.** For any natural numbers $k, r \geq 1$, we say that the family of density functions $\{f(\cdot|\theta), \theta \in \Theta\}$ with complexity level $k$ (or in short, $(f, k)$) is distinguishable up to the order $r$ from $h_0$ if the following holds:

A2. For any $k$ distinct components $\theta_1, \ldots, \theta_k$, if we have real coefficients $\alpha_\eta^{(i)}$ for $0 \leq i \leq k$ such that

$$\alpha^{(0)} h_0(x) + \sum_{l=0}^{r} \sum_{|\eta|=l} \sum_{i=1}^{k} \alpha_\eta^{(i)} \frac{\partial^{|\eta|} f}{\partial \theta^\eta}(x|\theta_i) = 0,$$

for almost surely $x \in \mathcal{X}$, then $\alpha^{(0)} = \alpha_\eta^{(i)} = 0$ for $1 \leq i \leq k$ and $|\eta| \leq r$.

We observe that the identifiable condition is a direct consequence of the corresponding distinguishable condition. A simple but non-trivial example of the distinguishability condition can be derived directly from the definitions.

**Example 2.3.** (a) When $h_0(x) = \sum_{i=1}^{k_0} p_i^0 f(x|\theta_i^0)$ for some given weights $(p_1^0, \ldots, p_{k_0}^0)$ and parameters $(\theta_1^0, \ldots, \theta_{k_0}^0)$ where $k_0 \geq 1$, then $(f, k)$ is distinguishable in the order $r$ from $h_0$ as long as $k < k_0$ and the family of density $f$ is identifiable in the order $r$.

(b) Given the choice of $h_0$ in (a), $(f, k)$ is not distinguishable in the order $r$ from $h_0$ when $k \geq k_0$.

More significantly, we can establish a broad class of $h_0$ and families $f$ for which distinguishability holds. This is exemplified by the following theorem, where $f$ represents a family of location or location-scale Gaussian kernels, and $h_0$ is subject to a relatively weak condition.

**Theorem 2.4.** *(a) Suppose that $-\log h_0(x) \gtrsim \|x\|_2^{\beta_1}$ or $-\log h_0(x) \lesssim \|x\|_2^{\beta_2}$ for all $\|x\|_2 > x_0$, for some $x_0 > 0$, $\beta_1 > 2$, and $\beta_2 < 2$. Then, for $f$ being family of location-scale Gaussian and any $k > 0$, $(f, k)$ is distinguishable from $h_0$ up to the first order, where the derivatives in Assumption A2 are taken with respect to both location and scale parameters, and $(f, k)$ is also distinguishable from $h_0$ up to any order, where the derivatives in Assumption A2 are taken only with location parameters.*

*(b) Suppose that $h_0$ is the pdf of a pushforward measure of $N(0, I_d)$ by a piecewise linear function with a finite and positive number of breakpoints. Then, the same conclusions as in part (a) hold.*

The proof of Theorem 2.4 is in Appendix C.1, where the main proof technique is carefully examining the tail densities of $h_0$ and $f$ at infinity. Note that in part (a), $h_0$ can be a pdf of any distribution possessing a lighter or heavier tail than Gaussian distributions, and in part (b), $h_0$ represents the pushforward of a Gaussian distribution by any piecewise linear function (recall that family of piecewise linear functions is dense in the Banach space of continuous functions with compact support). In the sequel we shall demonstrate several examples of interest that are applicable to Theorem 2.4 where $h_0$ may have been estimated by some popular "black box" methods.

**Kernel based representation.** Suppose that $h_0$ was obtained from a $m$-sample $Y_1, \ldots, Y_m \in \mathbb{R}^d$ by a classical kernel density estimation (KDE) method [22] or a RKHS-based method [24], so that

$$h_0(x) = \frac{1}{m} \sum_{j=1}^{m} k_\sigma(x, Y_j) \quad \forall x \in \mathbb{R}^d, \tag{2}$$

where $k_\sigma$ is a kernel function with bandwidth $\sigma$. Popular choices of kernels include the Gaussian kernel $k_\sigma(x, x') = \left(\frac{1}{\sqrt{2\pi}\sigma}\right)^d \exp\left(-\frac{\|x - x'\|_2^2}{2\sigma^2}\right)$ and the multivariate Student's kernel $k_\sigma(x, x') = \left(\frac{1}{\sqrt{\pi}\sigma}\right)^d \frac{\Gamma((\nu+d)/2)}{\Gamma(\nu/2)} \left(1 + \frac{\|x-x'\|_2^2}{\nu\sigma^2}\right)^{-\frac{\nu+d}{2}}$. The corresponding distinguishability guarantee is as follows.

**Corollary 2.5.** *Suppose $h_0$ is defined by Eq. (2), where $k_\sigma$ is Gaussian kernel and $m > K$, or $k_\sigma$ is the multivariate Student's kernel. Then, for $f$ being family of location-scale Gaussian, $(f, K)$ is distinguishable from $h_0$ up to the first order, where the derivatives in Assumption A2 are taken with respect to both location and scale parameters, and $(f, K)$ is also distinguishable from $h_0$ up to any order, where the derivatives in Assumption A2 are taken only with location parameters.*

In application, it is common that the condition $m > K$ is satisfied. It is also matches with the scenario that we consider in the paper, where $h_0$ is already trained using a big data set, and there is a small number of deviated components.

**Neural networks.** Deep neural networks represent a powerful, albeit black box, approximation device for constructing rich classes of distribution for generative models [13, 1]. Accordingly, $h_0$ is the pdf function of a Gaussian distribution being push-forwarded by a map $T$, which is represented by a neural network (NN). Suppose that the NN representing $T$ has a positive and finite number of layers $L$, and so

$$T(x) = a(W_L a(W_{L-1}(\ldots a(W_1 x + b_1)) + b_{L-1}) + b_L), \tag{3}$$

where $W_1, \ldots, W_L \in \mathbb{R}^{d \times d}$ are the weights and $b_1, \ldots, b_L \in \mathbb{R}^d$ are the biases. The activation function $a$ is chosen to be rectified linear unit (ReLU) function defined by $a(x) = \max\{x, 0\}$, and is applied elementwise to any vector in $\mathbb{R}^d$. The corresponding guarantee on the distinguishability condition is as follows.

**Corollary 2.6.** *Suppose that $h_0$ is the pdf a pushforward measure of $N(0, I_d)$ by a map $T$ defined by Eq. (3). Then, for $f$ being family of location-scale Gaussian and any $k > 0$, $(f, k)$ is distinguishable*

*from $h_0$ up to the first order, where the derivatives in Assumption A2 are taken with respect to both location and scale parameters, and $(f, k)$ is also distinguishable from $h_0$ up to any order, where the derivatives in Assumption A2 are taken only with location parameters.*

## 3 Convergence rates of density estimation

In this section, we first establish the rate of density estimation for the deviating mixture models in Section 3.1. We then describe a general procedure to obtain the convergence rate of parameter estimation based on that of density estimation via inverse bounds in Section 3.2. Finally, we provide comprehensive inverse bounds under several settings of the deviating mixture models in Section 3.3.

### 3.1 MLE for deviating mixture model

Given $n$ i.i.d. sample $X_1, X_2, \ldots, X_n$ from $p_{\lambda^* G_*}$ as in model (1), where $G_*$ has $k_*$ components, we want to estimate $\lambda^*$ and $G_*$ from the data. We refer to the problem as in *exact-fitted setting* if $k_*$ is known, and we refer to it as in *over-fitted setting* if $k_*$ is unknown but is known to be bounded by some number $K$. We denote the MLE for exact-fitted setting by

$$\widehat{\lambda}_n, \widehat{G}_n \in \underset{\lambda \in [0,1], G \in \mathcal{E}_{k_*}(\Theta)}{\arg\max} \sum_{i=1}^{n} \log(p_{\lambda G}(X_i)),$$

and for the over-fitted setting, we replace $\mathcal{E}_{k_*}(\Theta)$ in the equation above by $\mathcal{O}_K(\Theta)$, where $K \geq k_*$.

In order to state a rate of convergence for the density estimators $p_{\widehat{G}_n}$ under the Hellinger distance $h$ [26], we need a condition on the complexity of the function class

$$\overline{\mathcal{P}}_k^{1/2}(\Theta, \epsilon) = \left\{ \bar{p}_{\lambda G}^{1/2} : G \in \mathcal{O}_k(\Theta), \ h(\bar{p}_{\lambda G}, p_{\lambda^* G_*}) \leq \epsilon \right\}, \tag{4}$$

where for any $G \in \mathcal{O}_K(\Theta)$, we write $\bar{p}_{\lambda G} = (p_{\lambda G} + p_{\lambda^* G_*})/2$. The definition of $\overline{\mathcal{P}}_k(\Theta, \epsilon)$ originates from [26]. We measure the complexity of this class through the bracketing entropy integral

$$\mathcal{J}_B(\epsilon, \overline{\mathcal{P}}_k^{1/2}(\Theta, \epsilon), \nu) = \int_{\epsilon^2/2^{13}}^{\epsilon} \sqrt{\log N_B(u, \overline{\mathcal{P}}_k^{1/2}(\Theta, \epsilon), \nu)} du \vee \epsilon, \tag{5}$$

where $N_B(\epsilon, X, \eta)$ denotes the $\epsilon$-bracketing number of a metric space $(X, \eta)$ and $\nu$ is the Lebesgue measure. We require the following assumption.

A3. Given a universal constant $J > 0$, there exists $N > 0$, possibly depending on $\Theta$ and $k$, such that for all $n \geq N$ and all $\epsilon > (\log n / n)^{1/2}$,

$$\mathcal{J}_B(\epsilon, \overline{\mathcal{P}}_k^{1/2}(\Theta, \epsilon), \nu) \leq J \sqrt{n} \epsilon^2.$$

**Theorem 3.1.** *Assume that Assumption A3 holds, and let $k \geq 1$. There exists a constant $C > 0$ depending only on $\Theta, k$ such that for all $n \geq 1$,*

$$\sup_{G_* \in \mathcal{O}_k(\Theta), \lambda^* \in [0,1]} \mathbb{E}_{\lambda^*, G_*} h(p_{\widehat{\lambda}_n \widehat{G}_n}, p_{\lambda^* G_*}) \leq C\sqrt{\log n / n}.$$

Therefore, in order to get convergence rate for density functions based on MLE procedure, we only need to check assumption A3. This assumption holds true for a wide range class of parametric model [26]. For our model, we give an example that it holds when $h_0$ has an exponential tail (satisfied for KDE's and Neural networks above) and $f$ is location-scale Gaussian distribution.

**Proposition 3.2.** *Suppose $f$ is location-scale Gaussian family and $\Theta = [-a, a]^d \times \Omega$, where $\Omega$ is a subset of $S_d^{++}$ whose eigenvalues are bounded in $[\underline{\lambda}, \overline{\lambda}]$, $a, \underline{\lambda}, \overline{\lambda} > 0$, and $h_0$ is bounded with tail $-\log h_0(x) \gtrsim \|x\|_2^\beta$ for some $\beta > 0$. Then, the family of densities $\{p_{\lambda G} : \lambda \in [0, 1], G \in \mathcal{O}_k(\Theta)\}$ satisfies assumption A3.*

### 3.2 Parameter learning rates of deviated components

The core of this paper lies in establishing a collection of *inverse bounds*, provided that some distinguishability condition developed in Section 2 holds. The inverse bounds basically say that a small distance between $p_{\lambda G}$ and $p_{\lambda^* G_*}$ under the total variation distance entails that $(\lambda, G)$ and $(\lambda^*, G_*)$

are similar under appropriate distances, where $(\lambda^*, G_*)$ is fixed. To this end, we employ Wasserstein metrics [27] and their extensions.

**Wasserstein distances.** Wasserstein distances are natural and useful for assessing the convergence of latent mixing measures in mixture models [21, 16, 14]. Given two measures $G = \sum_{i=1}^{k} p_i \delta_{\theta_i}$ and $G' = \sum_{j=1}^{k'} p'_j \delta_{\theta'_j}$ on a space $\Theta$ endowed with a metric $\rho$, the Wasserstein metric of order $r \geq 1$ is:

$$W_r(G, G') = [\inf_q \sum_{i,j} q_{ij} \rho^r(\theta_i, \theta'_j)]^{1/r},$$

where the infimum is taken over all joint distribution on $[1, \ldots, k] \times [1, \ldots, k']$ such that $\sum_i q_{ij} = p'_j, \sum_j q_{ij} = p_i$. Note that if $G_n$ is a sequence of discrete measures that converges to $G$ in a Wasserstein distance, then for every atom of $G$, there is a subset of atoms of $G_n$ converges to it. Therefore, the convergence in Wasserstein metrics implies convergence of parameters in mixture models. In this paper, space $\Theta$ is often chosen to be a compact subset of $\mathbb{R}^d$ and $\rho$ is the usual $l^2$ distance. In the case of location-scale Gaussian mixtures, space $\Theta$ is a compact subset of $\mathbb{R}^d \times S_d^{++}$, where $S_d^{++}$ is the set of positive definite and symmetric matrices in $\mathbb{R}^{d \times d}$, and for every $(\mu, \Sigma), (\mu', \Sigma') \in \Theta$, the distance $\rho$ is defined by $\rho((\mu, \Sigma), (\mu', \Sigma')) = \|\mu - \mu'\|_2 + \|\Sigma - \Sigma'\|_2$.

**From inverse bounds to parameter learning rates.** Suppose that some distinguishablity condition is satisfied, then we will establish an inverse bound providing a guarantee that a small distance between $p_{\lambda^* G_*}$ and $p_{\lambda G}$ entails a small distance between $\lambda$ and $\lambda^*$ and between $G$ and $G_*$. More concretely, define a divergence between two measures $\lambda G$ and $\lambda^* G_*$ via

$$\overline{W}_r(\lambda G, \lambda^* G_*) := |\lambda - \lambda^*| + (\lambda + \lambda^*) W_r^r(G, G_*).$$

for all $r \geq 1$, and the inverse bounds will have the form that $V(p_{\lambda G}, p_{\lambda^*, G_*}) \gtrsim \overline{W}_r(\lambda G, \lambda^* G_*)$, for some $r$ that depends on the level of distinguishable level of the model. Since total variational distance is upper bounded by Hellinger distance, if Assumption A3. holds, then combining the aforementioned inverse bound with Theorem 3.1 we immediately obtain

$$\mathbb{E}_{\lambda^*, G_*} \overline{W}_r(\widehat{\lambda}_n \widehat{G}_n, \lambda^* G_*) \leq C \sqrt{\frac{\log n}{n}}.$$

This further implies that the convergence rate of $\hat{\lambda}_n$ to $\lambda^*$ is of order $(\log(n)/n)^{1/2}$ and the convergence rate of $W_r(\hat{G}_n, G_*)$ to 0 is of order $(\log(n)/n)^{1/2r}$.

### 3.3 Inverse bounds in distinguishable setting

We shall establish inverse bounds provided a distinguishability condition for model (1) holds under either exact-fitted and over-fitted settings regarding the true number of components $k_*$.

**Theorem 3.3.** *Assume that $k_*$ is known and $(f, k_*)$ is distinguishable in the first order from $h_0$. Then, for any $G \in \mathcal{E}_{k_*}(\Theta)$, there exist positive constant $C_1$ and $C_2$ depending only on $\lambda^*, G_*, h_0, \Theta$ such that the following holds:*

*(a) When $\lambda^* = 0$, then $V(p_{\lambda^* G_*}, p_{\lambda G}) \geq C_1 \lambda$.*

*(b) When $\lambda^* \in (0, 1]$, then $V(p_{\lambda^* G_*}, p_{\lambda G}) \geq C_2 \overline{W}_1(\lambda G, \lambda^* G_*)$.*

We now present a proof sketch for Theorem 3.3. It is a combination of the Taylor expansion around the true parameters and the Fatou's lemma; the proof technique for the remaining results also shares similar spirit as that of Theorem 3.3. Detailed proof of Theorem 3.3 is deferred to the Appendix.

**Proof sketch for part (b):** Suppose that the bound is not correct, so there exists a sequence $\lambda_n \in (0, 1]$ and $G_n \in \mathcal{E}_{k_*}(\Theta)$ such that $V(p_{\lambda^* G_*}, p_{\lambda_n G_n})/\overline{W}_1(\lambda^* G_*, \lambda_n G_n) \to 0$. Because of the compactness of the parameter space, by extracting a subsequence if necessary, we can assume $\lambda_n \to \lambda', G_n \xrightarrow{W_1} G'$. If $(\lambda', G') \neq (\lambda^*, G_*)$, we have $\overline{W}_1(\lambda^* G_*, \lambda_n G_n) \to \overline{W}_1(\lambda^* G_*, \lambda' G') \neq 0$. It indicates that $V(p_{\lambda^* G_*}, p_{\lambda_n G_n}) \to 0$, which leads to $p_{\lambda^* G_*} = p_{\lambda' G'}$. It contradicts to the distinguishable condition when $(\lambda', G') \neq (\lambda^*, G_*)$).

Otherwise, we have $\lambda_n \to \lambda^*, G_n \to G_*$, and can present $G_n = \sum_{i=1}^{k_*} p_i^n \delta_{\theta_i^n}$ and $G_* = \sum_{i=1}^{k_*} p_i^* \delta_{\theta_i^*}$ such that $p_i^n \to p^*, \theta_i^n \to \theta_i^*$. Because of these limits and by Taylor expansion, we can arrange the difference $(p_{\lambda_n G_n}(x) - p_{\lambda^* G_*}(x))/\overline{W}_1(\lambda_n G_n, \lambda^* G_*)$ in terms of a linear combination of

$h_0(x), f(x|\theta_i^*), \frac{\partial}{\partial\theta}f(x|\theta_i^*)$ such that at least one coefficient is different from 0. By Fatou's lemma,
$0 = \frac{\liminf V(p_{\lambda_n G_n}, p_{\lambda^* G_*})}{\overline{W}_1(\lambda_n G_n, \lambda^* G_*)}dx \geq \int \left|\liminf \frac{p_{\lambda_n G_n}(x) - p_{\lambda^* G_*}(x))}{\overline{W}_1(\lambda_n G_n, \lambda^* G_*)}\right|dx$, which equals to the
absolute integral of the linear combination above. Hence, there exists a non-trivial linear combination of $h_0(x), f(x|\theta_i^*), \frac{\partial}{\partial\theta}f(x|\theta_i^*)$ that equals 0, which contradict to the distinguishability condition. Therefore, we complete the proof.

In application, the true number of components $k_*$ might not be known and we often fit the model (1) with $G \in \mathcal{O}_K(\Theta)$ for some large $K \geq k_*$. The next result shows that similar bounds can also be established in this case, where we require distinguishability of $f$ and $h_0$ in a higher order.

**Theorem 3.4.** *Assume that $k_*$ is unknown and strictly upper bounded by a given $K$. Assume additionally that $(f, K)$ is distinguishable in second order from $h_0$. Then, for any $G \in \mathcal{O}_K(\Theta)$, there exist positive constant $C_1$ and $C_2$ depending only on $\lambda^*, G_*, h_0, \Theta$ such that the following holds:*

*(a) When $\lambda^* = 0$, then $V(p_{\lambda^* G_*}, p_{\lambda G}) \geq C_1\lambda$.*

*(b) When $\lambda^* \in (0, 1]$, then $V(p_{\lambda^* G_*}, p_{\lambda G}) \geq C_2\overline{W}_2(\lambda G, \lambda^* G_*)$.*

Thanks to the distinguishability up to second order, no matter how large the number of over-fitted components $K$ is, we always get the $\overline{W}_2$ lower bound for the total variation distances. Proof of this theorem shares the same spirit with what of Theorem 3.3. The difference here is when we overfit $G_*$ with some $\hat{G}$, there are some atoms of $\hat{G}$ that converges to the same atom of $G_*$, which requires us to do Taylor expansion up to second order and explain the higher order of Wasserstein distance here. Next, we relax the assumption of Theorem 3.4 by working on the setting where $f$ is not second order identifiable. This is an instance of the so-called *weakly identifiable* setting — One popular example of weakly identifiable $f$ is location-scale Gaussian distribution, which admits the partial differential equation (PDE) structure $\frac{\partial^2 f}{\partial\mu^2}(x|\mu, \Sigma) = 2\frac{\partial f}{\partial\Sigma}(x|\mu, \Sigma)$, for all $x \in \mathbb{R}^d$ where $f(x|\mu, \Sigma)$ stands for location-scale Gaussian density function with location $\mu$ and covariance $\Sigma$. In order to illustrate the result of our bound for that weak identifiability setting of $f$, we specifically consider $f$ to be location-scale Gaussian distribution. In this case, the parameter space $\Theta$ is a compact subset of $\mathbb{R}^d \times S_d^{++}$, where $S_d^{++}$ is the set of positive definite and symmetric matrices in $\mathbb{R}^{d\times d}$ equipped with the usual Frobenius norm. To put our result in context, we shall adopt a notion used in analyzing the convergence rate of parameter estimation in location-scale Gaussian mixtures in [16]. For any $k \geq 1$, let $\overline{r}(k)$ be the minimum value of $r$ such that the following system of polynomial equations:

$$\sum_{j=1}^{k+1}\sum_{n_1,n_2}\frac{c_j^2 a_j^{n_1} b_j^{n_2}}{n_1!n_2!} = 0 \text{ for each } \alpha = 1, \ldots, r, \tag{6}$$

does not have any nontrivial solution for the unknown variables $(a_j, b_j, c_j)_{j=1}^{k+1}$, where the ranges of $n_1$ and $n_2$ in the second sum consist of all natural pairs satisfying the equation $n_1 + 2n_2 = \alpha$. A solution to the above system is considered *nontrivial* if all of variables $c_j$ are non-zeroes, while at least one of the $a_j$ is non-zero. Some examples of known values of $\overline{r}$ are $\overline{r}(1) = 4$ and $\overline{r}(2) = 6$, and $\overline{r}(k) \geq 7$ for all $k \geq 3$. Using this notion, we can characterize the convergence of parameters of model (1) for the location-scale Gaussian family via the following theorem for inverse bounds.

**Theorem 3.5.** *Assume that $G^* \in \mathcal{E}_{k^*, c_0}(\Theta)$, and $k_*$ is unknown and strictly upper bounded by a given $K$. In addition, $f$ is location-scale Gaussian distribution and $(f, K)$ with varied location, fixed variance parameters is distinguishable in any order from $h_0$. Then, for any $G \in \mathcal{O}_{K,c_0}(\Theta)$, there exist positive constant $C_1$ and $C_2$ depending only on $\lambda^*, G_*, h_0, \Theta$ such that the following holds:*

*(a) When $\lambda^* = 0$, then $V(p_{\lambda^* G_*}, p_{\lambda G}) \geq C_1\lambda$.*

*(b) When $\lambda^* \in (0, 1]$, then $V(p_{\lambda^* G_*}, p_{\lambda G}) \geq C_2\overline{W}_{\overline{r}(K-k_*)}(\lambda G, \lambda^* G_*)$.*

The proof technique of this result involves doing Taylor expansion of both location and scale parameter up to order $\overline{r}$, then utilize the heat equation $\frac{\partial f}{\partial\Sigma}(x|\mu, \Sigma) = \frac{1}{2}\frac{\partial^2 f}{\partial\mu^2}(x|\mu, \Sigma)$ to compress this expression into linear combination of $h_0$ and derivatives of $f(x|\mu, \Sigma)$ with respect to $\mu$ only. This allows us to use the condition in this theorem to imply a contradiction, and gives rise to Eq. (6).

### 3.4 Inverse Bounds in Partially Distinguishable Setting

What happens if the distinguishability condition required by Def. 2.2 no longer holds generally? Recall in Example 2.3 (b) that this situation is not uncommon, specifically when

$$h_0(x) = f(x; G_0) = \sum_{i=1}^{k_0} p_i^0 f(x|\theta_i^0), \tag{7}$$

where $G_0 := \sum_{i=1}^{k_0} p_i^0 \delta_{\theta_i^0}$. In some specific cases of this setting, in fact, we fail to attain distinguishability, and the model may not even be identifiable in the classical sense, i.e. $p_{\lambda G} = p_{\lambda^* G_*}$ does not guarantee to have $\lambda G = \lambda^* G_*$. Since $h_0$ is the pdf of a mixture distribution — a popular choice for modeling complex forms of probability densities given its amenability to interpretation compared to black box type models — it is of interest to study the implication of parameter estimation for the deviated components in this setting, provided that the distinguisability condition may be at least partially achieved in some suitable sense. As we shall see, our theory demands a more refined analysis. To facilitate the presentation, denote $\mathcal{A} := \{1 \le i \le k_* : \theta_i^* \in \{\theta_1^0, \dots, \theta_{k_0}^0\}\}$. Also, set $\bar{k} := |\mathcal{A}|$, which stands for the cardinal of the set $\mathcal{A}$. Our results will be divided into three separate regimes of $\bar{k}$ and $\lambda^*$: (i) $\lambda^* = 0$, (ii) $\bar{k} < k_0$ and $\lambda^* \in (0,1]$, and (iii) $\bar{k} = k_0$ and $\lambda^* \in (0,1]$. We only choose to present results of the second regime (ii) in the main text because of limited space and because of its representativeness as it shows all the intriguing behaviours of the model in this partially distinguishable setting. The first and third regime are deferred to Appendix A.

#### 3.4.1 Regime B: $\bar{k} < k_0$ and $\lambda^* \in (0,1]$

First, we consider the exactly-specified setting of model (1), namely, $k_*$ is known. When $\bar{k} < k_0$, we can check that we still have dishtinguishability of $h_0$ and linear combinations of $\{f(x|\theta_i^*)\}_{i=1}^{k_*}$ and its derivatives. Therefore, as long as $f$ is first order identifiable, one can invoke the proof of Theorem 3.3 to establish the same lower bound $V(p_{\lambda G}, p_{\lambda^* G_*})$ in terms of $\overline{W}_r(\lambda G, \lambda^* G_*)$ for some $r \ge 1$. Thus, our focus in this subsection is the settings when $k_*$ is unknown.

**Over-fitted setting with strongly identifiable $f$.** Moving to the over-fitted settings of model setup (1), i.e., $k_*$ is unknown and strictly upper bounded by a given $K$, as long as $K \ge k_0$, $(f, K)$ is not distinguishable from $h_0$. Therefore, the results of Theorem 3.3 are not always applicable to the setting when $K \ge k_0$. Besides, in the over-fitted setting, the identifiability of model (1) no longer holds. Indeed, for any $\lambda > \lambda^*$, if we take

$$\overline{G}_*(\lambda) = (1 - \lambda^*/\lambda) G_0 + (\lambda^*/\lambda) G_*, \tag{8}$$

then $p_{\lambda^* G_*} = p_{\lambda \overline{G}_*(\lambda)}$. We present this pathological behavior in the following result.

**Theorem 3.6.** *Assume that $h_0$ takes the form (7) and $\bar{k} < k_0$. Besides that, $K \ge k_0$ and $f$ is second order identifiable. Then, for any $G \in \mathcal{O}_K(\Theta)$, there exist positive constants $C_1$ and $C_2$ depending only on $\lambda^*, G_*, h_0, \Theta$ such that the following hold:*

*(a) If $K \le k_* + k_0 - \bar{k} - 1$, then $V(p_{\lambda^*, G_*}, p_{\lambda, G}) \ge C_1 \overline{W}_2(\lambda G, \lambda^* G_*)$,*

*(b) If $K \ge k_* + k_0 - \bar{k}$, then*

$$V(p_{\lambda^*, G_*}, p_{\lambda, G}) \ge C_2 \left( 1_{\{\lambda \le \lambda^*\}} \overline{W}_2(\lambda G, \lambda^* G_*) + 1_{\{\lambda > \lambda^*\}} W_2^2(G, \overline{G}_*(\lambda)) \right).$$

*(c) As a special case, if $K = k_* + k_0 - \bar{k}$, we have*

$$V(p_{\lambda^*, G_*}, p_{\lambda, G}) \ge C_3 1_{\{\lambda > \lambda^* + \delta\}} W_1(G, \overline{G}_*(\lambda)),$$

*for all $\delta > 0$, where $C_3$ depends on $\lambda^*, G_*, h_0, \Theta, \delta$.*

As we can see, the magnitude of $\lambda$ compared to $\lambda^*$ will decide the solution of $(\lambda, G)$ to the identifiable equation $p_{\lambda G} = p_{\lambda^* G_*}$, therefore lead to different lower bounds such in part (b) of the theorem. In particular, if $\lambda \le \lambda^*$, the solution is $(\lambda, G) = (\lambda^*, G_*)$, and for any $\lambda > \lambda^*$, the solution is $G = \overline{G}_*(\lambda)$ given in Eq. (8). Specifically, when $\lambda$ is strictly larger than $\lambda^*$ by some amount $\delta > 0$, then the latter case is well separated from the former, and we have an exact-fitted result when $K = k_0 + k_* - \bar{k}$.

## 4 Experiments

We now would like to demonstrate the convergence rates in Section 3 via two synthetic experiments: one for distinguishable setting and one for partially distinguishable setting. For the partially distinguishable one, the experiments are in Appendix B.

**Distinguishable setting.** We conduct an experiment where the original data distribution comes from an uniform distribution on a curve (half circle) in $\mathbb{R}^2$ convoluted with Gaussian noises (red curve and blue points in Fig. 1(a)), and train a Normalizing Flow neural network [11] (Masked Autoregressive architecture) with 5 layers to get a good density estimation $h_0$ for this dataset. Then we assume that there are new data coming in, and the original distribution $h_0$ is deviated by a mixture of distributions in the location Gaussian family $f(x|\theta)$. So the true generating density now is

$$p_{\lambda^* G_*}(x) = (1 - \lambda^*)h_0(x) + \lambda^* \sum_{i=1}^{3} p_i^* f(x|\theta_i^*), \tag{9}$$

where $\lambda^* = 0.5, G_* = \sum_{i=1}^{3} p_i^* \delta_{\theta_i^*}$, where $p_1^* = 0.3, p_2^* = 0.3, p_3^* = 0.4, \theta_1^* = (-0.7, 1.5), \theta_2^* = (0.1, 2.0), \theta_3^* = (1.0, 1.5)$. Samples from the deviated component are green points in Fig. 1(a). It can be seen from Proposition 2.4(a) that $h_0$ is distinguishable with family $f$. For each $n$, we simulate $n$ data points from true model (9), estimate $\hat{\lambda}_n, \hat{G}_n$ by the EM algorithm (it is possible because Normalizing Flows provides exact density computation), and measure its convergence to the true $\lambda^*, G_*$. We conduct 16 replications for each sample size. The average error estimations with a 75% error bar can be seen in Fig. 1. The $W_1$ error in the exact-fitted case is of order $(\log(n)/n)^{1/2}$ and $W_2$ error in the over-fitted case is of order $(\log(n)/n)^{1/4}$. Meanwhile, thanks to the distinguishability, the estimation errors in both cases of $\lambda$ are all of the order $(\log(n)/n)^{1/2}$. These simulation results are matched with the theoretical results found in Theorem 3.3 and Theorem 3.4. From the result, we see that the deviating mixture model successfully learns the deviated components and reuses the pre-trained black box model $h_0$, which helps to reduce computational costs.

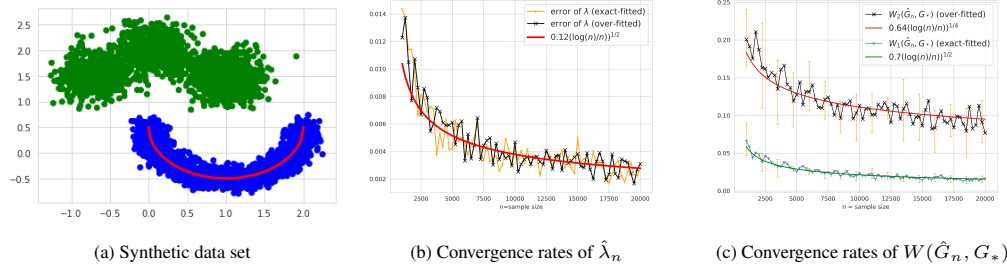

(a) Synthetic data set     (b) Convergence rates of $\hat{\lambda}_n$     (c) Convergence rates of $W(\hat{G}_n, G_*)$

Figure 1: Convergence rates for parameter estimation in the distinguishable case.

## 5 Discussion

In this work, we have presented the deviating mixture model and studied its parameter learning rates under MLE procedure. With a novel notion of distinguishability between distributions, we are able to prove inverse bounds for our model under several distinguishability settings, which allow us to deduce the parameter learning rates from the convergence rate of density functions. The distinguishability condition is shown to be satisfied for multiple families of distributions including those that come from black box models.

We now discuss practical implication of the theory. The deviating mixture model is designed to capture the deviated mixture components, and learning its parameters can reveal meaningful information about subpopulations in the data. When there is distinguishability in the model, our theory implies that we can learn the deviated proportion with the parametric rate and deviated components with a rate depending on the identifiablity of $f$. However, our theory does not support employing the deviating mixture model when the existing distribution $h_0$ itself is a mixture distributions in family $f$ and possesses parameters similar to deviated part, as the learning rate can be slow, and the deviated proportion estimator may not converge to the true value. Asymptotically, when $h_0$ is estimated using

a very complex model (eg. a wide and deep neural network) and somehow approximates a mixture of $f$, and/or the signal from deviating components is low, then the provided learning rates in the paper, while still the same with respect to sample size $n$, may deteriorate from a large multiplicative constant that depends on $h_0, \lambda^*$, and $G_*$.

We believe that this work is the first attempt in the effort of understanding a broader class of mixture models combining with black box models, and interpreting the learned model parameters. There is room for future work going forward. From a theoretical viewpoint, one may be interested in establishing minimax lower bounds for the learning behavior of the deviating mixture model, or show uniform inverse bounds for the model when $\lambda^*$ and $G_*$ are considered as signals that will change with samples. From a modeling viewpoint, it is worthwhile to explore mixtures of black box models and develop a suitable notion of identifiability and inverse bounds so that the learning process is efficient.

## Acknowledgements

Nhat Ho acknowledges support from the NSF IFML 2019844 and the NSF AI Institute for Foundations of Machine Learning. Long Nguyen is partially supported by NSF grant DMS-2015361.

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
