In the supplementary material, we collect proofs and results deferred from the main text. Section A provides remaining results for the partially distinguishable case. Section B presents the simulation studies that demonstrates the results in the partially distinguishable case. Section C contains proofs of results in Section 2, and Section D contains proofs of Section 3.

## A   Additional results

In this appendix, we provide theory for the inverse bounds in partially distinguishable setting when $\bar{k} = k_0$ and $\lambda^* \in (0, 1]$.

### A.1   Regime A: $\lambda^* = 0$.

**Theorem A.1.** *Assume that $h_0$ takes the form (7) and $\lambda^* = 0$. Then, there exist positive constants $C_1$ and $C_2$ depending only on $h_0, \Theta$ such that the following holds:*

*(a) (exact-fitted) If $f$ is first order identifiable, then for any $G \in \mathcal{E}_{k_0}(\Theta)$*

$$V(p_{\lambda^*, G_*}, p_{\lambda, G}) \geq C_1 \lambda W_1(G, G_0).$$

*(b) (over-fitted) If $f$ is second order identifiable, then for any $G \in \mathcal{O}_K(\Theta)$ that $K > k_0$*

$$V(p_{\lambda^*, G_*}, p_{\lambda, G}) \geq C_2 \lambda W_2^2(G, G_0).$$

*(c) (over-fitted and weakly identifiable) If $f$ is location-scale Gaussian distribution and we further assume that $G_* \in \mathcal{E}_{k_*, c_0}(\Theta)$, then for any $G \in \mathcal{O}_{K, c_0}(\Theta)$ that $K > k_0$, there exists $C_3$ depends on $h_0, \Theta_0, c_0$ such that*

$$V(p_{\lambda^*, G_*}, p_{\lambda, G}) \geq C_3 \lambda W_{\bar{r}(K-k_*)}^{\bar{r}(K-k_*)}(G, G_0).$$

We may also "underfit" the deviated components by imposing $G \in \mathcal{O}_K(\Theta)$ such that $K < k_0$. In that case, because of having less atoms, $p_{\lambda G}$ is $K$−distinguishable with $h_0$ and the result in Theorem 3.3 applies.

### A.2   Regime B: $\bar{k} < k_0$ and $\lambda^* \in (0, 1]$

We recall Theorem 3.6 in the main text, together with a similar theorem on weak identifiable family (Theorem A.3), and then provide some additional comments on the results.

**Theorem A.2.** *Assume that $h_0$ takes the form (7) and $\bar{k} < k_0$. Besides that, $K \geq k_0$ and $f$ is second order identifiable. Then, for any $G \in \mathcal{O}_K(\Theta)$, there exist positive constants $C_1$ and $C_2$ depending only on $\lambda^*, G_*, h_0, \Theta$ such that the following hold:*

*(a) If $K \leq k_* + k_0 - \bar{k} - 1$, then $V(p_{\lambda^*, G_*}, p_{\lambda, G}) \geq C_1 \overline{W}_2(\lambda G, \lambda^* G_*)$,*

*(b) If $K \geq k_* + k_0 - \bar{k}$, then*

$$V(p_{\lambda^*, G_*}, p_{\lambda, G}) \geq C_2 \left( 1_{\{\lambda \leq \lambda^*\}} \overline{W}_2(\lambda G, \lambda^* G_*) + 1_{\{\lambda > \lambda^*\}} W_2^2(G, \overline{G}_*(\lambda)) \right).$$

*(c) As a special case, if $K = k_* + k_0 - \bar{k}$, we have*

$$V(p_{\lambda^*, G_*}, p_{\lambda, G}) \geq C_3 1_{\{\lambda > \lambda^* + \delta\}} W_1(G, \overline{G}_*(\lambda)),$$

*for all $\delta > 0$, where $C_3$ depends on $\lambda^*, G_*, h_0, \Theta, \delta$.*

We can view $p_{\lambda G}$ as a mixture distributions with latent mixing measures $\widehat{G} = (1 - \lambda) \sum_{i=1}^{k_0} p_i^0 \delta_{\theta_i^0} + \sum_{i=1}^{K} p_i \delta_{\theta_i}$ having at most $K + k_0$ elements, while $p_{\lambda^* G_*}$ as a mixture with latent measure $\widehat{G}_* = \sum_{i=1}^{\bar{k}} \left[ (1-\lambda^*)p_i^0 + \lambda^* p_i^* \right] \delta_{\theta_i^0} + \sum_{i=\bar{k}+1}^{k_0} (1-\lambda^*)p_i^0 \delta_{\theta_i^0} + \sum_{i=\bar{k}+1}^{k_*} \lambda^* p_i^* \delta_{\theta_i^*}$ having exactly $k_0 + k_* - \bar{k}$

elements. Because $k_0 + k_* - \bar{k} < K + k_0$, a direct application of Theorem 3.2 in [17] gives us $V(p_{\lambda^*,G_*}, p_{\lambda,G}) \gtrsim W_2^2(\widehat{G}_*, \widehat{G})$. But this bound is not as tight as what in Theorem 3.6(c), since $W_1 \gtrsim W_2^2$. The bounds established in the theorem are possible as we carefully explore the structure of $\widehat{G}_*$ and $\widehat{G}$.

**Over-fitted setting with weakly identifiable $f$.** Similar to Theorem 3.5, when $f$ is the location-scale Gaussian, the weak identifiability can worsen the power of the bound in the over-fitted case.

**Theorem A.3.** *Assume that $h_0$ takes the form* (7). *Besides that, $K \geq k_0$ and $f$ is location-scale Gaussian distribution. Then, for any $\lambda \in [0,1]$ and $G \in \mathcal{O}_{K,c_0}(\Theta)$ for some $c_0 > 0$, there exist positive constants $C_1, C_2, C_3, C_4$ depending only on $\lambda^*, G_*, G_0, \Theta$ ($C_3$ and $C_4$ also depend on $\delta$) such that the following holds:*

*(a) When $K \leq k_* + k_0 - \bar{k} - 1$, then $V(p_{\lambda^*,G_*}, p_{\lambda,G}) \geq C_1 \overline{W}_{\bar{r}(K-k_*)}(\lambda G, \lambda^* G_*)$.*

*(b) When $K \geq k_* + k_0 - \bar{k}$, then*

$$V(p_{\lambda^*,G_*}, p_{\lambda,G}) \geq C_2 \left( 1_{\{\lambda \leq \lambda^*\}} \overline{W}_{\bar{r}(K-k_*)}(\lambda G, \lambda^* G_*) + 1_{\{\lambda > \lambda^*\}} W_{\bar{r}(K-k_*)}^{\bar{r}(K-k_*)}(G, \overline{G}_*(\lambda)) \right).$$

*(c) For $\delta > 0$, when $K = k_* + k_0 - \bar{k}$, we have*
$$V(p_{\lambda^*,G_*}, p_{\lambda,G}) \geq C_3 1_{\{\lambda > \lambda^* + \delta\}} W_1(G, \overline{G}_*(\lambda)),$$

*and when $K > k_* + k_0 - \bar{k}$, we have*

$$V(p_{\lambda^*,G_*}, p_{\lambda,G}) \geq C_4 1_{\{\lambda > \lambda^* + \delta\}} W_{\bar{r}(K-k_0-k_*+\bar{k})}^{\bar{r}(K-k_0-k_*+\bar{k})}(G, \overline{G}_*(\lambda)).$$

In this theorem, we once again observe the pathological behavior of the lower bound by Wasserstein distances caused by the unidentifiability of the model (1). In part (c), when there is a well separation between two region of solutions of equation $p_{\lambda G} = p_{\lambda^* G_*}$, we can improve the order of Wasserstein distances for both exact-fitted case and over-fitted case. In application, if $\hat{\lambda}_n$ and $\hat{G}_n$ are the MLE of model (1) estimated by $n$ i.i.d. data, then the convergence of $(\hat{\lambda}_n, \hat{G}_n)$ depends on the limit of $\hat{\lambda}_n$ (or its subsequence) comparing to $\lambda^*$. If $K = k_0 + k_* - \bar{k}$, any subsequence of $(\hat{\lambda}_n)$ having limit greater than $\lambda^*$ can achieve $W_1$ convergence rate of the distance between $\hat{G}_n$ and $\overline{G}_*(\hat{\lambda}_n)$. If $K > k_0 + k_* - \bar{k}$, any subsequence of $(\hat{\lambda}_n)$ having limit greater than $\lambda^*$ can achieve $W_{\bar{r}(K-k_0-k_*+\bar{k})}^{\bar{r}(K-k_0-k_*+\bar{k})}$ convergence rate of the distance between $\hat{G}_n$ and $\overline{G}_*(\hat{\lambda}_n)$, where $\bar{r}(K - k_0 - k_* + \bar{k})$ is smaller than $\bar{r}(K - k_*)$ in part (b).

## A.3   Regime C: $\bar{k} = k_0$ and $\lambda^* \in (0,1]$.

When $\bar{k} = k_0$, $(f, k_*)$ and $(f, K)$ are not distinguishable from $h_0$. It indicates that the results of Theorem 3.3 are no longer applicable to this setting. If $G^* = G_0$, the setting goes back to the case $\lambda^* = 0$ and it is already considered, so from this section, we assume that $G_* \neq G_0$. To streamline the argument, we further denote a few more notations. As $\bar{k} = k_0$, we can rewrite $G_*$ as follows:

$$G_* = \sum_{i=1}^{k_0} p_i^* \delta_{\theta_i^0} + \sum_{i=k_0+1}^{k_*} p_i^* \delta_{\theta_i^*}. \tag{10}$$

Because of the non-identifiability, the lower bound of $V(p_{\lambda G}, p_{\lambda^* G_*})$ must be inspected carefully based on the magnitude of mixing proportions of $p_{\lambda G}$ compared to what of $p_{\lambda^* G_*}$. To serve this purpose, we denote

$$\mathcal{B} := \{\lambda \in [0,1] : (\lambda^* - \lambda)p_i^0 \leq \lambda^* p_i^* \ \forall \ 1 \leq i \leq k_0\},$$
$$\mathcal{I}(\lambda) := \{1 \leq i \leq k_0 : (\lambda^* - \lambda)p_i^0 > \lambda^* p_i^*\}.$$

For any $\lambda \in [0,1]$, we say that the set $\mathcal{I}(\lambda)$ is *ratio-independent* if and only if $|\mathcal{I}(\lambda)| = 1$ or $p_i/p_i^* = p_j/p_j^*$ for all $i,j \in \mathcal{I}(\lambda)$ when $|\mathcal{I}(\lambda)| \geq 2$. Moreover, we define

$$\widetilde{G}_*(\lambda) := \frac{1}{\mathcal{S}(\mathcal{I}(\lambda))} \left( \sum_{i \in \mathcal{I}(\lambda)^c} \left[ p_i^* \lambda^* + (\lambda - \lambda^*) p_i^0 \right] \delta_{\theta_i^0} \right.$$

$$\left. + \lambda^* \sum_{i=k_0+1}^{k_*} p_i^* \delta_{\theta_i^*} \right), \tag{11}$$

where $\mathcal{S}(\mathcal{I}(\lambda)) := \sum_{i \in \mathcal{I}(\lambda)^c} \left[ p_i^* \lambda^* + (\lambda - \lambda^*) p_i^0 \right] + \lambda^* \sum_{i=k_0+1}^{k} p_i^*$. In the case $\mathcal{I}(\lambda)$ is ratio-independent, the identifiable equation $p_{\lambda G} = p_{\lambda^* G_*}$ attains a solution $G = \widetilde{G}_*(\lambda)$ as in equation (11). Hence, in the following, we need to divide $\lambda$ into several regimes to specify the lower bound for $V(p_{\lambda G}, p_{\lambda^* G_*})$ based on appropriate distances of $(\lambda, G)$ and $(\lambda^*, G_*)$.

**Setting with second order identifiable $f$:** We first consider the setting when $f$ is second order identifiable and the model setup (1) is over-fitted. The following result demonstrates that under different settings of $\lambda$ and $\mathcal{I}(\lambda)$, the lower bound of $V(p_{\lambda G}, p_{\lambda^* G_*})$ in terms of its corresponding parameters $(\lambda, G)$ and $(\lambda^*, G_*)$ can be very different.

**Theorem A.4.** *Assume that $h_0$ takes the form (7) and $\bar{k} = k_0$. Besides that, $f$ is second order identifiable. Then, for any $\lambda \in [0, 1]$ and $G \in \mathcal{O}_K(\Theta)$ that $K \geq k_*$, there exist positive constants $C_1$ and $C_2$ depending only on $\lambda^*, G_*, G_0, \Theta$ such that the following holds:*

(a) *If $\mathcal{I}(\lambda)$ is not ratio-independent, then*

$$V(p_{\lambda^* G_*}, p_{\lambda G}) \geq C_1 \left[ 1_{\{\lambda \in \mathcal{B}^c\}} + 1_{\{\lambda \in \mathcal{B}\}} W_2^2(G, \overline{G}_*(\lambda)) \right]. \tag{12}$$

(b) *If $\mathcal{I}(\lambda)$ is ratio-independent, then*

$$V(p_{\lambda^*, G_*}, p_{\lambda, G}) \geq C_2 \left[ 1_{\{\lambda \in \mathcal{B}^c\}} \left( \sum_{i \in \mathcal{I}(\lambda)} \left[ (\lambda^* - \lambda) p_i^0 \right. \right. \right.$$
$$\left. \left. - \lambda^* p_i^* \right] + \mathcal{S}(\mathcal{I}(\lambda)) W_2^2(G, \widetilde{G}_*(\lambda)) \right)$$
$$\left. + 1_{\{\lambda \in \mathcal{B}\}} W_2^2(G, \overline{G}_*(\lambda)) \right]. \tag{13}$$

We can see that when $\lambda \in \mathcal{B}^c$ and $\mathcal{I}(\lambda)$ is not ratio-independent, the bound in equation (12) shows that $V(p_{\lambda^* G_*}, p_{\lambda G}) \geq C_1$. It is due to the fact that $(\lambda^* - \lambda) p_i^0 - \lambda^* p_i^*$ cannot be simultaneously arbitrarily small as $i \in \mathcal{I}(\lambda)$. On the other hand, these terms can become very small at the same time when $\mathcal{I}(\lambda)$ is ratio-independent. It implies that $V(p_{\lambda^* G_*}, p_{\lambda G})$ can become arbitrarily close to 0 under this setting of $\mathcal{I}(\lambda)$. It explains the difference of bounds between two settings of $\mathcal{I}(\lambda)$.

**Setting with weakly identifiable $f$:** Finally, we consider the settings of model setup (1) when $f$ is weakly identifiable. We specifically choose $f$ to be location-scale Gaussian distribution and study the lower bounds of $V(p_{\lambda G}, p_{\lambda^* G_*})$ in terms of their parameters.

**Theorem A.5.** *Assume that $h_0$ takes the form (7) and $\bar{k} = k_0$. Besides that, $f$ is location-scale Gaussian distribution. Then, for $\tilde{k} := \max\{k_* - k_0, 1\}$, and for any $\lambda \in [0, 1]$ and $G \in \mathcal{O}_{K, c_0}(\Theta)$ for some $K \geq k_*$ and $c_0 > 0$, there exist positive constants $C_1$ and $C_2$ depending only on $\lambda^*, G_*, G_0, \Theta$ such that on $\lambda^*, G_*, G_0, \Theta$ such that*

(a) *If $\mathcal{I}(\lambda)$ is not ratio-independent, then*

$$V(p_{\lambda^* G_*}, p_{\lambda G}) \geq C_1 \left[ 1_{\{\lambda \in \mathcal{B}^c\}} \right.$$
$$\left. + 1_{\{\lambda \in \mathcal{B}\}} W_{\overline{r}(K-\tilde{k})}^{\overline{r}(K-\tilde{k})}(G, \overline{G}_*(\lambda)) \right]. \tag{14}$$

(b) *If $\mathcal{I}(\lambda)$ is ratio-independent, then*

$$V(p_{\lambda^*, G_*}, p_{\lambda, G}) \geq C_2 \left[ 1_{\{\lambda \in \mathcal{B}^c\}} \left( \sum_{i \in \mathcal{I}(\lambda)} \left[ (\lambda^* - \lambda) p_i^0 \right. \right. \right.$$
$$\left. \left. - \lambda^* p_i^* \right] + \mathcal{S}(\mathcal{I}(\lambda)) W_{\overline{r}(K-\tilde{k})}^{\overline{r}(K-\tilde{k})}(G, \widetilde{G}_*(\lambda)) \right)$$
$$\left. + 1_{\{\lambda \in \mathcal{B}\}} W_{\overline{r}(K-\tilde{k})}^{\overline{r}(K-\tilde{k})}(G, \overline{G}_*(\lambda)) \right]. \tag{15}$$

# B   Additional Experiment

We provide a simulation experiment with partially distinguishable setting in this section to demonstrate the theoretical results in Section 3.4.

**Partially distinguishable setting.** Consider the partial distinguishable case as in Theorem A.3 with weakly identifiable $f$, we will conduct an experiment to distinguish two regimes in part (b) and (c) of the theorem, which are $\lambda > \lambda^*$ and $\lambda \leq \lambda^*$. We simulate $n$ data from the true data generating model (1), where $p_1^0 = 0.4, p_2^0 = 0.6, p_1^* = 1, \lambda^* = 0.3, \mu_1^0 = \mu_1^* = (-2, 3), \Sigma_1^0 = \Sigma_1^* = \begin{pmatrix} 3 & -1 \\ -1 & 2 \end{pmatrix}, \mu_2^0 = (1, -4), \Sigma_2^0 = \begin{pmatrix} 1 & 0 \\ 0 & 4 \end{pmatrix}$. In this case, $k_* = 1, k_0 = 2, \bar{k} = 1, k_* + k_0 - \bar{k} = 2$ and we will fit the data with model $p_{\lambda G}$, where $G$ has 3 atoms. The MLE $(\hat\lambda_n, \hat{G}_n)$ is found by the EM algorithm. In the regime $\hat\lambda_n < \lambda^*$, we see that $\hat\lambda_n \to \lambda^*$ in the parametric rate and the convergence of $\hat{G}_n$ to $G_*$ is of order $(\log(n)/n)^{2\bar{r}(K - k_*)} = (\log(n)/n)^{12}$ (Fig. 2). When $\hat\lambda_n > \lambda^*$, because of the indistinguishability of the model, we do not expect $\hat\lambda_n \to \lambda^*$ but the Wasserstein distance between $\hat{G}_n$ and $\overline{G}_*(\hat\lambda_n)$ converges to 0 with the rate $(\log(n)/n)^{2\bar{r}(2)} = (\log(n)/n)^{1/8}$. The simulation study matches with this result, where $\hat\lambda_n$ converges to some number greater than $\lambda^*$, and the rate that $W_4(G, \overline{G}_*(\hat\lambda_n))$ converges to 0 is of order $(\log(n)/n)^{1/8}$ (Fig. 3).

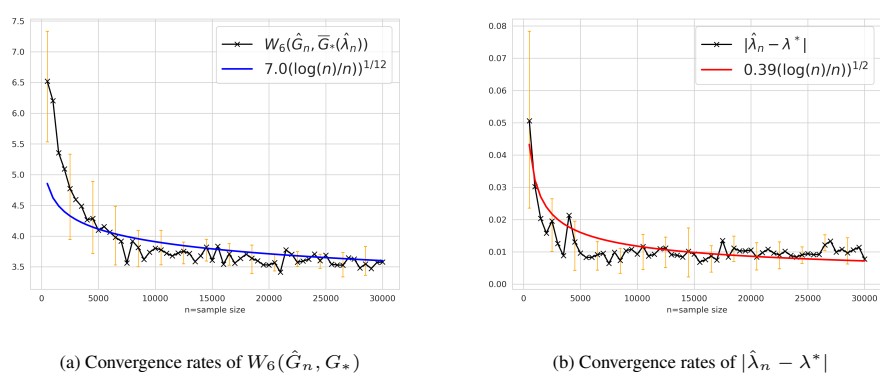

(a) Convergence rates of $W_6(\hat{G}_n, G_*)$ $\qquad\qquad$ (b) Convergence rates of $|\hat\lambda_n - \lambda^*|$

Figure 2: Parameter learning rates in regime $\lambda \leq \lambda^*$.

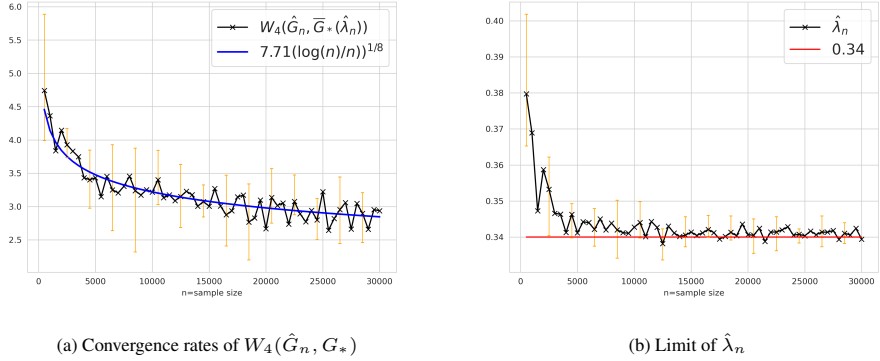

(a) Convergence rates of $W_4(\hat{G}_n, G_*)$ $\qquad\qquad$ (b) Limit of $\hat\lambda_n$

Figure 3: Parameter learning rates in regime $\lambda > \lambda^*$.

# C Proofs of Section 2

## C.1 Proof of Theorem 2.4

(a) We first prove that $h_0$ is distinguishable with $(f, k)$ up to first order with any $k$ and $f$ being location-scale Gaussian family, i.e., if there exists $\lambda, \alpha_j \in \mathbb{R}, \beta_j \in \mathbb{R}^d$, symmetric matrices $\gamma_i \in \mathbb{R}^{d \times d}$, $\theta_j \in \mathbb{R}^d$, and positive definite symmetric $\Sigma_j \in \mathbb{R}^{d \times d}$ for $j = 1, \ldots, k$ such that

$$\lambda h_0(x) + \sum_{j=1}^{k} \alpha_j f(x|\theta_j, \Sigma_j) + \sum_{j=1}^{k} \beta_j^T \frac{\partial f}{\partial \theta}(x|\theta_j, \Sigma_j) + \mathrm{tr}\left( \frac{\partial f}{\partial \Sigma}(x|\theta_j, \Sigma_j)^T \gamma_j \right) = 0,$$

then $\lambda = \alpha_j = \beta_j = \gamma_j = 0$ for all $j = 1, \ldots, k$, where $f(x|\theta, \Sigma)$ is the density evaluated at $x$ of Gaussian distribution with mean $\theta$ and covariance $\Sigma$ and $(\theta_j, \Sigma_j)_{j=1}^{k}$ are pairwise different. Suppose there exists such $(\lambda, \alpha_j, \beta_j, \gamma_j)_{j=1}^{k}$. We borrow a technique from [17, 28], where we find a one-dimensional space to project $x \in \mathbb{R}^d$ onto and work with the order of means and variances in that space to show that the solution must be trivial. Calculating the first derivatives of $f$ gives

$$\lambda h_0(x) + \sum_{j=1}^{k} \left( \alpha_j' + (\beta_j')^T(x - \theta_j) + (x - \theta_j)^T \gamma_j^{-1}(x - \theta_j) \right) e^{-\frac{1}{2}(x-\theta_j)^T \Sigma_j^{-1}(x-\theta_j)} = 0, \quad (16)$$

where

$$\alpha_j' = \frac{2\alpha_j - \mathrm{tr}(\Sigma_j^{-1}\gamma_j)}{2\pi^{d/2}|\Sigma_j|^{1/2}}, \quad \beta_j' = \frac{2}{\pi^{d/2}|\Sigma_j|^{1/2}}\Sigma_j^{-1}\beta_j, \quad \gamma_j' = \frac{1}{\pi^{d/2}|\Sigma_j|^{1/2}}\Sigma_j^{-1}\gamma_j\Sigma_j^{-1},$$

for all $j = 1, \ldots, k$. If all the covariance matrices are equal, i.e., $\Sigma_1 = \cdots = \Sigma_k$, then $(\theta_j)_{j=1}^{k}$ are pairwise different. Denote by $\delta_{ij} = \theta_i - \theta_j$, then for any $x' \notin \cup_{1 \leq i \leq j \leq k}\{u \in \mathbb{R}^d : \delta_{ij}^T u = 0\}$, we have $(x')^T\theta_1, \ldots, (x')^T\theta_k$ are distinct. Otherwise, if (without loss of generality) there are $\Sigma_1, \ldots, \Sigma_m$ different matrices among $\Sigma_1, \ldots, \Sigma_k$, then for every $x' \notin \cup_{1 \leq i \leq j \leq m}\{u \in \mathbb{R}^d : u^T(\Sigma_i - \Sigma_j)u = 0\}$, we have $(x')^T\Sigma_1(x'), \ldots, (x')^T\Sigma_m(x')$ are distinct. In both cases, we find a finite collection of hyperplanes and cones such that for every $x'$ not belongs to any set of this collection, we have $((x')^T\theta_1, (x')^T\Sigma_1(x')), \ldots, ((x')^T\theta_k, (x')^T\Sigma_k(x'))$ are pairwise different. Note that because the union of these collection of $(d-1)$ dimensional manifolds can not be $\mathbb{R}^d$, such a non-zero $x'$ exists. Now we only consider $x$ belongs to the one-dimensional linear space spanned by this $x'$, i.e., $x = y(x')$, where $y \in \mathbb{R}$. Denote by

$$a_j = (x')^T\gamma_j'x', \quad b_j = [(\beta_j')^T - 2\theta_j^T\gamma_j']x', \quad c_j = \theta_j^T\gamma_j'\gamma_j - (\beta_j')^T\theta_j,$$

$$d_j = (x')^T\Sigma_j^{-1}x', \quad e_j = (x')^T\Sigma_j^{-1}\theta_j', \quad f_i = \theta_j^T\Sigma_j^{-1}\theta_j,$$

for $j = 1, \ldots, k$, we proved that $((d_j, e_j))_{j=1}^{k}$ are distinct. Equation (16) implies that

$$\lambda h_0(yx') + \sum_{j=1}^{k}(\alpha_j' + a_jy^2 + b_jy + c_j)\exp(d_jy^2 + e_jy + f_j) = 0. \quad (17)$$

**Case 1.** If $-\log h_0(x) \gtrsim \|x\|_2^{\beta_1}$ for some $\beta_1 > 2$ and for all $\|x\|_2 > x_0$, we have $h_0(x) \lesssim \exp^{-\|x\|_2^{\beta_1}}$. Choose $d_{i_1} = \max_{1 \leq i \leq k} d_k$ and $e_{i_2} = \max\{e_j : d_j = d_{j_1}\}$. Because $h_0$ has a lighter tail than Gaussian and

$$d_jy^2 + e_jy + f_j < d_{i_2}y^2 + e_{i_2}y + f_{i_2}, \quad \forall j \neq i_2,$$

for all $y$ large enough, divide both sides of (17) by $\exp(d_{i_2}y^2 + e_{i_2}y + f_{i_2})$ and let $y \to \infty$, we have $a_{i_2} = b_{i_2} = 0$. It implies that $(x')^T\gamma_{i_2}'x' = [(\beta_{i_2}')^T - 2\theta_{i_2}^T\gamma_{i_2}']x' = 0$. If $\gamma_{i_2}' \neq 0$ then we can further choose $x'$ outside a cone such that $(x')^T\gamma_{i_2}'x' \neq 0$. Hence, $\gamma_{i_2} = 0$, which implies $(\beta_{i_2}')^T(x') = 0$. If $\beta_{i_2} \neq 0$ then we can further choose $x'$ outside a hyperplane such that $(\beta_{i_2}')^T(x') \neq 0$. Hence, in any case, we can argue so that $\beta_{i_2}' = \theta_{i_2}' = 0$. Put it back to (17), we also have $\alpha_{i_2}' = 0$. Therefore, $\alpha_{i_2} = \beta_{i_2} = \gamma_{i_2} = 0$. Repeat the same argument, notice that the tail of $h_0$ is lighter than any Gaussian distribution, we have $\alpha_j = \beta_j = \gamma_j = 0$ for all $j = 1, \ldots, k$. It finally leads to $\lambda = 0$. Hence, we have the distinguishability of $h_0$ with family of location-scale Gaussians up to first order.

**Case 2.** If $-\log h_0(x) \lesssim \|x\|_2^{\beta_2}$ for some $\beta_2 < 2$ and for all $\|x\|_2 > x_0$. We have $p(x|\theta_j, \Sigma_j)/h_0(x) \to 0$ as $x \to \infty$ for all $j = 1, \ldots, k$. Therefore, dividing both sides of (16) by $h_0(x)$ and let $x \to \infty$ by some direction, we have $\lambda = 0$. Now proceed to argue similar to Case 1, we also have the distinguishability of $h_0$ with family of location-scale Gaussians up to first order.

Now we proceed to prove that $h_0$ is distinguishable with $(f, k)$ up to the any order, for $f$ being family of location Gaussian and any $k > 0$. Arguing similar to above, we only need to work on one-dimensional space. Suppose that there exists $\lambda, (c_{i,j})_{i=1,\ldots,k, j=1,\ldots,r}$ such that

$$\lambda h_0(x) + \sum_{i=1}^{k} \sum_{j=0}^{r} c_{i,j} \frac{\partial^j f}{\partial \theta^j}(x|\theta_i, v_i) = 0, \tag{18}$$

where $f(\cdot|\theta, v)$ is the density function of normal distribution with mean $\theta$ and variance $v$, and $(\theta_1, v_1), \ldots, (\theta_k, v_k)$ are distinct. We need to prove that $\lambda = c_{i,j} = 0$ for all $i = 1, \ldots, k, j = 1, \ldots, r$. Calculating the partial derivatives of $f$, we have

$$\lambda h_0(x) + \sum_{i=1}^{k} \left( \sum_{j=0}^{r} \gamma_{i,j}(x - \theta_i)^j \right) \exp\left( -\frac{(x - \theta_i)^2}{2v_i} \right) = 0, \tag{19}$$

such that $\gamma_{i,j}$ for odd j are linear combination of $(c_{i,l})$ with odd $l \leq j$, $\gamma_{i,j}$ for even j are linear combination of $(c_{i,l})$ with even $l \leq j$, and one can prove (for example, by induction) that $\gamma_{i,j} = 0 \forall j$ is equivalent to $c_{i,j} = 0 \forall j$. Now we can argue similar to Case 1 and Case 2 above to get the contradiction, with the notice that polynomials grow slower than exponential functions.

(b) Let $T$ be a piecewise linear function with a positive finite number of breakpoints and $h_0$ is the density function of $N(0, I_d)$ being pushforwarded by $T$. Argue similar to above, we only need to prove the result in one-dimensional case. In order to prove the distinguishable of $h_0$ with mixtures of location Gaussians family or mixtures of location-scale Gaussians family, it all boils down to prove that if there exists $\lambda \in \mathbb{R}$ and polynomials $Q_1(x), Q_2(x), \ldots, Q_k(x)$ such that

$$\lambda h_0(x) + \sum_{i=1}^{k} Q_i(x) f(x|\theta_i, v_i^2) = 0, \tag{20}$$

where $(\theta_1, v_1^2), \ldots, (\theta_k, v_k^2)$ are distinct, then $\lambda = Q_1(x) = \cdots = Q_k(x) = 0$. We will prove this by induction in $k$. Consider the case $k = 1$, we have

$$\lambda h_0(x) + Q_1(x) f(x|\theta_1, v_1^2) = 0. \tag{21}$$

Because $T$ has finite number of break points, there exists some $x_0$ large enough so that for all $x > x_0$, $T$ is a linear one-to-one function between $[x_0, \infty)$ and its image. Denote by $T(x) = ax + b$ when $x > x_0$. We can argue that $a \neq 0$, because otherwise the distribution of $h_0$ will has an atom, which directly leads to distinguishability between $h_0$ and mixtures of Gaussians. Then, $h_0(x) = f(x|b, a^2)$ and we have
$$\lambda f(x|b, a^2) + Q_1(x) f(x|\theta_1, v_1^2) = 0.$$
Argue similar to part (a), if $(b, a^2) \neq (\theta_1, v_1^2)$, we have $\lambda = Q_1(x) = 0$, which implies the distinguishability. Otherwise, we have $b = \theta_1, a^2 = v_1^2$, and $Q_1(x) = -\lambda$ for all $x \in \mathbb{R}$. We can rewrite (21) as
$$h_0(x) - f(x|\theta_1, v_1^2) = 0.$$
Because $h_0$ is $N(0, 1)$ being pushforwarded by a piecewise linear function, we can write $\mathbb{R}$ as a partition $(-\infty, c_1], (c_1, c_2], \ldots, [c_m, \infty)$ such that each semi-open interval is image of some linear functions of $T$. Consider a semi-open interval $(c_i, c_{i+i}]$ being image of $T_j(z) = a_j z + b_j$ for $j = 1, \ldots, h$, by the change of variable formula for many-to-one map, we have

$$0 = h_0(x) - f(x|\theta_1, v_1^2) = \sum_{j=1}^{h} f(x|b_j, a_j^2) - f(x|\theta_1, v_1^2), \tag{22}$$

for all $x \in (c_i, c_{i+i}]$. Applying Lemma C.1, we have equation (22) is true for all $x \in \mathbb{R}$. Hence, by integrating both side, we get $h = 1$, and then $b_1 = \theta_1, a_1^2 = v_1^2$. Because this is true for all semi-open intervals $(c_i, c_{i+i}]$, we have $T(x) = a_1 x + b_1$ for all $x \in \mathbb{R}$, which is contradict to our assumption that $T$ is non-linear.

Suppose that our inductive hypothesis is correct for $k = n$, now we proceed to prove it is true for $k = n + 1$. If there exists $\lambda \in \mathbb{R}$ and polynomials $Q_1(x), Q_2(x), \ldots, Q_{n+1}(x)$ such that

$$\lambda h_0(x) + \sum_{i=1}^{n+1} Q_i(x) f(x|\theta_i, v_i^2) = 0, \tag{23}$$

where $(\theta_1, v_1), \ldots, (\theta_{n+1}, v_{n+1}^2)$ are distinct. Without loss of generality, assume that $v_1^2 = \max_{1 \leq i \leq n+1} v_k^2$ and $\theta_1 = \max\{\theta_j : v_j^2 = v_1^2\}$. Because $T$ has finite number of break points, there exists some $x_0$ large enough so that for all $x > x_0$, $T$ is a linear one-to-one function between $[x_0, \infty)$ and its image. Denote by $T(x) = ax + b$ when $x > x_0$. We have

$$\lambda f(x|b, a^2) + \sum_{i=1}^{n+1} Q_i(x) f(x|\theta_i, v_i^2) = 0, \quad \forall x > x_0. \tag{24}$$

If $a^2 > v_1^2$ or $a^2 = v_1^2, b > \theta_1$, divide both sides of equation (24) by $\exp((x - b)/2a^2)$ and let $x \to \infty$, we have $\lambda = 0$ and the conclusion follows from the identifiability of Gaussians family.

If $v_1^2 > a^2$ or $v_1^2 = a^2, \theta_1 > b$, divide both sides of equation (24) by $\exp((x - \theta_1)/2v_1^2)$ and let $x \to \infty$, we have $Q_1(x) = 0$. The problem is back to the case $k = n$ and is proved using the inductive hypothesis.

If $a^2 = v_1^2, b = \theta_1$, divide both sides of equation (24) by $\exp((x - b)/2a^2)$ and let $x \to \infty$, we have $Q_1(x) = -\lambda$ for all $x \in \mathbb{R}$. Hence for $x$ large enough,

$$\sum_{i=2}^{n+1} Q_i(x) f(x|\theta_i, v_i^2) = 0,$$

which implies $Q_2(x) = \cdots = Q_{n+1}(x) = 0$. The problem is back to the case $k = 1$ and is proved using the inductive hypothesis.

The following lemma presents the local identifiability of location-scale Gaussians mixtures.

**Lemma C.1.** *Denote by $f(\cdot|\theta, \sigma^2)$ the density function of Gaussian distribution with mean $\theta$ and variance $\sigma^2$. For all $a < b$ and pairs $\{(\theta_i, \sigma_i^2)\}_{i=1}^k$, if there exists $\alpha_1, \alpha_2, \ldots, \alpha_n \in \mathbb{R}$ such that*

$$\alpha_1 f(x|\theta_1, \sigma_1^2) + \cdots + \alpha_k f(x|\theta_k, \sigma_k^2) = 0$$

*for all $x \in [a, b]$, then*

$$\alpha_1 f(x|\theta_1, \sigma_1^2) + \cdots + \alpha_k f(x|\theta_k, \sigma_k^2) = 0, \tag{25}$$

*for all $x \in \mathbb{R}$.*

*Proof.* Step 1. (Centralize and normalize coefficients). Suppose that there exists $\alpha_1, \alpha_2, \ldots, \alpha_n \in \mathbb{R}$ such that

$$\alpha_1 f(x|\theta_1, \sigma_1^2) + \cdots + \alpha_k f(x|\theta_k, \sigma_k^2) = 0$$

for all $x \in [a, b]$. Denote by $\theta_i' = \theta_i - \dfrac{a+b}{2}$ for all $i = 1, \ldots, k$, then

$$\alpha_1 \frac{1}{\sqrt{2\pi}\sigma_1} \exp\left(-\frac{(x - \theta_1')^2}{2\sigma_1^2}\right) + \cdots + \alpha_k \frac{1}{\sqrt{2\pi}\sigma_k} \exp\left(-\frac{(x - \theta_k')^2}{2\sigma_k^2}\right) = 0, \tag{26}$$

for all $x \in [-\frac{b-a}{2}, \frac{b-a}{2}]$. Denote by $\sigma_{i_1} = \min\{\alpha_1, \ldots, \alpha_k\}$. Multiple both sides of (26) by $\exp(-\frac{x^2}{\sigma_{i_1}^2})$, and denote by $s_i^2 = \frac{1}{\sigma_{i_1}^2} - \frac{1}{2\sigma_i^2}, m_i = \theta_i'/\sigma_i^2, \beta_i = \dfrac{1}{\sqrt{2\pi}\sigma_i} \exp(-(\theta_i')^2/2\sigma_i^2)$ for all $i = 1, \ldots, k$, we have

$$\beta_1 \exp\left(s_1^2 x^2 + m_1 x\right) + \cdots + \beta_k \exp\left(s_k^2 x^2 + m_k x\right) = 0, \tag{27}$$

for all $x \in [-\frac{b-a}{2}, \frac{b-a}{2}]$.

*Step 2. (Use properties of Laplace transformation).* The left-hand side of equation (27) is the Laplace transformation of $\sum_{i=1}^k \beta_i f(x|m_i, s_i^2)$ and is identical to 0 in an open set around 0. Hence

$$\sum_{i=1}^k \beta_i f(x|m_i, s_i^2) = 0,$$

for all $x \in \mathbb{R}$. It implies that

$$\beta_1 \exp\left(s_1^2 x^2 + m_1 x\right) + \cdots + \beta_k \exp\left(s_k^2 x^2 + m_k x\right) = 0,$$

for all $x \in \mathbb{R}$, which is equivalent to equation (25). $\qquad\square$

## C.2 Proof of Proposition 2.5

If $k_\sigma$ is the Gaussian kernel with $m > K$, then we get the conclusions as direct consequences of Example 2.3(a). If $k_\sigma$ is the multivariate Student kernel, then $h_0$ has a tail heavier than Gaussian tail, so that we get the conclusions as consequences of Proposition 2.4(a).

## C.3 Proof of Proposition 2.6

Because $T$ has a finite and postive number of layers, it is a piecewise linear and non-linear function. So the conclusions are direct consequences of Proposition 2.4(b).

## C.4 Proof of Theorem 3.1

This result can be obtained by modifying the proof of Theorem 7.4 in [26]. Recall that we defined the function class

$$\overline{\mathcal{P}}_k^{1/2}(\Theta, \epsilon) = \left\{ \bar{p}_{\lambda G}^{1/2} : G \in \mathcal{O}_k(\Theta),\ h(\bar{p}_{\lambda G}, p_{\lambda^* G_*}) \le \epsilon \right\}, \tag{28}$$

where for any $G \in \mathcal{O}_K(\Theta)$, we write $\bar{p}_{\lambda G} = (p_{\lambda G} + p_{\lambda^* G_*})/2$, and measure the complexity of this class through the bracketing entropy integral

$$\mathcal{J}_B(\epsilon, \overline{\mathcal{P}}_k^{1/2}(\Theta, \epsilon), \nu) = \int_{\epsilon^2/2^{13}}^\epsilon \sqrt{\log N_B(u, \overline{\mathcal{P}}_k^{1/2}(\Theta, u), \nu)}\,du \vee \epsilon,$$

where $N_B(\epsilon, X, \eta)$ denotes the $\epsilon$-bracketing number of a metric space $(X, \eta)$ and $\nu$ is the Lebesgue measure. We denote by $P_{\lambda G}$ the distribution corresponding to the density $p_{\lambda G}$. The technique to prove this theorem is to bound the convergence rate by the increments of an empirical processes:

$$\nu_n(\lambda G) = \sqrt{n} \int_{\{p_{\lambda^* G_*}\}>0} \frac{1}{2} \log \frac{\bar{p}_{\lambda G}}{p_{\lambda^* G_*}} d(P_n - P_{\lambda^* G_*}),$$

where $P_n = \frac{1}{n} \sum_{i=1}^n \delta_{X_i}$ is the empirical measure $(X_1, \ldots, X_n \overset{iid}{\sim} p_{\lambda^* G_*})$. We first recall Theorem 5.11 in [26] with the notations adapted from our setting:

**Theorem C.2.** *Let $R > 0$, $k \ge 1$, and $\mathcal{G}$ be a subset of $\mathcal{O}_k(\Theta)$, which contains $G_*$. Given $C_1 < \infty$, for all $C$ sufficiently large, and for $n \in \mathbb{N}$ and $t > 0$ satisfying*

$$t \le \sqrt{n}((8R) \wedge (C_1 R^2)), \tag{29}$$

*and*

$$t \ge C^2(C_1 + 1)\left(R \vee \int_{t/(2^6\sqrt{n})}^R H_B^{1/2}\left(\frac{u}{\sqrt{2}}, \overline{\mathcal{P}}_k^{1/2}(\Theta, R), \nu\right) du\right), \tag{30}$$

*we have*

$$\mathbb{P}_{\lambda^* G_*}\left(\sup_{G \in \mathcal{G}, h(\bar{p}_{\lambda G}, p_{\lambda^* G_*}) \le R} |\nu_n(\lambda G)| \ge t\right) \le C \exp\left(-\frac{t^2}{C^2(C_1 + 1)R^2}\right). \tag{31}$$

Now we proceed to prove Theorem 3.1, the proof is divided into three parts: Bounding the tail probability of $h(p_{\hat{\lambda}_n \hat{G}_n}, p_{\lambda^* G_*})$ by sums of empirical processes increments using chaining technique, bounding the empirical processes increments using Theorem C.2, and bounding the expectation of $h(p_{\hat{\lambda}_n \hat{G}_n}, p_{\lambda^* G_*})$ using its tail probability.

**Step 1 (Bounding the tail probability $h(p_{\hat{\lambda}_n \hat{G}_n}, p_{\lambda^* G_*})$ by sums of empirical processes increments):** Firstly, by Lemma 4.1 and 4.2 of [26], we have

$$\frac{1}{16} h^2(p_{\hat{\lambda}_n \hat{G}_n}, p_{\lambda^* G_*}) \leq h^2(\overline{p}_{\hat{\lambda}_n \hat{G}_n}, p_{\lambda^* G_*}) \leq \frac{1}{\sqrt{n}} \nu_n(\hat{\lambda}_n \hat{G}_n).$$

Hence, for any $\delta > \delta_n := (\log n/n)^{1/2}$, we have

$$\mathbb{P}_{\lambda^* G_*}(h(p_{\hat{\lambda}_n \hat{G}_n}, p_{\lambda^* G_*}) \geq \delta) \leq \mathbb{P}_{\lambda^* G_*}\left( \nu_n(\hat{\lambda}_n \hat{G}_n) - \sqrt{n} h^2(\overline{p}_{\hat{\lambda}_n \hat{G}_n}, p_{\lambda^* G_*}) \geq 0, \right.$$

$$\left. h(\overline{p}_{\hat{\lambda}_n \hat{G}_n}, p_{\lambda^* G_*}) \geq \delta/4 \right)$$

$$\leq \mathbb{P}_{\lambda^* G_*}\left( \sup_{\lambda, G: h(\overline{p}_{\lambda G}, p_{\lambda^* G_*}) \geq \delta/4} [\nu_n(\lambda G) - \sqrt{n} h^2(\overline{p}_{\lambda G}, p_{\lambda^* G_*})] \geq 0 \right)$$

$$\leq \sum_{s=0}^{S} \mathbb{P}_{\lambda^* G_*}\left( \sup_{\lambda, G: 2^s \delta/4 \leq h(\overline{p}_{\lambda G}, p_{\lambda^* G_*}) \leq 2^{s+1} \delta/4} |\nu_n(\lambda G)| \geq \sqrt{n} 2^{2s} (\delta/4)^2 \right)$$

$$\leq \sum_{s=0}^{S} \mathbb{P}_{\lambda^* G_*}\left( \sup_{\lambda, G: h(\overline{p}_{\lambda G}, p_{\lambda^* G_*}) \leq 2^{s+1} \delta/4} |\nu_n(\lambda G)| \geq \sqrt{n} 2^{2s} (\delta/4)^2 \right),$$

where $S$ is a smallest number such that $2^S \delta/4 > 1$, as $h(\overline{p}_{\lambda G}, p_{\lambda^* G_*}) \leq 1$. Now we will bound the each term above using Theorem C.2.

**Step 2 (Bounding the empirical processes increments using Theorem C.2):** In Theorem C.2, choose $R = 2^{s+1} \delta, C_1 = 15$ and $t = \sqrt{n} 2^{2s} (\delta/4)^2$, we can readily check that Condition (29) satisfies (because $2^{s-1} \delta/4 \leq 1$ for all $s = 0, \ldots, S$). Condition (30) satisfies thanks to Assumption A3:

$$\int_{t/(2^6 \sqrt{n})}^{R} H_B^{1/2}\left( \frac{u}{\sqrt{2}}, \mathcal{P}_k^{1/2}(\Theta, R), \nu \right) du \vee 2^{s+1} \delta = \sqrt{2} \int_{R^2/2^{13}}^{R/\sqrt{2}} H_B^{1/2}\left( u, \mathcal{P}_k^{1/2}(\Theta, R), \nu \right) du \vee 2^{s+1} \delta$$

$$\leq 2 \mathcal{J}_B(R, \mathcal{P}^{1/2}(\Theta, R), \nu)$$

$$\leq 2 J \sqrt{n} 2^{2s+1} \delta^2 = 2^6 J t.$$

So the conclusion of Theorem C.2 gives us

$$\mathbb{P}_{\lambda^* G_*}(h(p_{\hat{\lambda}_n \hat{G}_n}, p_{\lambda^* G_*}) > \delta) \leq C \sum_{s=0}^{\infty} \exp\left( \frac{2^{2s} n \delta^2}{J^2 2^{14}} \right) \leq c \exp\left( \frac{n \delta^2}{c^2} \right), \qquad (32)$$

where $c$ is a large constants that does not depend on $\lambda^*, G_*$.

**Step 3 (Implying the bound on supremum of expectation):** Thus, we have

$$\mathbb{E} h(p_{\hat{\lambda}_n \hat{G}_n}, p_{\lambda^* G_*}) = \int_0^\infty \mathbb{P}(h(p_{\hat{\lambda}_n \hat{G}_n}, p_{\lambda^* G_*}) > \delta) d\delta \leq \delta_n + c \int_{\delta_n}^\infty \exp\left( -\frac{n \delta^2}{c^2} \right) \leq \tilde{c} \delta_n,$$

for some $\tilde{c}$ does not depend on $\lambda^*, G_*$. Hence, we finally proved that

$$\sup_{G_* \in \mathcal{O}_k(\Theta), \lambda^* \in [0,1]} \mathbb{E}_{\lambda^*, G_*} h(p_{\hat{\lambda}_n \hat{G}_n}, p_{\lambda^* G_*}) \leq C \sqrt{\log n/n}.$$

As a consequence, we obtain the conclusion of the theorem.

# D  Proof of Section 3

## D.1  Proof of Proposition 3.2

We first need to denote some notations that are required for the proof. Those notations are well-known in Empirical Processes field [26]. Denote by

$$\mathcal{P}_k(\Theta) = \{p_{\lambda G} : \lambda \in [0,1], G \in \mathcal{O}_k(\Theta)\},$$

and let $N(\epsilon, \mathcal{P}_k(\Theta), \|\cdot\|_\infty)$ be the $\epsilon-$covering number of $(\mathcal{P}_k(\Theta), \|\cdot\|_\infty)$ and $N_B(\epsilon, \mathcal{P}_k(\Theta), h)$ be the bracketing number of $\mathcal{P}_k(\Theta)$ measured by Hellinger metric $h$. $H_B(\epsilon, \mathcal{P}_k(\Theta), h) = \log N_B(\epsilon, \mathcal{P}_k(\Theta), h)$ is called the bracketing entropy of $\mathcal{P}_k(\Theta)$ under metric $h$. Let $\overline{\mathcal{P}}_k(\Theta) = \{(p_{\lambda G} + p_{\lambda^* G_*})/2 : \lambda \in [0,1], G \in \mathcal{O}_k(\Theta)\}$ and $\overline{\mathcal{P}}_k^{1/2}(\Theta) = \{p^{1/2} : p \in \overline{\mathcal{P}}_k(\Theta)\}$. We want to show that

$$\mathcal{J}_B(\epsilon, \overline{\mathcal{P}}_k^{1/2}(\Theta, \epsilon), L^2(\mu)) = \left( \int_{\epsilon^2/2^{13}}^\epsilon H_B^{1/2}(\delta, \overline{\mathcal{P}}_k^{1/2}(\Theta, \delta), \nu)d\delta \vee \delta \right) \lesssim \sqrt{n}\epsilon^2, \qquad (33)$$

for all $n > N$ large enough and $\epsilon > (\log n/n)^{1/2}$. We proceed to show that claim (33) will be proved if

$$\log N(\epsilon, \mathcal{P}_k(\Theta), \|\cdot\|_\infty) \lesssim \log(1/\epsilon), \qquad (34)$$

$$H_B(\epsilon, \mathcal{P}_k(\Theta), h) \lesssim \log(1/\epsilon), \qquad (35)$$

and then prove claim (34) and (35).

**Proof of that claim** (35) **implies claim** (33)    Because $\overline{\mathcal{P}}_k^{1/2}(\Theta, \delta) \subset \overline{\mathcal{P}}_k^{1/2}(\Theta)$ and from the definition of Hellinger distance,

$$H_B(\delta, \overline{\mathcal{P}}_k^{1/2}(\Theta, \delta), \mu) \leq H_B(\delta, \overline{\mathcal{P}}_k^{1/2}(\Theta), \mu) = H_B(\frac{\delta}{\sqrt{2}}, \overline{\mathcal{P}}_k(\Theta), h).$$

Now use the fact that for densities $f_*, f_1, f_2$, we have $h^2((f_1 + f_*)/2, (f_2 + f_*)/2) \leq h^2(f_1, f_2)/2$, oen can readily check that $H_B(\frac{\delta}{\sqrt{2}}, \overline{\mathcal{P}}_k(\Theta), h) \leq H_B(\delta, \mathcal{P}_k(\Theta), h)$. Hence, if claim (35) holds true, then

$$H_B(\delta, \overline{\mathcal{P}}_k^{1/2}(\Theta, \delta), \mu) \leq H_B(\delta, \mathcal{P}_k(\Theta), h) \lesssim \log(1/\delta),$$

which implies that

$$\mathcal{J}_B(\epsilon, \overline{\mathcal{P}}_k^{1/2}(\Theta, \delta), \mu) \lesssim \epsilon(\log(2^{13}/\epsilon^2))^{1/2} < n\epsilon^2,$$

for all $\epsilon > (\log n/n)^{1/2}$. Hence, claim (33) is proved.

**Proof of claim** (34)    By invoking the proof of Lemma 2.1. of [16], we have a $\epsilon$-net $\mathcal{S}$ for $(\{p_G : G \in \mathcal{O}_k(\Theta), h\})$ with the cardinality being bounded as follows

$$|\mathcal{S}| \leq \left( \frac{2d\overline{\lambda}}{\epsilon} \right)^{d(d+1)k/2} \times \left( \frac{2a}{\epsilon} \right)^{dk} \left( \frac{5}{\epsilon} \right)^k.$$

Denote by $\mathcal{G}$ the set of latent mixing measures $G$ in that net. Let $\mathcal{S}_0$ be an $\epsilon-$net in $[0,1]$ for $\lambda$, it is seen that $|\mathcal{S}_0| \leq 1/\epsilon$. Now we form a net for $\mathcal{P}_k(\Theta)$ by $\{p_{\lambda G} : \lambda \in \mathcal{S}_0, G \in \mathcal{G}\}$. Hence, for any $\lambda, G$, there exists $\tilde{\lambda} \in \mathcal{S}_0, G \in \mathcal{G}$ such that

$$|\lambda - \tilde{\lambda}| \leq \epsilon, \|p_G - p_{\tilde{G}}\|_\infty \leq \epsilon.$$

This implies

$$\begin{aligned}
\left\| p_{\lambda G} - p_{\tilde{\lambda}\tilde{G}} \right\|_\infty &\leq \left\| p_{\lambda G} - p_{\tilde{\lambda}G} \right\|_\infty + \left\| p_{\tilde{\lambda}G} - p_{\tilde{\lambda}\tilde{G}} \right\|_\infty \\
&\leq |\lambda - \tilde{\lambda}|(\|h_0\|_\infty + \|p_G\|_\infty) + \tilde{\lambda}\|p_G - p_{\tilde{G}}\|_\infty \\
&\leq \epsilon \left( \|h_0\|_\infty + \frac{1}{(\sqrt{2\pi\underline{\lambda}})^{d/2}} \right) + \epsilon \\
&\lesssim \epsilon.
\end{aligned}$$

Hence, we get an $\epsilon-$net for $\mathcal{P}_k(\Theta)$ with the cardinality less than or equal

$$|\mathcal{S}_0| \times |\mathcal{S}| = \frac{1}{\epsilon} \times \left( \frac{2d\overline{\lambda}}{\epsilon} \right)^{d(d+1)k/2} \times \left( \frac{2a}{\epsilon} \right)^{dk} \left( \frac{5}{\epsilon} \right)^k.$$

Thus,

$$\log N(\epsilon, \mathcal{P}_k(\Theta), \|\cdot\|_\infty) \lesssim \log(1/\epsilon).$$

**Proof of claim** (35)  Now, from the entropy number to get the bracketing number, we let $\eta \leq \epsilon$ which will be chosen later. Let $f_1, \ldots, f_N$ be a $\eta$-net for $\mathcal{P}_k(\Theta)$. We have

$$(x - \theta)^T \Sigma^{-1} (x - \theta) \geq \frac{\|x - \theta\|_2^2}{\bar{\lambda}} \geq \frac{\|x\|_2^2}{4\bar{\lambda}}, \quad \forall \|x\| \geq 2\sqrt{d}a, (\theta, \Sigma) \in \Theta, \tag{36}$$

Moreover, $h_0$ has an exponential tail $-\log h_0(x) \gtrsim \|x\|_2^\beta$ for some $\beta > 0$, and $\|h_0\|_\infty < C$ for some constant $C$. Therefore, if we let $\beta' = \min\{\beta, 2\} > 0$ and $C' = \max\left\{C, \dfrac{1}{(2\pi)^{d/2}\underline{\lambda}^d}\right\}$, then

$$H(x) = \begin{cases} C_1 \exp(-\|x\|_2^{\beta'}), & \|x\|_2 \geq B_1, \\ C', & \text{otherwise} \end{cases} \tag{37}$$

is an envelop for $\mathcal{P}_k(\Theta)$, where $C_1$ depends only on $\underline{\lambda}$ and $h_0$, $B_1$ depends on $a, \bar{\lambda}, h_0$. We can construct brackets $[p_i^L, p_i^U]$ as follows.

$$p_i^L(x) = \max\{f_i(x) - \eta, 0\}, p_i^U(x) = \min\{f_i(x) + \eta, H(x)\}. \tag{38}$$

Because for each $f \in \mathcal{P}_k(\Theta)$, there is $f_i$ such that $\|f - f_i\|_\infty < \eta$, therefore $p_i^L \leq f \leq p_i^U$. Moreover, for any $B \geq B_1$,

$$\int_{\mathbb{R}^d} (p_i^U - p_i^L) d\mu \leq \int_{\|x\|_2 \leq B} 2\eta dx + \int_{\|x\|_2 \geq B} H(x) dx$$
$$\lesssim \eta B^d + B^d \exp\left(-B^{\beta'}\right), \tag{39}$$

where we use spherical coordinate to have

$$\int_{\|x\| \leq B} dx = \frac{\pi^{d/2}}{\Gamma(d/2 + 1)} B^d \lesssim B^d,$$

and

$$\int_{\|x\| \geq B} \exp\left(-\|x\|_2^{\beta'}\right) \lesssim \int_{r \geq B} r^{d-1} \exp\left(-r^{\beta'}\right) dr$$
$$= \frac{1}{\beta} \int_{B^{\beta'}}^\infty u^{d/\beta'-1} \exp(-u) du \quad \text{(change of variable } u = r^{\beta'})$$
$$\leq \frac{1}{\beta'} B^{d-\beta'} \exp(-B^{\beta'}),$$

in which the last step we use the inequality (with change of variable formula)

$$\int_z^\infty u^{d/\beta-1} e^{-u} du = z^{d/\beta} e^{-z} \int_0^\infty (1+s)^{d/\beta-1} e^{-zs} ds \leq z^{d/\beta} e^{-z} \frac{1}{z - d/\beta + 1} < z^{d/\beta} e^{-z}, \tag{40}$$

whenever $z > d/\beta'$, and we use $z = B^{\beta'}$. Hence, in (39), if we choose $B = B_1(\log(1/\eta))^{1/\beta'}$ then

$$\int_{\mathbb{R}^d} (p_i^U - p_i^L) d\mu \lesssim \eta \left(\log\left(\frac{1}{\eta}\right)\right)^{d/\beta'}. \tag{41}$$

Therefore, there exists a positive constant $c$ which does not depend on $\eta$ such that

$$H_B(c\eta \log(1/\eta)^{d/\beta'}, \mathcal{P}_k(\Theta), \|\cdot\|_1) \lesssim \log(1/\eta).$$

Let $\epsilon = c\eta(\log(1/\eta))^{d/\beta'}$, we have $\log(1/\epsilon) \asymp \log(1/\eta)$, which combines with inequality $\|\cdot\|_1 \leq h^2$ leads to

$$H_B(\epsilon, \mathcal{P}_k(\Theta), h) \leq H_B(\epsilon^2, \mathcal{P}_k(\Theta), \|\cdot\|_1) \lesssim \log(1/\epsilon^2) \lesssim \log(1/\epsilon).$$

Thus, we have proved claim (35).

We put a remark here that the technique in this proof can be generalized for any family of $f(x|\theta)$ that have sub-exponential tails, i.e. $f(x|\theta) \lesssim \exp(-\|x\|^\gamma)$ for all $x$ large enough and $\gamma > 0$. We can substitute this condition into equation (36), then proceed to continue the proof similarly.

Next, we provide proofs for inverse bounds in Section 3 of the paper. Because there are several results with the same spirit in this section, to make it easy for reader, we recall each result before proving it.

## D.2 Proof of Theorem 3.3

*Theorem* 3.3. Assume that $k_*$ is known, $f$ is first order identifiable and $(f, k_*)$ is distinguishable from $h_0$. Then, for any $G \in \mathcal{E}_{k_*}(\Theta)$, there exist positive constant $C_1$ and $C_2$ depending only on $\lambda^*, G_*, h_0, \Theta$ such that the following holds:

(a) When $\lambda^* = 0$, then $V(p_{\lambda^* G_*}, p_{\lambda G}) \geq C_1 \lambda$.

(b) When $\lambda^* \in (0, 1]$, then
$$V(p_{\lambda^* G_*}, p_{\lambda G}) \geq C_2 \underbrace{[|\lambda - \lambda^*| + (\lambda + \lambda^*)W_1(G, G_*)]}_{\overline{W}_1(\lambda G, \lambda^* G_*)}.$$

We first provide the proof of the theorem for the setting $\lambda^* \in (0, 1]$ in Section D.2.1. Then, the proof for the setting $\lambda^* = 0$ is presented in Section D.2.2.

### D.2.1 Proof of setting $\lambda^* \in (0, 1]$

Recall that, we define $\overline{W}_1(\lambda G, \lambda^* G_*) := |\lambda - \lambda^*| + (\lambda + \lambda^*)W_1(G, G_*)$. Besides that, $G_* = \sum_{i=1}^{k_*} p_i^* \delta_{\theta_i^*}$. In order to obtain the proof of the theorem for the setting $\lambda^* \in (0, 1]$, it is sufficient to verify the following two claims:

$$\lim_{\epsilon \to 0} \inf_{\lambda \in [0,1], G \in \mathcal{E}_{k_*}(\Theta)} \left\{ \frac{V(p_{\lambda G}, p_{\lambda^* G_*})}{\overline{W}_1(\lambda G, \lambda^* G_*)} : \overline{W}_1(\lambda G, \lambda^* G_*) \leq \epsilon \right\} > 0, \tag{42}$$

$$\inf_{\lambda \in [0,1], G \in \mathcal{E}_{k_*}(\Theta): \overline{W}_1(\lambda G, \lambda^* G_*) > \epsilon'} \frac{V(p_{\lambda G}, p_{\lambda^* G_*})}{\overline{W}_1(\lambda G, \lambda^* G_*)} > 0, \tag{43}$$

for any $\epsilon' > 0$.

**Proof of claim (42):** Assume that claim (42) does not hold. It indicates that there exists a sequence of probability measures $G_n \in \mathcal{E}_{k_*}(\Theta)$ and a sequence of $\lambda_n \in [0, 1]$ such that $\overline{W}_1(\lambda_n G_n, \lambda^* G_*) \to 0$ and $V(p_{\lambda_n G_n}, p_{\lambda^* G_*})/\overline{W}_1(\lambda_n G_n, \lambda^* G_*) \to 0$ as $n \to \infty$. Therefore, we have $\lambda_n \to \lambda^*$ and $W_1(G_n, G_*) \to 0$ as $n \to \infty$. We can relabel the atoms and weights of $G_n$ such that it admits the following form:

$$G_n = \sum_{i=1}^{k_*} p_i^n \delta_{\theta_i^n}, \tag{44}$$

where $p_i^n \to p_i^*$ and $\theta_i^n \to \theta_i^*$ for all $i \in [k_*]$. To ease the ensuing presentation, we denote $\Delta \theta_i^n := \theta_i^n - \theta_i^*$ and $\Delta p_i^n := p_i^n - p_i^*$ for $i \in [k_*]$. Then, using the coupling between $G_n$ and $G_*$ such that it put mass $\min\{p_i^n, p_i^*\}$ on $\delta_{(\theta_i^n, \theta_i^*)}$, we can verify that

$$W_1(G_n, G_*) \asymp \sum_{i=1}^{k_*} |\Delta p_i^n| + p_i^n \|\Delta \theta_i^n\|_2. \tag{45}$$

Our proof is divided into three steps.

**Step 1 - Taylor expansion:** Invoking Taylor expansion up to the first order, we find that
$$f(x|\theta_i^n) = f(x|\theta_i^*) + (\Delta \theta_i^n)^\top \frac{\partial f}{\partial \theta}(x|\theta_i^*) + R_i(x),$$

where $R_i(x)$ is Taylor remainder such that $R_i(x) = o(\|\Delta \theta_i^n\|_2)$ for $i \in [k_*]$. Given the above expressions, we obtain that

$$p_{\lambda_n G_n}(x) - p_{\lambda^* G_*}(x) = (\lambda^* - \lambda_n)h_0(x) + \sum_{i=1}^{k_*} (\lambda_n p_i^n - \lambda^* p_i^*) f(x|\theta_i^*)$$
$$+ \lambda_n p_i^n (\Delta \theta_i^n)^\top \frac{\partial f}{\partial \theta}(x|\theta_i^*) + R(x), \tag{46}$$

where $R(x) = \lambda_n \sum_{i=1}^{n} p_i^n R_i(x) = o\left(\lambda_n \sum_{i=1}^{k_*} p_i^n \|\Delta \theta_i^n\|_2\right)$. From the expression of $W_1(G_n, G_*)$ in (45), we have $R(x)/\overline{W}_1(\lambda_n G_n, \lambda^* G_*) \to 0$ as $n \to \infty$ for all $x$.

**Step 2 - Non-vanishing coefficients:** From equation (46), we can represent the ratio $(p_{\lambda_n G_n}(x) - p_{\lambda^* G_*}(x))/\overline{W}_1(\lambda_n G_n, \lambda^* G_*)$ as a linear combination of elements of $h_0(x)$, $f(x|\theta_i^*)$, $\frac{\partial f}{\partial \theta}(x|\theta_i^*)$ for $i \in [k_*]$. Assume that all of the coefficients associated with these terms go to 0 as $n \to \infty$. As the coefficient with $h_0(x)$ goes to 0, we obtain that $(\lambda^* - \lambda_n)/\overline{W}_1(\lambda_n G_n, \lambda^* G_*) \to 0$ as $n \to \infty$. Furthermore, the coefficients of $f(x|\theta_i^*)$, $\frac{\partial f}{\partial \theta}(x|\theta_i^*)$ vanish to 0 are equivalent to the following limits

$$(\lambda_n p_i^n - \lambda^* p_i^*)/\overline{W}_1(\lambda_n G_n, \lambda^* G_*) \to 0, \quad p_i^n \|\Delta \theta_i^n\|_2/\overline{W}_1(\lambda_n G_n, \lambda^* G_*) \to 0.$$

As we have $(\lambda^* - \lambda_n)/\overline{W}_1(\lambda_n G_n, \lambda^* G_*) \to 0$, the above limits lead to

$$\lambda^* (\Delta p_i^n)/\overline{W}_1(\lambda_n G_n, \lambda^* G_*) \to 0.$$

Putting the above results together, we obtain $1 = \overline{W}_1(\lambda_n G_n, \lambda^* G_*)/\overline{W}_1(\lambda_n G_n, \lambda^* G_*) \to 0$, which is a contraction. As a consequence, not all the coefficients of $h_0(x)$, $f(x|\theta_i^*)$, $\frac{\partial f}{\partial \theta}(x|\theta_i^*)$ go to 0 for $i \in [k_*]$.

**Step 3: Show the contradiction using the distinguishability condition and Fatou's lemma:** Denote $m_n$ as the maximum of the absolute values of the coefficients of $h_0(x)$, $f(x|\theta_i^*)$, $\frac{\partial f}{\partial \theta}(x|\theta_i^*)$ as $i \in [k_*]$. Since not all of these coefficients vanish to 0, we have $m_n \not\to 0$ as $n \to \infty$. Therefore, $d_n = 1/m_n \not\to \infty$ as $n \to \infty$. Given the previous results, there exist $\alpha_0, \alpha_1, \ldots, \alpha_{k_*}$ and $\beta_1, \ldots, \beta_{k_*}$ such that not all of them are 0 and the following limit holds:

$$d_n \cdot \frac{p_{\lambda_n G_n}(x) - p_{\lambda^* G_*}(x)}{\overline{W}_1(\lambda_n G_n, \lambda^* G_*)} \to \alpha_0 h_0(x) + \sum_{i=1}^{k_*} \alpha_i f(x|\theta_i^*) + \beta_i^\top \frac{\partial f}{\partial \theta}(x|\theta_i^*).$$

By means of Fatou's lemma, we have

$$0 = \lim_{n \to \infty} d_n \cdot \frac{V(p_{\lambda_n G_n}, p_{\lambda^* G_*})}{\overline{W}_1(\lambda_n G_n, \lambda^* G_*)} \geq \int \liminf_{n \to \infty} d_n \cdot \frac{p_{\lambda_n G_n}(x) - p_{\lambda^* G_*}(x)}{\overline{W}_1(\lambda_n G_n, \lambda^* G_*)} dx,$$

$$= \int \left( \alpha_0 h_0(x) + \sum_{i=1}^{k_*} \alpha_i f(x|\theta_i^*) + \beta_i^\top \frac{\partial f}{\partial \theta}(x|\theta_i^*) \right) dx. \quad (47)$$

The above equation indicates that

$$\alpha_0 h_0(x) + \sum_{i=1}^{k_*} \alpha_i f(x|\theta_i^*) + \beta_i^\top \frac{\partial f}{\partial \theta}(x|\theta_i^*) = 0,$$

for almost surely $x$. Since $(f, k_*)$ is distinguishable from $h_0$ and $f$ is first order identifiable, the above equation suggests that $\alpha_0 = \alpha_1 = \ldots = \alpha_{k_*} = 0$ and $\beta_1 = \ldots = \beta_{k_*} = \mathbf{0}$, which is a contradiction.

As a consequence, we achieve the conclusion of claim (42).

**Proof of claim** (43) Similar to the proof of claim (42), we also prove claim (43) by contradiction. Assume that claim (43) does not hold. It implies that we can find sequences $\lambda_n' \in [0, 1]$ and $G_n' \in \mathcal{E}_{k_*}(\Theta)$ such that $\overline{W}_1(\lambda_n' G_n', \lambda^* G_*) > \epsilon'$ and $V(p_{\lambda_n' G_n'}, p_{\lambda^* G_*})/\overline{W}_1(\lambda_n' G_n', \lambda^* G_*) \to 0$ as $n \to \infty$. Since $[0, 1]$ and $\Theta$ are bounded sets, there exist $\lambda' \in [0, 1]$ and $G' \in \mathcal{E}_{k_*}(\Theta)$ such that $\lambda_n' \to \lambda'$ and $W_1(G_n', G') \to 0$ as $n \to \infty$. Since $\overline{W}_1(\lambda_n' G_n', \lambda^* G_*) > \epsilon'$ for all $n$, the previous limits indicate that $\overline{W}_1(\lambda' G', \lambda^* G_*) \geq \epsilon'$.

On the other hand, since $V(p_{\lambda_n' G_n'}, p_{\lambda^* G_*})/\overline{W}_1(\lambda_n' G_n', \lambda^* G_*) \to 0$, we have $V(p_{\lambda_n' G_n'}, p_{\lambda^* G_*}) \to 0$ as $n \to \infty$. An application of Fatou's lemma leads to

$$0 = \lim_{n \to \infty} V(p_{\lambda_n' G_n'}, p_{\lambda^* G_*}) \geq \frac{1}{2} \int \liminf_{n \to \infty} \left| p_{\lambda_n' G_n'}(x) - p_{\lambda^* G_*}(x) \right| dx = V(p_{\lambda' G', \lambda^* G_*}).$$

Due to the identifiability of model (1), the above equation leads to $(\lambda', G') \equiv (\lambda^*, G_*)$, which is a contradiction to the condition that $\overline{W}_1(\lambda' G', \lambda^* G_*) \geq \epsilon'$. As a consequence, we achieve the conclusion of claim (43).

### D.2.2 Proof of setting $\lambda^* = 0$

We want to show that

$$\inf_{G \in \mathcal{E}_{k_*}(\Theta)} \frac{V(p_{\lambda G}, p_{\lambda^* G_*})}{\lambda} > 0 \tag{48}$$

**Proof of claim** (48): Assume that claim (48) does not hold. We can find two sequences $\bar{\lambda}_n \in [0,1]$ and $\bar{G}_n \in \mathcal{E}_{k_*}(\Theta)$ such that $V(p_{\bar{\lambda}_n \bar{G}_n}, p_{\lambda^* G_*})/\bar{\lambda}_n \to 0$ as $n \to \infty$. We denote $\bar{G}_n = \sum_{i=1}^{k_*} \bar{p}_i^n \delta_{\bar{\theta}_i^n}$. Since $\Theta$ is a bounded set, there exists $\bar{G} = \sum_{i=1}^{k_*} \bar{p}_i \delta_{\bar{\theta}_i} \in \mathcal{E}_{k_*}(\Theta)$ such that $W_1(\bar{G}_n, \bar{G}) \to 0$ as $n \to \infty$. Invoking Fatou's lemma, we obtain that

$$0 = \lim_{n \to \infty} \frac{V(p_{\bar{\lambda}_n \bar{G}_n}, p_{\lambda^* G_*})}{\lambda_n} \geq \frac{1}{2} \int \liminf_{n \to \infty} \left| \sum_{i=1}^{k_*} \bar{p}_i^n f(x|\bar{\theta}_i^n) - h_0(x) \right| dx$$

$$= V\left( \sum_{i=1}^{k_*} \bar{p}_i f(.|\bar{\theta}_i), h_0(.) \right).$$

The above equation shows that $\sum_{i=1}^{k_*} \bar{p}_i f(x|\bar{\theta}_i) = h_0(x)$ for almost surely $x$, which is a contradiction to the hypothesis that $(f, k_*)$ is distinguishable from $h_0$. Hence, we reach the conclusion of claim (48).

### D.3 Proof of Theorem 3.4

*Theorem* 3.4. Assume that $k_*$ is unknown and strictly upper bounded by a given $K$. Besides that, $f$ is second order identifiable and $(f, K)$ is distinguishable from $h_0$. Then, for any $G \in \mathcal{O}_K(\Theta)$, there exist positive constant $C_1$ and $C_2$ depending only on $\lambda^*, G_*, h_0, \Theta$ such that the following holds:

(a) When $\lambda^* = 0$, then $V(p_{\lambda^* G_*}, p_{\lambda G}) \geq C_1 \lambda$.

(b) When $\lambda^* \in (0, 1]$, then

$$V(p_{\lambda^* G_*}, p_{\lambda G}) \geq C_2 \underbrace{\left[ |\lambda - \lambda^*| + (\lambda + \lambda^*) W_2^2(G, G_*) \right]}_{\overline{W}_2(\lambda G, \lambda^* G_*)}.$$

The proof argument for the setting $\lambda^* = 0$ is similar to that in Section D.2.2; therefore, it is omitted. We focus only on the proof of the setting $\lambda^* \in (0, 1]$.

Similar to the proof of Theorem 3.3, in order to reach the conclusion of Theorem 3.4 for the setting $\lambda^* \in (0, 1]$, it is sufficient to demonstrate the following claims:

$$\lim_{\epsilon \to 0} \inf_{\lambda \in [0,1], G \in \mathcal{O}_K(\Theta)} \left\{ \frac{V(p_{\lambda G}, p_{\lambda^* G_*})}{\overline{W}_2(\lambda G, \lambda^* G_*)} : \overline{W}_2(\lambda G, \lambda^* G_*) \leq \epsilon \right\} > 0, \tag{49}$$

$$\inf_{\lambda \in [0,1], G \in \mathcal{O}_K(\Theta): \overline{W}_2(\lambda G, \lambda^* G_*) > \epsilon'} \frac{V(p_{\lambda G}, p_{\lambda^* G_*})}{\overline{W}_2(\lambda G, \lambda^* G_*)} > 0,$$

for any $\epsilon' > 0$. Since the proof of the second claim is similar to that of claim (43) in Section D.2; therefore, it is omitted.

**Proof of claim** (49): Similar to the proof of claim (42), we use proof by contradiction for claim (49). Assume that claim (49) does not hold. Given that assumption, we can find sequences $G_n \in \mathcal{O}_K(\Theta)$ and $\lambda_n \in [0, 1]$ such that $\overline{W}_2(\lambda_n G_n, \lambda^* G_*) \to 0$ and $V(p_{\lambda_n G_n}, p_{\lambda^* G_*})/\overline{W}_2(\lambda_n G_n, \lambda^* G_*) \to 0$ as $n \to \infty$. As $W_2(G_n, G_*) \to 0$ as $n \to \infty$, using the similar argument as that in Section 3.2 in Ho et al. [18], we can find a subsequence of $G_n$ (without loss of generality, we replace that subsequence by the whole sequence of $G_n$ with $k' \in [k_*, K]$ supports such that

$$G_n = \sum_{i=1}^{k_* + \bar{l}} \sum_{j=1}^{s_i} p_{ij}^n \delta_{\theta_{ij}^n}, \tag{50}$$

where $\sum_{j=1}^{s_i} p_{ij}^n \to p_i^*$ and $\theta_{ij}^n \to \theta_i^*$ for all $i \in [k_* + \bar{l}]$. Here, $p_i^* = 0$ for $k_* + 1 \le i \le k_* + \bar{l}$. In addition, $s_1, \dots, s_{k_*+\bar{l}} \ge 1$ are such that $\sum_{i=1}^{k_*+\bar{l}} s_i = k'$. To ease the ensuing presentation, we denote $\Delta\theta_{ij}^n := \theta_{ij}^n - \theta_i^*$ and $\Delta p_{i\cdot}^n := \sum_{j=1}^{s_i} p_{ij}^n - p_i^*$ for $i \in [k_* + \bar{l}]$. Then, based on Lemma 3.1 in Ho et al. [18], we have

$$W_2^2(G_n, G_*) \asymp \sum_{i=1}^{k_*+\bar{l}} |\Delta p_{i\cdot}^n| + \sum_{i=1}^{k_*+\bar{l}} \sum_{j=1}^{s_i} p_{ij}^n \left\| \Delta\theta_{ij}^n \right\|_2^2. \tag{51}$$

We divide our proof of claim (49) into three steps.

**Step 1 - Taylor expansion:** An application of Taylor expansion up to the second order leads to

$$f(x|\theta_{ij}^n) = f(x|\theta_i^*) + (\Delta\theta_{ij})^\top \frac{\partial f}{\partial \theta}(x|\theta_i^*) + (\Delta\theta_{ij})^\top \frac{\partial^2 f}{\partial \theta^2}(x|\theta_i^*)(\Delta\theta_{ij}) + R_{ij}(x),$$

where $R_{ij}(x)$ is Taylor remainder such that $R_{ij}(x) = o(\|\Delta\theta_{ij}\|_2^2)$ for all $i \in [k_* + \bar{l}]$ and $j \in [s_i]$. Collecting the above equations, we obtain that

$$p_{\lambda_n G_n}(x) - p_{\lambda^* G_*}(x) = (\lambda^* - \lambda_n)h_0(x) + \sum_{i=1}^{k_*+\bar{l}} \left( \sum_{j=1}^{s_i} \lambda_n p_{ij}^n - \lambda^* p_i^* \right) f(x|\theta_i^*)$$

$$+ \lambda_n \left( \sum_{j=1}^{s_i} p_{ij}^n \Delta\theta_{ij}^n \right)^\top \frac{\partial f}{\partial \theta}(x|\theta_i^*) + \lambda_n \left( \sum_{j=1}^{s_i} p_{ij}^n \left( \Delta\theta_{ij}^n \right)^\top \frac{\partial^2 f}{\partial \theta^2}(x|\theta_i^*)(\Delta\theta_{ij}^n) \right) + R(x),$$

$$\tag{52}$$

where $R(x) = \lambda_n \sum_{i=1}^{k_*+\bar{l}} \sum_{j=1}^{s_i} p_{ij}^n R_{ij}(x) = o\left( \lambda_n \sum_{i=1}^{k_*+\bar{l}} \sum_{j=1}^{s_i} p_{ij}^n \left\| \Delta\theta_{ij}^n \right\|_2^2 \right)$. Given the expression of $W_2^2(G_n, G_*)$ in equation (77), we can verify that $R(x)/\overline{W}_2(\lambda_n G_n, \lambda^* G_*) \to 0$ as $n \to \infty$.

**Step 2 - Non-vanishing coefficients:** Given the expression in equation (52), we can view $(p_{\lambda_n G_n}(x) - p_{\lambda^* G_*}(x))/\overline{W}_2(\lambda_n G_n, \lambda^* G_*)$ as a linear combination of elements of the forms $h_0(x), f(x|\theta_i^*), \frac{\partial f}{\partial \theta}(x|\theta_i^*)$, and $\frac{\partial^2 f}{\partial \theta^2}(x|\theta_i^*)$ for all $i \in [k_* + \bar{l}]$. Assume that their coefficients go to 0 as $n$ tends to infinity. As the coefficient of $h_0(x)$ goes to 0, we have $(\lambda_n - \lambda^*)/\overline{W}_2(\lambda_n G_n, \lambda^* G_*) \to 0$.

Similarly, by learning the coefficients of $f(x|\theta_i^*)$ and $\left[ \frac{\partial^2 f}{\partial \theta^2}(x|\theta_i^*) \right]_{jj}$ for $j \in [d]$, we obtain the following limits:

$$\left( \sum_{j=1}^{s_i} \lambda_n p_{ij}^n - \lambda^* p_i^* \right) / \overline{W}_2(\lambda_n G_n, \lambda^* G_*) \to 0, \quad \lambda_n \left( \sum_{j=1}^{s_i} p_{ij}^n \left\| \Delta\theta_{ij}^n \right\|_2^2 \right) / \overline{W}_2(\lambda_n G_n, \lambda^* G_*) \to 0.$$

Collecting the above limits, we find that

$$\frac{\lambda^* \Delta p_{i\cdot}^n}{\overline{W}_2(\lambda_n G_n, \lambda^* G_*)} = \frac{(\lambda^* - \lambda_n)\left( \sum_{j=1}^{s_i} p_{ij}^n \right) + \left( \sum_{j=1}^{s_i} \lambda_n p_{ij}^n - \lambda^* p_i^* \right)}{\overline{W}_2(\lambda_n G_n, \lambda^* G_*)} \to 0.$$

Putting the above results together, we achieve that $1 = \overline{W}_2(\lambda_n G_n, \lambda^* G_*)/\overline{W}_2(\lambda_n G_n, \lambda^* G_*) \to 0$, which is a contraction. Therefore, not all the coefficients associated with $h_0(x), f(x|\theta_i^*), \frac{\partial f}{\partial \theta}(x|\theta_i^*)$, and $\frac{\partial^2 f}{\partial \theta^2}(x|\theta_i^*)$ for $i \in [k_* + \bar{l}]$ go to 0 as $n$ tends to infinity.

**Step 3: Show the contradiction using the distinguishability condition and Fatou's lemma:** Similar to Step 3 in Section D.2.1, by denoting $d_n = 1/m_n$ where $m_n$ is the maximum values of the absolute values of the coefficients of $h_0(x), f(x|\theta_i^*), \frac{\partial f}{\partial \theta}(x|\theta_i^*)$, and $\frac{\partial^2 f}{\partial \theta^2}(x|\theta_i^*)$, we have

$$d_n \cdot \frac{p_{\lambda_n G_n}(x) - p_{\lambda^* G_*}(x)}{\overline{W}_1(\lambda_n G_n, \lambda^* G_*)} \to \alpha_0 h_0(x) + \sum_{i=1}^{k_*+\bar{l}} \alpha_i f(x|\theta_i^*) + \beta_i^\top \frac{\partial f}{\partial \theta}(x|\theta_i^*) + \gamma_i^\top \frac{\partial^2 f}{\partial \theta^2}(x|\theta_i^*)\gamma_i,$$

where $\alpha_i, \beta_i, \gamma_i$ are some coefficients such that not all of them are 0. However, the Fatou's lemma suggests that the RHS of the above equation is 0 for almost surely $x$. Since $(f, K)$ is distinguishable from $h_0$, it shows that $\alpha_i = 0$, $\beta_i = \mathbf{0} \in \mathbb{R}^d$, and $\gamma_i = \mathbf{0} \in \mathbb{R}^{d \times d}$ for all $i \in [k_* + \bar{l}]$— a contradiction. As a consequence, we obtain the conclusion of claim (49).

### D.4  Proof of Theorem 3.5

*Theorem* 3.5. Assume that $k_*$ is unknown and strictly upper bounded by a given $K$. Besides that, $f$ is location-scale Gaussian distribution and $(f, K)$ with fixed variance is distinguishable in any order from $h_0$. Then, for any $G \in \mathcal{O}_K(\Theta)$, there exist positive constant $C_1$ and $C_2$ depending only on $\lambda^*, G_*, h_0, \Theta$ such that the following holds:

(a) When $\lambda^* = 0$, then $V(p_{\lambda^* G_*}, p_{\lambda G}) \geq C_1 \lambda$.

(b) When $\lambda^* \in (0, 1]$, then

$$V(p_{\lambda^* G_*}, p_{\lambda G}) \geq C_2 \overline{W}_{\overline{r}(K - k_*)}(\lambda G, \lambda^* G_*).$$

The proof argument for the setting $\lambda^* = 0$ is similar to that in Section D.2.2; therefore, it is omitted. We focus only on the proof of the setting $\lambda^* \in (0, 1]$.

Denote by $\overline{r}_1 = \overline{r}(K - k_*)$. Similar to the proof of Theorem 3.3, in order to reach the conclusion of Theorem 3.5 for the setting $\lambda^* \in (0, 1]$, it is sufficient to demonstrate the following claims:

$$\lim_{\epsilon \to 0} \inf_{\lambda \in [0,1], G \in \mathcal{O}_K(\Theta)} \left\{ \frac{V(p_{\lambda G}, p_{\lambda^* G_*})}{\overline{W}_{\overline{r}_1}(\lambda G, \lambda^* G_*)} : \overline{W}_{\overline{r}_1}(\lambda G, \lambda^* G_*) \leq \epsilon \right\} > 0, \tag{53}$$

$$\inf_{\lambda \in [0,1], G \in \mathcal{O}_K(\Theta): \overline{W}_{\overline{r}_1}(\lambda G, \lambda^* G_*) > \epsilon'} \frac{V(p_{\lambda G}, p_{\lambda^* G_*})}{\overline{W}_{\overline{r}_1}(\lambda G, \lambda^* G_*)} > 0,$$

for any $\epsilon' > 0$. Since the proof of the second claim is similar to that of claim (43) in Section D.2; therefore, it is omitted. We now proceed to prove claim (53). Suppose that it is not correct, that is, there exist sequences $\lambda_n$ and $G_n = \sum_{i=1}^{k_n} p_i^n \delta_{\theta_i^n} \in \mathcal{O}_K(\Theta)$ such that $\overline{W}_{\overline{r}_1}(\lambda_n G_n, \lambda^* G_*) \to 0$ and $V(p_{\lambda_n G_n}, p_{\lambda^* G_*}) / \overline{W}_{\overline{r}_1}(\lambda_n G_n, \lambda^* G_*) \to 0$. For the ease of presentation, we consider the one dimension Gaussian case where $(\mu, \Sigma) = (\theta, v)$, the higher dimension cases are treated similar.

We can use the subsequence argument to have $\lambda^* \geq \lambda_n$ for all $n$ and $G_n$ can be assumed to have a fixed number of atoms $k'$ (less than or equals $K$) and have a representation as in (54), that is,

$$G_n = \sum_{i=1}^{k_* + \bar{l}} \sum_{j=1}^{s_i} p_{ij}^n \delta_{(\theta_{ij}^n, v_{ij}^n)}, \tag{54}$$

where $\sum_{j=1}^{s_i} p_{ij}^n \to p_i^*$ and $\theta_{ij}^n \to \theta_i^*, v_{ij}^n \to v_i^*$ for all $i \in [k_* + \bar{l}]$. Here, $p_i^* = 0$ for $k_* + 1 \leq i \leq k_* + \bar{l}$. In addition, $s_1, \ldots, s_{k_* + \bar{l}} \geq 1$ are such that $\sum_{i=1}^{k_* + \bar{l}} s_i = k'$.

**Step 1 - Taylor expansion:**  Using Taylor expansion of $f$ around $\{(\theta_i^*, v_i^*)\}_{i=1}^{k_*}$ to the $\overline{r}_1$−th order we have

$$p_{\lambda_n G_n}(x) - p_{\lambda^* G_*}(x) = (\lambda^* - \lambda_n) h_0(x) + \lambda_n \left( \sum_{i=1}^{k_* + \underline{l}} \sum_{j=1}^{s_i} p_{ij}^n f(x | \theta_{ij}^n, v_{ij}^n) \right) - \sum_{i=1}^{k_*} p_i^* f(x | \theta_i^*, v_i^*)$$

$$= (\lambda^* - \lambda_n) h_0(x) + \sum_{i=1}^{k_* + \underline{l}} \sum_{j=1}^{s_i} \lambda_n p_{ij}^n \sum_{|\boldsymbol{\alpha}|=1}^{\overline{r}_1} (\Delta \theta_{ij}^n)^{\alpha_1} (\Delta v_{ij}^n)^{\alpha_2} \frac{1}{\boldsymbol{\alpha}!} \frac{\partial^{|\boldsymbol{\alpha}|} f(\theta_i^*, v_i^*)}{\partial^{\alpha_1} \theta \partial^{\alpha_2} v}$$

$$+ \sum_{i=1}^{k_* + \underline{l}} (\Delta \overline{p}_{i\cdot}^n) f(x | \theta_i^*, v_i^*) + R(x),$$

where $\boldsymbol{\alpha} = (\alpha_1, \alpha_2), |\boldsymbol{\alpha}| = \alpha_1 + \alpha_2, \boldsymbol{\alpha}! = \alpha_1! \alpha_2!, \Delta \overline{p}_{i\cdot}^n = \lambda_n \sum_j p_{ij}^n - p_i^*, \Delta \theta_{ij}^n = \theta_{ij}^n - \theta_i^*, \Delta v_{ij}^n = v_{ij}^n - v_i^*$ and $R(x) = o(\sum_{i=1}^{k_* + \bar{l}} \sum_{j=1}^{s_i} p_{ij}^n (|\Delta \theta_{ij}^n|^{\overline{r}_1} + |\Delta v_{ij}^n|^{\overline{r}_1}))$. Now we can use the character

equation $\dfrac{\partial^2 f}{\partial \theta^2} = 2\dfrac{\partial f}{\partial v}$ to rewrite the formula above as

$$(\lambda^* - \lambda_n)h_0(x) + \sum_{\alpha=1}^{2\bar{r}_1} \sum_{i=1}^{k_*+\underline{l}} \left( \sum_{j=1}^{s_i} \lambda_n p_{ij}^n \sum_{n_1,n_2} \frac{(\Delta\theta_{ij}^n)^{n_1}(\Delta v_{ij}^n)^{n_2}}{2^{n_2} n_1! n_2!} \right) \frac{\partial^\alpha f(\theta_i^*, v_i^*)}{\partial \theta^\alpha}$$

$$+ \sum_{i=1}^{k_*+\underline{l}} (\Delta p_{i\cdot}^n) f(x|\theta_i^*, v_i^*) + R(x), \qquad (55)$$

where we sum over $n_1, n_2$ such that $n_1 + 2n_2 = \alpha, n_1 + n_2 \leq \bar{r}_1$.

**Step 2 - Non-vanishing coefficients:** Assume that all coefficients in the formula above vanish when dividing by $W_{\bar{r}_1}^{\bar{r}_1}(\lambda_n G_n, \lambda^* G_*)$ when $n \to \infty$. Because

$$W_{\bar{r}_1}^{\bar{r}_1}(\lambda_n G_n, \lambda^* G_*) \asymp |\lambda_n - \lambda^*| + (\lambda_n + \lambda^*)\left( \sum_{i=1}^{k_*+\bar{l}} |\Delta p_{i\cdot}^n| + \sum_{i=1}^{k_*+\bar{l}} \sum_{j=1}^{s_i} p_{ij}^n(\|\Delta\theta_{ij}^n\|_2^{\bar{r}_1} + \|\Delta v_{ij}^n\|_2^{\bar{r}_1}) \right) := D_{\bar{r}_1}(G_n, G_*),$$

$$(56)$$

we have

$$\frac{\lambda^* - \lambda_n}{D_{\bar{r}_1}(G_n, G_*)} \to 0, \quad \frac{\Delta p_{i\cdot}^n}{D_{\bar{r}_1}(G_n, G_*)} \to 0. \qquad (57)$$

These limits together imply

$$\frac{(\lambda^* + \lambda_n)\Delta p_{i\cdot}^n}{D_{\bar{r}_1}(G_n, G_*)} \to 0, \quad \forall i = 1, \ldots, k_* + \bar{l}.$$

From the definition of $D_{\bar{r}_1}$, it can be deduced that there exists at least an index $i^*$ such that

$$\sum_{j=1}^{s_{i*}} \frac{(\lambda_n + \lambda^*)p_{i^*j}^n((\theta_{ij}^n)^{\bar{r}_1} + (v_{ij}^n)^{\bar{r}_1})}{D_{\bar{r}_1}(G_n, G_*)} \nrightarrow 0.$$

Without loss of generality, assign $i^* = 1$. But as we assume all the coefficients in equation (55) go to 0 for all $\alpha$ and $i$, we have

$$\frac{\sum_{j=1}^{s_1} \lambda_n p_{1j}^n \sum_{\substack{n_1+2n_2=\alpha \\ n_1+n_2 \leq \bar{r}_1}} \dfrac{(\theta_{1j}^n)^{n_1}(v_{1j}^n)^{n_2}}{2^{n_2} n_1! n_2!}}{D_{\bar{r}_1}(G_n, G_*)} \to 0,$$

for all $\alpha = 1, \ldots, 2\bar{r}_1$. From two expressions above combining with equation (57), we have for all $\alpha = 1, \ldots, 2\bar{r}_1$,

$$F_\alpha := \frac{\sum_{j=1}^{s_1} p_{1j}^n \sum_{\substack{n_1+2n_2=\alpha \\ n_1+n_2 \leq \bar{r}_1}} \dfrac{(\Delta\theta_{1j}^n)^{n_1}(\Delta v_{1j}^n)^{n_2}}{2^{n_2} n_1! n_2!}}{\sum_{j=1}^{s_1} p_{1j}^n((\Delta\theta_{ij}^n)^{\bar{r}_1} + (\Delta v_{ij}^n)^{\bar{r}_1})} \to 0. \qquad (58)$$

If $s_1 = 1$ then substituting $\alpha = 1$ and $\alpha = 2\bar{r}_1$ gives

$$\frac{|\Delta\theta_{11}^n|^{\bar{r}_1}}{|\Delta\theta_{11}^n|^{\bar{r}_1} + |\Delta v_{11}^n|^{\bar{r}_1}}, \quad \frac{|\Delta v_{11}^n|^{\bar{r}_1}}{|\Delta\theta_{11}^n|^{\bar{r}_1} + |\Delta v_{11}^n|^{\bar{r}_1}} \to 0,$$

which is impossible as they are sum up to 1 for all $n$. Hence $s_1 \geq 2$. Now we proceed to show the contradiction using the system of equations (6). Denote by $\bar{p}_n = \max_{1 \leq j \leq s_1}\{p_{1j}^n\}, \overline{M}_n = \max_{1 \leq j \leq s_1}\{|\Delta\theta_{1j}^n|, |\Delta v_{1j}^n|^{1/2}\}$. By the subsequence argument in compact sets, without loss of generality, we can denote $c_j^2 := \lim_{n\to\infty} p_{1j}^n/\bar{p}_n$, $a_j = \lim \Delta\theta_{1j}^n/\overline{M}_n$, and $b_j = \lim \Delta v_{1j}^n/\overline{M}_n$ for all $j = 1, \ldots, k_* + \bar{l}$. Because of the definition of $\mathcal{O}_{K,c_0}$, we have $p_{ij}^n \geq c_0$ for all $j$, which implies all $c_j$ are different from 0 and at least one of them is 1. Similarly, in $(a_j, b_j)_j$, there is at least one of them equals to 1 or $-1$. Dividing both numerators and denominators of equation (58) by $\bar{p}_n \overline{M}_n^\alpha$, we have

$$\sum_{j=1}^{s_1} \sum_{n_1+2n_2=\alpha} \frac{c_j^2 a_j^{n_1} b_j^{n_2}}{n_1! n_2!} = 0,$$

for all $\alpha = 1, \ldots, \bar{r}_1$. Hence, we get the contradiction, where we use the fact that $s_1 \leq K - k_* + 1$ (as $s_i \geq 1$ for all $i \geq 2$) and $\bar{r}_1 = \bar{r}(K - k_*)$ is the smallest number such that equation (6), where $k = K - k_*$, has the trivial solution only. Hence, when dividing by $W_{\bar{r}_1}^{\bar{r}_1}(\lambda_n G_n, \lambda^* G_*)$, not all coefficients of equation (55) vanish as $n \to \infty$.

**Step 3: Show the contradiction using the distinguishability condition and Fatou's lemma:** Denote by

$$E_{i,\alpha} = \sum_{j=1}^{s_i} \lambda_n p_{ij}^n \sum_{n_1, n_2} \frac{(\Delta \theta_{ij}^n)^{n_1} (\Delta v_{ij}^n)^{n_2}}{2^{n_2} n_1! n_2!} \Big/ W_{\bar{r}_1}^{\bar{r}_1}(\lambda_n G_n, \lambda^* G_*), \quad \forall i, \alpha \geq 1.$$

$$E_{i,0} = \Delta p_{i\cdot}^n \Big/ W_{\bar{r}_1}^{\bar{r}_1}(\lambda_n G_n, \lambda^* G_*), \forall i \geq 1, E_{0,0} = (\lambda^* - \lambda_n) \Big/ W_{\bar{r}_1}^{\bar{r}_1}(\lambda_n G_n, \lambda^* G_*).$$

We have proved that not all $E_{i,\alpha}$ go to 0. Let $d_n = \max_{0 \leq \alpha \leq 2\bar{r}_1, 0 \leq i \leq k'} |E_{i,\alpha}|$. Because $E_{i,\alpha}/d_n \in [-1, 1]$ for all $n$, by the subsequence argument if needed, we have $E_{i,\alpha}/m_n \to \beta_{i,\alpha}$ as $n \to \infty$, where at least one of the limits are different from 0. But Fatou's argument implies that

$$\beta_{0,0} h_0(x) + \sum_{i=1}^{k_*} \sum_{\alpha=0}^{2\bar{r}_1} \beta_{i,\alpha} \frac{\partial^\alpha f}{\partial \theta^\alpha}(x | \theta_i^*, v_i^*) = 0,$$

which contradicts our assumption. Hence, claim (53) is proved.

## D.5 Proof Theorem A.1

*Theorem* A.1. Assume that $h_0$ takes the form (7) and $\lambda^* = 0$. Then, there exist positive constants $C_1$ and $C_2$ depending only on $h_0, \Theta$ such that the following holds:

(a) (exact-fitted) If $f$ is first order identifiable, then for any $G \in \mathcal{E}_{k_0}(\Theta)$
$$V(p_{\lambda^*, G_*}, p_{\lambda, G}) \geq C_1 \lambda W_1(G, G_0),$$

(b) (over-fitted) If $f$ is second order identifiable, then for any $G \in \mathcal{O}_K(\Theta)$ that $K > k_0$
$$V(p_{\lambda^*, G_*}, p_{\lambda, G}) \geq C_2 \lambda W_2^2(G, G_0),$$

(c) (over-fitted and weakly identifiable) If $f$ is location-scale Gaussian distribution and we further assume that $G_* \in \mathcal{E}_{k_*, c_0}(\Theta)$, then for any $G \in \mathcal{O}_{K, c_0}(\Theta)$ that $K > k_0$, there exists $C_3$ depends on $h_0, \Theta_0, c_0$ such that
$$V(p_{\lambda^*, G_*}, p_{\lambda, G}) \geq C_3 \lambda W_{\bar{r}(K-k_*)}^{\bar{r}(K-k_*)}(G, G_0)$$

(a) We can write
$$\frac{V(p_0, p_{\lambda G})}{\lambda W_1(G, G_0)} = \int \frac{|\sum_{i=1}^{k_0} p_i^0 f(x|\theta_i^0) - \sum_{i=1}^{k_0} p_i f(x|\theta_i)|}{W_1(G, G_0)} dx$$
$$= \frac{V(p_0, p_G)}{W_1(G, G_0)},$$
because this is the exact-fitted and first-order identifiable, we can apply Theorem 3.1. in Ho et al. [17]

(b) Similar to the last part, we can write
$$\frac{V(p_0, p_{\lambda G})}{\lambda W_2^2(G, G_0)} = \int \frac{|\sum_{i=1}^{k_0} p_i^0 f(x|\theta_i^0) - \sum_{i=1}^{K} p_i f(x|\theta_i)|}{W_2^2(G, G_0)} dx$$
$$= \frac{V(p_0, p_G)}{W_2^2(G, G_0)},$$
as this is the over-fitted and second-order identifiable, we can apply Theorem 3.2. in Ho et al. [17].

(c) Similar to last two cases, we can write
$$\frac{V(p_0, p_{\lambda G})}{\lambda W_{\bar{r}(K-k_*)}^{\bar{r}(K-k_*)}(G, G_0)} = \frac{V(p_0, p_G)}{W_{\bar{r}(K-k_*)}^{\bar{r}(K-k_*)}(G, G_0)},$$
and apply Proposition 2.2. in [16].

## D.6 Proof of Theorem 3.6

*Theorem* 3.6. Assume that $h_0$ takes the form (7). Besides that, $K \geq k_0$ and $f$ is location-scale Gaussian distribution. Then, for any $\lambda \in [0,1]$ and $G \in \mathcal{O}_{K,c_0}(\Theta)$ for some $c_0 > 0$, there exist positive constants $C_1, C_2, C_3, C_4$ depending only on $\lambda^*, G_*, G_0, \Theta$ ($C_3$ and $C_4$ also depends on $\delta$) such that the following holds:

(a) When $K \leq k_* + k_0 - \bar{k} - 1$, then

$$V(p_{\lambda^*,G_*}, p_{\lambda,G}) \geq C_1 \overline{W}_{\bar{r}(K-k_*)}(\lambda G, \lambda^* G_*).$$

(b) When $K \geq k_* + k_0 - \bar{k}$, then

$$V(p_{\lambda^*,G_*}, p_{\lambda,G}) \geq C_2 \left( 1_{\{\lambda \leq \lambda^*\}} \overline{W}_{\bar{r}(K-k_*)}(\lambda G, \lambda^* G_*) \right.$$

$$\left. + 1_{\{\lambda > \lambda^*\}} W_{\bar{r}(K-k_*)}^{\bar{r}(K-k_*)}(G, \overline{G}_*(\lambda)) \right)$$

(c) For $\delta > 0$, when $K = k_* + k_0 - \bar{k}$, we have

$$V(p_{\lambda^*,G_*}, p_{\lambda,G}) \geq C_3 1_{\{\lambda > \lambda^* + \delta\}} W_1(G, \overline{G}_*(\lambda)),$$

and when $K > k_* + k_0 - \bar{k}$, we have

$$V(p_{\lambda^*,G_*}, p_{\lambda,G}) \geq C_4 1_{\{\lambda > \lambda^* + \delta\}}$$

$$\times W_{\bar{r}(K-k_0-k_*+\bar{k})}^{\bar{r}(K-k_0-k_*+\bar{k})}(G, \overline{G}_*(\lambda)).$$

To facilitate the proof argument, we denote $\mathcal{T} := k_* + k_0 - \bar{k}$. In addition, we assume without loss of generality that $\theta_i^* = \theta_i^0$ for $i \in [\bar{k}]$. Moreover, we introduce the following shorthand:

$$D(\lambda G, \lambda^* G_*) = \begin{cases} \overline{W}_2(\lambda G, \lambda^* G_*), & \text{when } K \leq \mathcal{T} - 1 \\ 1_{\{\lambda \leq \lambda^*\}} \overline{W}_2(\lambda G, \lambda^* G_*) + 1_{\{\lambda > \lambda^*\}} (\lambda + \lambda^*) W_2^2(G, \overline{G}_*(\lambda)), & \text{when } K \geq \mathcal{T} \end{cases}.$$

Similar to the previous proofs, in order to obtain the conclusion of the theorem, we need to prove the following claims:

$$\lim_{\epsilon \to 0} \inf_{\lambda \in [0,1], G \in \mathcal{O}_K(\Theta)} \left\{ \frac{V(p_{\lambda G}, p_{\lambda^* G_*})}{D(\lambda G, \lambda^* G_*)} : D(\lambda G, \lambda^* G_*) \leq \epsilon \right\} > 0. \tag{59}$$

**Proof of claim** (59): Assume that the above claim is not true. It indicates that we can find sequences $G_n = \sum_{i=1}^{k_n} p_i^n \delta_{\theta_i^n} \in \mathcal{O}_K(\Theta)$ and $\lambda_n \in [0,1]$ such that $D(\lambda_n G_n, \lambda^* G_*)$ and $V(p_{\lambda_n G_n}, p_{\lambda^* G_*})/D(\lambda_n G_n, \lambda^* G_*)$ go to 0 as $n$ approaches to infinity. Given the assumption that $\theta_i^* = \theta_i^0$ for $i \in [\bar{k}]$, we obtain that

$$p_{\lambda_n G_n}(x) - p_{\lambda^* G_*}(x) = (\lambda^* - \lambda_n) \sum_{i=\bar{k}+1}^{k_0} p_i^0 f(x|\theta_i^0) + \lambda_n \left( \sum_{i=1}^{k_n} p_i^n f(x|\theta_i^n) \right) - \sum_{i=1}^{k_*} \bar{p}_i^* f(x|\theta_i^*),$$

$$\tag{60}$$

where $\bar{p}_i^* = \lambda^* p_i^* + (\lambda_n - \lambda^*) p_i^0$ when $1 \leq i \leq \bar{k}$ and $\bar{p}_i^* = \lambda^* p_i^*$ otherwise. Now, we prove the contradiction of our assumption under two separate settings of $\lambda_n$.

**Case 1:** $\lambda^* \geq \lambda_n$ for infinitely many $n$. Without loss of generality, we assume that $\lambda^* \geq \lambda_n$ for all $n \geq 1$. Under this case, $D(\lambda_n G_n, \lambda^* G_*) = \overline{W}_2(\lambda_n G_n, \lambda^* G_*)$. As $D(\lambda_n G_n, \lambda^* G_*) \to 0$, we have $\lambda_n \to \lambda^*$ and $W_2(G_n, G_*) \to 0$ as $n \to \infty$. Therefore, we can rewrite $G_n$ like equation (54).

In light of equation (60) and the assumption $\lambda^* \geq \lambda_n$, by means of Taylor expansion up to the second order around $\theta_1^*, \ldots, \theta_{k_*}^*$ as that in the proof of Theorem D.3, we can view $(p_{\lambda_n G_n}(x) - p_{\lambda^* G_*}(x))/D(\lambda_n G_n, \lambda^* G_*)$ as a linear combination of elements of the forms $f(x|\theta_i^0)$, $f(x|\theta_j^*)$, $\frac{\partial f}{\partial \theta}(x|\theta_j^*)$, and $\frac{\partial^2 f}{\partial \theta^2}(x|\theta_j^*)$ for $\bar{k}+1 \leq i \leq k_0$ and $j \in [k_*]$.

It is sufficient to argue that not all the coefficients of these elements go 0 as the remaining Fatou's argument is similar to Step 3 of the proof of Theorem D.3. Indeed, assume that all of these coefficients go to 0 as $n$ tends to infinity. Since $\bar{k} < k_0$, we always have at least one index $I \in [\bar{k}+1, k_0]$. Studying the coefficient of $f(x|\theta_I^0)$ proves that $(\lambda^* - \lambda_n)/D(\lambda_n G_n, \lambda^* G_*) \to 0$ as $n \to \infty$. From here, with similar argument as in Step 2 of claim (49), we can show that $1 = D(\lambda_n G_n, \lambda^* G_*)/D(\lambda_n G_n, \lambda^* G_*) \to 0$, which is a contradiction. Therefore, we obtain the conclusion of claim (59).

**Case 2:** $\lambda^* < \lambda_n$ for infinitely many $n$. Without loss of generality, we assume that $\lambda^* < \lambda_n$ for all $n \geq 1$. Under this case, we can rewrite equation (60) as follows:

$$p_{\lambda_n G_n}(x) - p_{\lambda^* G_*}(x) = \lambda_n \left( \underbrace{\sum_{i=1}^{k_n} p_i^n f(x|\theta_i^n)}_{:=f(x;G_n)} - \underbrace{\left[ \left( 1 - \frac{\lambda^*}{\lambda_n} \right) \sum_{i=\bar{k}+1}^{k_0} p_i^0 f(x|\theta_i^0) + \sum_{i=1}^{k_*} \frac{\bar{p}_i^*}{\lambda_n} f(x|\theta_i^*) \right]}_{:=f\left(x;\overline{G}_*(\lambda_n)\right)} \right),$$

where $\overline{G}_*(\lambda_n) := \left( 1 - \frac{\lambda^*}{\lambda_n} \right) G_0 + \frac{\lambda^*}{\lambda_n} G_*$. Under Case 2, $\bar{p}_i^* > \lambda^* p_i^* > 0$ for $i \in [k_*]$. Therefore, we can treat $f(x; G_n)$ and $f\left(x; \overline{G}_*(\lambda_n)\right)$ respectively as mixtures with $k_n$ and $k_0 + k_* - \bar{k}$ elements.

Without loss of generality, we assume $k_n = K$ for all $n$, namely, the setting where $G_n$ have full $K$ supports. We consider three separate settings of $K$.

**Case 2.1:** $K \leq k_* + k_0 - \bar{k} - 1$. Under this case, $G_n$ has fewer supports than $\overline{G}_*(\lambda_n)$. Hence, there always exists one element in the set $\{\theta_i^0 : \bar{k}+1 \leq i \leq k_0\} \cup \{\theta_j^* : 1 \leq j \leq k_*\}$ such that no supports of $G_n$ converge to. We first show that this element cannot belong to the set $\{\theta_j^* : 1 \leq j \leq k_*\}$. Assume by contrary that this element is in that set. Without loss of generality, we assume this element is $\theta_1^*$. Since $V(p_{\lambda_n G_n}, p_{\lambda^* G_*})/D(\lambda_n G_n, \lambda^* G_*) \to 0$, we have $f(x; G_n) - f(x; \overline{G}_*(\lambda_n)) \to 0$ for almost surely $x$. Since $\theta_i^n$ do not converge to $\theta_1^*$, the identifiability of $f$ and the previous limit imply that $\bar{p}_1^*/\lambda_n$ goes to 0 as $n \to \infty$, which is a contradiction as $\bar{p}_1^*/\lambda_n > \lambda^* p_1^*$.

Therefore, there exists an element in the set $\{\theta_i^0 : \bar{k} + 1 \leq i \leq k_0\}$ such that no elements of $G_n$ converge to. We assume without loss of generality that this element is $\theta_1^0$. In addition, all the elements in the set $\{\theta_j^* : 1 \leq j \leq k_*\}$ have at least one support of $G_n$ converge to. By performing Taylor expansion up to the second order around the limit points of the supports of $G_n$, we can view $(p_{\lambda_n G_n}(x) - p_{\lambda^* G_*}(x))/D(\lambda_n G_n, \lambda^* G_*)$ as a linear combination of elements of the forms $f(x|\theta_i^0), f(x|\theta_j^*), \frac{\partial f}{\partial \theta}(x|\theta_i^0), \frac{\partial f}{\partial \theta}(x|\theta_j^*), \frac{\partial^2 f}{\partial \theta^2}(x|\theta_i^0)$, and $\frac{\partial^2 f}{\partial \theta^2}(x|\theta_j^*)$ for some but not all $\bar{k}+1 \leq i \leq k_0$ and for all $j \in [k_*]$. Assume that all of the coefficients associated with these elements go to 0 as $n$ goes to infinity. Since no support of $G_n$ converges to $\theta_1^0$, the previous assumptions mean that $(\lambda_n - \lambda^*)/D(\lambda_n G_n, \lambda^* G_*) \to 0$. Given that result, we have

$$0 = \lim_{n \to \infty} \frac{V(p_{\lambda_n G_n}, p_{\lambda^* G_*})}{D(\lambda_n G_n, \lambda^* G_*)} = \lim_{n \to \infty} \frac{\lambda_n V(f(.; G_n), f(.; G_*))}{(\lambda_n + \lambda^*) W_2^2(G_n, G_*)},$$

which is a contradiction as $V(f(.; G_n), f(.; G_*))/W_2^2(G_n, G_*) \nrightarrow 0$ based on the result of Theorem 3.2 in [17]. Hence, not all the coefficients with $f(x|\theta_i^0), f(x|\theta_j^*), \frac{\partial f}{\partial \theta}(x|\theta_i^0), \frac{\partial f}{\partial \theta}(x|\theta_j^*), \frac{\partial^2 f}{\partial \theta^2}(x|\theta_i^0)$, and $\frac{\partial^2 f}{\partial \theta^2}(x|\theta_j^*)$ go to 0 as $n \to \infty$. From here, invoking the Fatou's argument and the identifiability of $f$, we conclude the claim (59) under Case 2.1.

**Case 2.2:** $K \geq k_* + k_0 - \bar{k}$. We see that the number of support points of $\bar{G}_*(\lambda_n)$ decreases to $k_*$ if $\lambda_n \to \lambda^*$ as $n \to \infty$ or keeps being $k_* + k_0 - \bar{k}$ for any subsequence of $\lambda_n$ does not converge to $\lambda^*$. In both cases, we are in the over-fitted setting as $K \geq k_* + k_0 - \bar{k}$. If $\lambda_n \to \lambda^*$, our assumption $W_2(G_n, \bar{G}_*(\lambda_n)) \to 0$ indicates that we can write $G_n$ as in equation (54) so that the atoms of $G_n$ converge to $\theta_i^*$ for $i \in [k_*]$ or 0. The proof of claim (59) goes through similar to what of Theorem 3.4 (or Theorem 3.2. in Ho et al. [17]).

If $\lambda_n \nrightarrow \lambda^*$ as $n \to \infty$ then $\bar{G}_*(\lambda_n)$ has $k_0 + k_* - \bar{k}$ in any of its limits. Hence this is over-fitted setting when $K \geq k_* + k_0 - \bar{k}$ and we can proceed similar to above to have claim (59).

**Case 2.3:** $K = k_* + k_0 - \bar{k}$ and $\lambda_n > \lambda^* + \delta > \lambda^*$ for all $n$. In this case, $\lambda_n \not\to \lambda^*$, so that $\bar{G}_*(\lambda_n)$ has $k_0 + k_* - \bar{k}$ in any of its limits. Hence, this is an exact-fitted setting and we can apply Theorem 3.1. in Ho et al. [17]. As a consequence, claim (59) is shown under Case 2.3.

### D.7 Proof of Theorem A.3

*Theorem* A.3. Assume that $h_0$ takes the form (7). Besides that, $K \geq k_0$ and $f$ is location-scale Gaussian distribution. Then, for any $\lambda \in [0,1]$ and $G \in \mathcal{O}_{K,c_0}(\Theta)$ for some $c_0 > 0$, there exist positive constants $C_1, C_2, C_3, C_4$ depending only on $\lambda^*, G_*, G_0, \Theta$ ($C_3$ and $C_4$ also depends on $\delta$) such that the following holds:

(a) When $K \leq k_* + k_0 - \bar{k} - 1$, then
$$V(p_{\lambda^*, G_*}, p_{\lambda, G}) \geq C_1 \overline{W}_{\bar{r}(K - k_*)}(\lambda G, \lambda^* G_*).$$

(b) When $K \geq k_* + k_0 - \bar{k}$, then
$$V(p_{\lambda^*, G_*}, p_{\lambda, G}) \geq C_2 \bigg( 1_{\{\lambda \leq \lambda^*\}} \overline{W}_{\bar{r}(K - k_*)}(\lambda G, \lambda^* G_*)$$
$$+ 1_{\{\lambda > \lambda^*\}} W_{\bar{r}(K - k_*)}^{\bar{r}(K - k_*)}(G, \overline{G}_*(\lambda)) \bigg)$$

(c) For $\delta > 0$, when $K = k_* + k_0 - \bar{k}$, we have
$$V(p_{\lambda^*, G_*}, p_{\lambda, G}) \geq C_3 1_{\{\lambda > \lambda^* + \delta\}} W_1(G, \overline{G}_*(\lambda)),$$
and when $K > k_* + k_0 - \bar{k}$, we have
$$V(p_{\lambda^*, G_*}, p_{\lambda, G}) \geq C_4 1_{\{\lambda > \lambda^* + \delta\}}$$
$$\times W_{\bar{r}(K - k_0 - k_* + \bar{k})}^{\bar{r}(K - k_0 - k_* + \bar{k})}(G, \overline{G}_*(\lambda)).$$

We still denote $\mathcal{T} = k_* + k_0 - \bar{k}$ and follow the path of Theorem 3.6 to prove by contradiction. We denote by $\bar{r}_1 = \bar{r}(K - k_*)$, $\bar{r}_2 = \bar{r}(K - k_0 - k_* + \bar{k})$, and
$$D(\lambda G, \lambda^* G_*) = \begin{cases} \overline{W}_{\bar{r}_1}(\lambda G, \lambda^* G_*), & \text{when } K \leq \mathcal{T} - 1 \\ 1_{\{\lambda \leq \lambda^*\}} \overline{W}_{\bar{r}_1}(\lambda G, \lambda^* G_*) + 1_{\{\lambda > \lambda^*\}}(\lambda + \lambda^*) W_{\bar{r}_2}^{\bar{r}_2}(G, \overline{G}_*(\lambda)), & \text{when } K \geq \mathcal{T} \end{cases}.$$

We need to show the following claim:
$$\lim_{\epsilon \to 0} \inf_{\lambda \in [0,1], G \in \mathcal{O}_K(\Theta)} \left\{ \frac{V(p_{\lambda G}, p_{\lambda^* G_*})}{D(\lambda G, \lambda^* G_*)} : D(\lambda G, \lambda^* G_*) \leq \epsilon \right\} > 0. \tag{61}$$

There exists sequences $\lambda_n$ and $G_n = \sum_{i=1}^{k_n} p_i^n \delta_{\theta_i^n} \in \mathcal{O}_K(\Theta)$ such that $D(\lambda_n G_n, \lambda^* G_*) \to 0$ and $V(p_{\lambda_n G_n}, p_{\lambda^* G_*})/D(\lambda_n G_n, \lambda^* G_*) \to 0$, where $D$ is the lower bound in the theorem statement. For the ease of presentation, we consider the one dimension Gaussian case where $(\mu, \Sigma) = (\theta, v)$, the higher dimension cases are treated similar.

**Case 1:** $\lambda^* \geq \lambda_n$ for infinitely many $n$. We can use the subsequence argument to have $\lambda^* \geq \lambda_n$ for all $n$ and $G_n$ can be assumed to have a fixed number of atoms (less than or equals $K$) and have a representation as in (54). In this case,
$$D(\lambda_n G_n, \lambda^* G_*) = |\lambda_n - \lambda^*| + (\lambda_n + \lambda^*) \overline{W}_{\bar{r}_1}^{\bar{r}_1}(G_n, G_*) \to 0, \quad \frac{V(p_{\lambda^* G_*}, p_{\lambda_n G_n})}{D(\lambda_n G_n, \lambda^* G_*)} \to 0. \tag{62}$$

Using Taylor expansion of $f$ around $\{(\theta_i^*, v_i^*)\}_{i=1}^{k_*}$ to the $\bar{r}_1$-th order we have
$$p_{\lambda_n G_n}(x) - p_{\lambda^* G_*}(x) = (\lambda^* - \lambda_n) \sum_{i=\bar{k}+1}^{k_0} p_i^0 f(x|\theta_i^0, v_i^0) + \lambda_n \bigg( \sum_{i=1}^{k_* + \underline{l}} \sum_{j=1}^{s_i} p_{ij}^n f(x|\theta_{ij}^n, v_{ij}^n) \bigg) - \sum_{i=1}^{k_*} \bar{p}_i^* f(x|\theta_i^*, v_i^*)$$
$$= (\lambda^* - \lambda_n) \sum_{i=\bar{k}+1}^{k_0} p_i^0 f(x|\theta_i^0, v_i^0) + \sum_{i=1}^{k_* + \underline{l}} \sum_{j=1}^{s_i} \lambda_n p_{ij}^n \sum_{|\alpha|=1}^{\bar{r}_1} (\Delta \theta_{ij}^n)^{\alpha_1} (\Delta v_{ij}^n)^{\alpha_2} \frac{1}{\alpha!} \frac{\partial^{|\alpha|} f(\theta_i^*, v_i^*)}{\partial^{\alpha_1} \theta \partial^{\alpha_2} v}$$
$$+ \sum_{i=1}^{k_* + \underline{l}} (\Delta \bar{p}_i^n) f(x|\theta_i^*, v_i^*) + R(x),$$

where $\boldsymbol{\alpha} = (\alpha_1, \alpha_2)$, $|\boldsymbol{\alpha}| = \alpha_1 + \alpha_2$, $\boldsymbol{\alpha}! = \alpha_1!\alpha_2!$, $\Delta\overline{p}_{i\cdot}^n = \lambda_n \sum_j p_{ij}^n - \overline{p}_i^*$, $\Delta\theta_{ij}^n = \theta_{ij}^n - \theta_i^*$, $\Delta v_{ij}^n = v_{ij}^n - v_i^*$ and $R(x) = O(\sum_{i=1}^{k_*+\underline{l}} \sum_{j=1}^{s_i} p_{ij}^n(|\Delta\theta_{ij}^n|^{\overline{r}_1} + |\Delta v_{ij}^n|^{\overline{r}_1}))$. Now we can use the character equation $\dfrac{\partial^2 f}{\partial\theta^2} = 2\dfrac{\partial f}{\partial v}$ to rewrite the formula above as

$$(\lambda^* - \lambda_n)\sum_{i=\overline{k}+1}^{k_0} p_i^0 f(x|\theta_i^0, v_i^0) + \sum_{\alpha=1}^{2\overline{r}_1}\sum_{i=1}^{k_*+\underline{l}}\left(\sum_{j=1}^{s_i}\lambda_n p_{ij}^n \sum_{n_1,n_2}\frac{(\Delta\theta_{ij}^n)^{n_1}(\Delta v_{ij}^n)^{n_2}}{2^{n_2}n_1!n_2!}\right)\frac{\partial^\alpha f(\theta_i^*, v_i^*)}{\partial\theta^\alpha}$$
$$+ \sum_{i=1}^{k_*+\underline{l}}(\Delta\overline{p}_{i\cdot}^n)f(x|\theta_i^*, v_i^*) + R(x), \quad (63)$$

where we sum over $n_1, n_2$ such that $n_1 + 2n_2 = \alpha, n_1 + n_2 \leq \overline{r}_1$. Now we turn into proving the non-vanishing coefficients. Assume that all coefficients in the formula above vanish when dividing by $D(\lambda_n G_n, \lambda^* G_*)$ when $n \to \infty$. Because

$$D(\lambda_n G_n, \lambda^* G_*) \asymp |\lambda_n - \lambda^*| + (\lambda_n + \lambda^*)\left(\sum_{i=1}^{k_*+\overline{l}}|\Delta p_{i\cdot}^n| + \sum_{i=1}^{k_*+\overline{l}}\sum_{j=1}^{s_i}p_{ij}^n(\|\Delta\theta_{ij}^n\|_2^{\overline{r}_1} + \|\Delta v_{ij}^n\|_2^{\overline{r}_1})\right) := D_{\overline{r}_1}(G_n, G_*),$$
$$(64)$$

we have

$$\frac{\lambda^* - \lambda_n}{D_{\overline{r}_1}(G_n, G_*)} \to 0, \quad \frac{\Delta\overline{p}_{i\cdot}^n}{D_{\overline{r}_1}(G_n, G_*)} \to 0. \quad (65)$$

These limits together imply

$$\frac{(\lambda^* + \lambda_n)\Delta\overline{p}_{i\cdot}^n}{D_{\overline{r}_1}(G_n, G_*)} \to 0, \quad \forall i = 1, \ldots, k_* + \overline{l}. \quad (66)$$

From the definition of $D_{\overline{r}_1}$, it can be deduced that there exists at least an index $i^*$ such that

$$\sum_{j=1}^{s_{i*}}\frac{(\lambda_n + \lambda^*)p_{i^*j}^n((\theta_{ij}^n)^{\overline{r}_1} + (v_{ij}^n)^{\overline{r}_1})}{D_{\overline{r}_1}(G_n, G_*)} \not\to 0. \quad (67)$$

Without loss of generality, assign $i^* = 1$. But as we assume all the coefficients in equation (63) go to 0 for all $\alpha$ and $i$, we have

$$\frac{\sum_{j=1}^{s_1}\lambda_n p_{1j}^n \sum_{\substack{n_1+2n_2=\alpha \\ n_1+n_2\leq\overline{r}_1}}\frac{(\theta_{1j}^n)^{n_1}(v_{1j}^n)^{n_2}}{2^{n_2}n_1!n_2!}}{D_{\overline{r}_1}(G_n, G_*)} \to 0, \quad (68)$$

for all $\alpha = 1, \ldots, 2\overline{r}_1$. From two expressions above combining with equation (65), we have for all $\alpha = 1, \ldots, 2\overline{r}_1$,

$$F_\alpha := \frac{\sum_{j=1}^{s_1}p_{1j}^n \sum_{\substack{n_1+2n_2=\alpha \\ n_1+n_2\leq\overline{r}_1}}\frac{(\Delta\theta_{1j}^n)^{n_1}(\Delta v_{1j}^n)^{n_2}}{2^{n_2}n_1!n_2!}}{\sum_{j=1}^{s_1}p_{1j}^n((\Delta\theta_{ij}^n)^{\overline{r}_1} + (\Delta v_{ij}^n)^{\overline{r}_1})} \to 0. \quad (69)$$

If $s_1 = 1$ then substituting $\alpha = 1$ and $\alpha = 2\overline{r}_1$ gives

$$\frac{|\Delta\theta_{11}^n|^{\overline{r}_1}}{|\Delta\theta_{11}^n|^{\overline{r}_1} + |\Delta v_{11}^n|^{\overline{r}_1}}, \quad \frac{|\Delta v_{11}^n|^{\overline{r}_1}}{|\Delta\theta_{11}^n|^{\overline{r}_1} + |\Delta v_{11}^n|^{\overline{r}_1}} \to 0,$$

which is impossible as they are sum up to 1 for all $n$. Hence $s_1 \geq 2$. Now we proceed to show the contradiction using the system of equations (6). Denote by $\overline{p}_n = \max_{1\leq j\leq s_1}\{p_{1j}^n\}, \overline{M}_n = \max_{1\leq j\leq s_1}\{|\Delta\theta_{1j}^n|, |\Delta v_{1j}^n|^{1/2}\}$. By the subsequence argument in compact sets, without loss of generality, we can denote $c_j^2 := \lim_{n\to\infty} p_{1j}^n/\overline{p}_n, a_j = \lim \Delta\theta_{1j}^n/\overline{M}_n$, and $b_j = \lim \Delta v_{1j}^n/\overline{M}_n$ for all $j = 1, \ldots, k_* + \overline{l}$. Because of the definition of $\mathcal{O}_{K,c_0}$, we have $p_j \geq c_0$ for all $j$, which implies all $c_j$ are different from 0 and at least one of them is 1. Similarly, in $(a_j, b_j)_j$, there is at least one of

them equals to 1 or $-1$. Dividing both numerators and denominators of equation (69) by $\bar{p}_n \overline{M}_n^\alpha$, we have

$$\sum_{j=1}^{s_1} \sum_{n_1+2n_2=\alpha} \frac{c_j^2 a_j^{n_1} b_j^{n_2}}{n_1! n_2!} = 0,$$

for all $\alpha = 1, \ldots, \bar{r}_1$. Hence, we get the contradiction, where we use the fact that $s_1 \leq K - k_* + 1$ (as $s_i \geq 1$ for all $i \geq 2$) and $\bar{r}_1 = \bar{r}(K - k_*)$ is the smallest number such that equation (6), where $k = K - k_*$, has the trivial solution only. After that, we can argue as in the Step 9 of Proposition 2.2. in [16] to get the contradiction to the assumption proposed in the beginning, where we use the fact that Gaussian family is identifiable up to any order with respect to the location parameters.

**Case 2:** $\lambda^* \leq \lambda_n$ for all $n$. We rewrite

$$p_{\lambda_n G_n}(x) - p_{\lambda^* G_*}(x) = \lambda_n \bigg( \underbrace{\sum_{i=1}^{k_n} p_i^n f(x|\theta_i^n)}_{:=f(x;G_n)} - \underbrace{\bigg[ \bigg(1 - \frac{\lambda^*}{\lambda_n}\bigg) \sum_{i=\bar{k}+1}^{k_0} p_i^0 f(x|\theta_i^0) + \sum_{i=1}^{k_*} \frac{\bar{p}_i^*}{\lambda_n} f(x|\theta_i^*) \bigg]}_{:=f\left(x;\overline{G}_*(\lambda_n)\right)} \bigg),$$

(70)

**Cases 2.1.** $K \leq \mathcal{T} - 1$, argue similarly to Case 2.1. of the proof of Theorem 3.6, we have $\dfrac{\lambda_n - \lambda^*}{D(\lambda_n G_n, \lambda^* G_*)} \to 0$ as $n \to \infty$. Now we arrive at the equation (65) of Case 1. Follow the argument above, we can prove claim (61).

**Case 2.2.** $K \geq \mathcal{T}$, we can see equation (70) as an over-fitted mixture of location-scale Gaussian setting where the number of over-fitted atoms is at most $K - k_*$. Hence we can argue similar to Case 1 or the Proposition 2.2. in [16] to obtain the conclusion.

**Cases 2.3.** $K = \mathcal{T}$ and $\lambda_n > \lambda^* + \delta$ for all $n$. From the presentation as in equation (70), we can see that $1 - \dfrac{\lambda^*}{\lambda_n}$ does not vanish in any of it limits. Therefore $\overline{G}_*(\lambda_n)$ has $k_* + k_0 - \bar{k} = \mathcal{T}$ number of components in its limits. Because this is an exact-fitted setting, we can apply Theorem 3.1. in Ho et al. [17] to get the result of claim (61)

**Cases 2.4.** $K > \mathcal{T}$ and $\lambda_n > \lambda^* + \delta$ for all $n$, we can also see that $\overline{G}_*(\lambda_n)$ has $k_* + k_0 - \bar{k} = \mathcal{T}$ number of components in its limits. We can apply Proposition 2.2. in Ho et al. [16] to get the result of claim (61).

### D.8 Proof of Theorem A.4

*Theorem* A.4. Assume that $h_0$ takes the form (7) and $\bar{k} = k_0$. Besides that, $f$ is second order identifiable. Then, for any $\lambda \in [0, 1]$ and $G \in \mathcal{O}_K(\Theta)$ that $K \geq k_*$, there exist positive constants $C_1$ and $C_2$ depending only on $\lambda^*, G_*, G_0, \Theta$ such that the following holds:

(a) If $\mathcal{I}(\lambda)$ is not ratio-independent, then

$$V(p_{\lambda^* G_*}, p_{\lambda G}) \geq C_1 \bigg[ 1_{\{\lambda \in \mathcal{B}^c\}} + 1_{\{\lambda \in \mathcal{B}\}} W_2^2(G, \bar{G}_*(\lambda)) \bigg]. \tag{71}$$

(b) If $\mathcal{I}(\lambda)$ is ratio-independent, then

$$V(p_{\lambda^* G_*}, p_{\lambda G}) \geq C_2 \bigg[ 1_{\{\lambda \in \mathcal{B}^c\}} \bigg( \sum_{i \in \mathcal{I}(\lambda)} \Big[ (\lambda^* - \lambda)p_i^0 - \lambda^* p_i^* \Big]$$

$$+ \mathcal{S}(\mathcal{I}(\lambda)) W_2^2(G, \widetilde{G}_*(\lambda)) \bigg)$$

$$+ 1_{\{\lambda \in \mathcal{B}\}} W_2^2(G, \bar{G}_*(\lambda)) \bigg]. \tag{72}$$

To ease the ensuing presentation, we denote $D(\lambda G, \lambda^* G_*) = 1_{\{\lambda \in \mathcal{B}^c\}} \Big( \sum_{i \in \mathcal{I}(\lambda)} \Big[ (\lambda^* - \lambda) p_i^0 - \lambda^* p_i^* \Big] + \mathcal{S}(\mathcal{I}(\lambda)) W_2^2(G, \widetilde{G}_*(\lambda)) \Big) + 1_{\{\lambda \in \mathcal{B}\}} W_2^2(G, \bar{G}_*(\lambda))$ when $\mathcal{I}(\lambda)$ is ratio-independent or $D(\lambda G, \lambda^* G_*) = 1_{\{\lambda \in \mathcal{B}^c\}} + 1_{\{\lambda \in \mathcal{B}\}} W_2^2(G, \bar{G}_*(\lambda))$ when $\mathcal{I}(\lambda)$ is not ratio-independent.

In order to prove the theorem, it is sufficient to verify the following inequality:

$$\lim_{\epsilon \to 0} \inf_{\lambda \in [0,1], G \in \mathcal{E}_{k_*}(\Theta)} \left\{ \frac{V(p_{\lambda G}, p_{\lambda^* G_*})}{D(\lambda G, \lambda^* G_*)} \, : \, D(\lambda G, \lambda^* G_*) \leq \epsilon \right\} > 0. \tag{73}$$

**Proof of claim** (73): Assume that the above claim is not true. It implies that there exist sequences $G_n = \sum_{i=1}^{k_n} p_i^n \delta_{\theta_i^n} \in \mathcal{O}_K(\Theta)$ and $\lambda_n \in [0,1]$ such that $D(\lambda_n G_n, \lambda^* G_*)$ and $V(p_{\lambda_n G_n}, p_{\lambda^* G_*}) / D(\lambda_n G_n, \lambda^* G_*)$ go to 0 as $n$ approaches to infinity. Since $\bar{k} = k_0$ and $G_*$ admits the form (10), we find that

$$p_{\lambda_n G_n}(x) - p_{\lambda^* G_*}(x) = \lambda_n \left( \sum_{i=1}^{k_n} p_i^n f(x|\theta_i^n) \right) - \sum_{i=1}^{k_*} \bar{p}_i^* f(x|\theta_i^*), \tag{74}$$

where $\bar{p}_i^* = \lambda^* p_i^* + (\lambda_n - \lambda^*) p_i^0$ when $1 \leq i \leq k_0$ and $\bar{p}_i^* = \lambda^* p_i^*$ otherwise. In addition, $\theta_i^* = \theta_i^0$ for $i \in [k_0]$. From this presentation, we see that there must exists a constant $C$ depending on $\lambda^*, G_*, G_0$ such that $\liminf \lambda_n > C$. Indeed, suppose it is not the case, then by the subsequence argument, we can assume that $\lambda_n \to 0$. Besides, $V(\lambda_n G_n, \lambda^* G_*) \to 0$, we have $\bar{p}_i^* \to 0$ for all $i \in [k_*]$. These conditions lead to $p_i^* = 0$ for all $i > k_0$ and $p_i^0 = p_i^*$ for all $i \in [k_0]$, which mean that $G_* = G_0$ (a contradiction to our assumption). Hence, limits of $(\lambda_n)$ is bounded below. We have two settings with $\lambda_n$.

**Case 1:** $\lambda_n \in \mathcal{B}$ for infinitely many $n$. Without loss of generality, we assume that $\lambda_n \in \mathcal{B}$ for all $n \geq 1$. If $k_* = k_0$ then we see that $\bar{p}_i^*$ can not vanish simultaneously when $n \to \infty$ for all $i$, otherwise we have $G^* = G_0$, which contradicts to the assumption in this section. Otherwise, $k_* > k_0$, and $\bar{p}_i^*$ does not vanish for all $i > k_0$. Therefore, every limit of $\sum_{i=1}^{k_*} \bar{p}_i^* f(x|\theta_i^*)$ has a number of atoms ranging from $\max\{1, k_* - k_0\}$ to $k_*$, which is less than or equal to $K$. So that this is an over-fitted scenario. In addition, $D(\lambda_n G_n, \lambda^* G_*) = W_2^2(G_n, \bar{G}_*(\lambda_n))$. We can further rewrite equation (74) as:

$$p_{\lambda_n G_n}(x) - p_{\lambda^* G_*}(x) = \lambda_n (f(x; G_n) - f(x; \bar{G}_*(\lambda_n))).$$

From Theorem 3.2 in Ho et al. [17], we have $V(f(.; G_n), f(.; \overline{G}_*(\lambda_n)))/W_2^2(G_n, \bar{G}_*(\lambda_n)) \not\to 0$ as $n \to \infty$. Putting the above results together, we obtain that $V(p_{\lambda_n G_n}, p_{\lambda^* G_*})/D(\lambda_n G_n, \lambda^* G_*) \not\to 0$, which is a contradiction. Hence, we reach the conclusion of claim (74).

**Case 2:** $\lambda_n \notin \mathcal{B}$ for infinitely many $n$. Without loss of generality, we assume that $\lambda_n \notin \mathcal{B}$ for all $n \geq 1$. Under this setting, $\mathcal{I}(\lambda_n) \neq \emptyset$. In addition, for any $i \in \mathcal{I}(\lambda_n)$, $\bar{p}_i^* < 0$. Given these conditions, we can rewrite equation (74) as follows:

$$p_{\lambda_n G_n}(x) - p_{\lambda^* G_*}(x) = \sum_{i \in \mathcal{I}(\lambda_n)} (-\bar{p}_i^*) f(x|\theta_i^0) + \Big[ \lambda_n \left( \sum_{i=1}^{k_n} p_i^n f(x|\theta_i^n) \right) - \sum_{i \in \mathcal{I}(\lambda_n)^c} \bar{p}_i^* f(x|\theta_i^0)$$
$$- \sum_{i=k_0+1}^{k_*} \bar{p}_i^* f(x|\theta_i^*) \Big] \tag{75}$$

We have two separate settings with $\mathcal{I}(\lambda_n)$.

**Case 2.1:** $\mathcal{I}(\lambda_n)$ is not ratio-independent. Under this case, $D(\lambda_n G_n, \lambda^* G_*) = 1$. Since $V(p_{\lambda_n G_n}, p_{\lambda^* G_*})/D(\lambda_n G_n, \lambda^* G_*) \to 0$, we have $V(p_{\lambda_n G_n}, p_{\lambda^* G_*}) \to 0$. It indicates that $p_{\lambda_n G_n}(x) - p_{\lambda^* G_*}(x) \to 0$ almost surely $x$. Since $-\bar{p}_i^* > 0$ for all $i \in \mathcal{I}(\lambda_n)$, the previous limit demonstrates that $\bar{p}_i^* \to 0$ for all $i \in \mathcal{I}(\lambda_n)$, which leads to $p_i^*/p_i^0 = p_j^*/p_j^0$ for all $i, j \in \mathcal{I}(\lambda_n)$. It contradicts the assumption that $\mathcal{I}(\lambda_n)$ is not ratio-independent. Hence, we achieve the conclusion of claim (74) under Case 2.1.

**Case 2.2:** $\mathcal{I}(\lambda_n)$ is ratio-independent. Under this case, $D(\lambda_n G_n, \lambda^* G_*) = \sum_{i \in \mathcal{I}(\lambda_n)} \Big[ (\lambda^* -$
$\lambda_n) p_i^0 - \lambda^* p_i^* \Big] + \mathcal{S}(\mathcal{I}(\lambda_n)) W_2^2(G_n, \widetilde{G}_*(\lambda_n)) \to 0$ and $V(p_{\lambda_n G_n, \lambda^* G_*})/D(\lambda_n G_n, \lambda^* G_*) \to 0$,
which imply $V(p_{\lambda_n G_n, \lambda^* G_*}) \to 0$. We first prove that $\mathcal{S}(\mathcal{I}(\lambda_n)) \not\to 0$. Indeed, suppose it is not the
case, then $p_i^* = 0$ for all $i > k_0$ and $(\lambda^* p_i^* + (\lambda_n - \lambda^*) p_i^0) \to 0$ for all $i \in \mathcal{I}(\lambda_n)$. From equation
(75) and the fact that $V(p_{\lambda_n G_n, \lambda^* G_*}) \to 0$, we also see that $\bar{p}_i^* \to 0$ for all $i \in \mathcal{I}(\lambda_n)$ and $\lambda_n \to 0$.
But that means

$$\lambda_n \to 0, \lambda^* p_i^* + (\lambda_n - \lambda^*) p_i^0 \to 0, \quad \forall i \in [k_0].$$

Those limits together imply that $\lambda^*(p_i^0 - p_i^*) = 0$ for all $i \in [k_0]$, which is contradictory with
our assumption that $G^* \neq G_0$. Hence $\mathcal{S}(\mathcal{I}(\lambda_n)) \not\to 0$. As $D(\lambda_n G_n, \lambda^* G_*) \to 0$, we have
$W_2^2(G_n, \widetilde{G}_*(\lambda_n)) \to 0$ as $n \to \infty$. It implies that we can rewrite $G_n$ as follows:

$$G_n = \sum_{i \in \mathcal{I}(\lambda_n)^c \cup \{k_0+1, \ldots, k_*+\bar{l}\}} \sum_{j=1}^{s_i} p_{ij}^n \delta_{\theta_{ij}^n}, \tag{76}$$

where $\sum_{j=1}^{s_i} p_{ij}^n \to \bar{p}_i^*/\mathcal{S}(\mathcal{I}(\lambda_n))$ and $\theta_{ij}^n \to \theta_i^*$ for all $i \in \mathcal{J} := \mathcal{I}(\lambda_n)^c \cup \{k_0 + 1, \ldots, k_* + \bar{l}\}$.
Here, $\bar{p}_i^* = 0$ for $k_* + 1 \leq i \leq k_* + \bar{l}$. In addition, $\sum_{i \in \mathcal{J}} s_i = k'$ for some $k'$ such that
$k_* - k_0 + |\mathcal{I}(\lambda_n)^c| \leq k' \leq k_*$. To faciliate the proof argument, we denote $\Delta \theta_{ij}^n := \theta_{ij}^n - \theta_i^*$ and
$\Delta p_{i.}^n := \sum_{j=1}^{s_i} p_{ij}^n - \bar{p}_i^*/\mathcal{S}(\mathcal{I}(\lambda_n))$ for $i \in \mathcal{J}$. The result of Lemma 3.1 in Ho et al. [18] leads to

$$W_2^2(G_n, \tilde{G}_*(\lambda_n)) \asymp \sum_{i \in \mathcal{J}} |\Delta p_{i.}^n| + \sum_{i \in \mathcal{J}} \sum_{j=1}^{s_i} p_{ij}^n \left\| \Delta \theta_{ij}^n \right\|_2^2. \tag{77}$$

Invoking Taylor's expansion up to the second order, we have

$$p_{\lambda_n G_n}(x) - p_{\lambda^* G_*}(x) = \sum_{i \in \mathcal{I}(\lambda_n)} (-\bar{p}_i^*) f(x|\theta_i^0) + \sum_{i \in \mathcal{J}} (\lambda_n \sum_{j=1}^{s_i} p_{ij}^n - \bar{p}_i^*) f(x|\theta_i^*)$$
$$+ \lambda_n \left( \sum_{j=1}^{s_i} p_{ij}^n \Delta \theta_{ij}^n \right)^\top \frac{\partial f}{\partial \theta}(x|\theta_i^*) + \lambda_n \left( \sum_{j=1}^{s_i} p_{ij}^n \left( \Delta \theta_{ij}^n \right)^\top \frac{\partial^2 f}{\partial \theta^2}(x|\theta_i^*)(\Delta \theta_{ij}^n) \right) + R(x),$$
$$\tag{78}$$

where $R(x)$ is Taylor remainder such that $R(x) = o\left( \lambda_n \sum_{i \in \mathcal{J}} \sum_{j=1}^{s_i} p_{ij}^n \left\| \Delta \theta_{ij}^n \right\|_2^2 \right)$. Therefore, we
have $R(x)/D(\lambda_n G_n, \lambda^* G_*) \to 0$ as $n \to \infty$.

The expression in equation (78) indicates that we can view $(p_{\lambda_n G_n}(x) - p_{\lambda^* G_*}(x))/D(\lambda_n G_n, \lambda^* G_*)$
as a linear combination of elements of the forms $f(x|\theta_i^0)$, $f(x|\theta_j^*)$, $\frac{\partial f}{\partial \theta}(x|\theta_j^*)$, $\frac{\partial^2 f}{\partial \theta^2}(x|\theta_j^*)$ for $i \in$
$\mathcal{I}(\lambda_n)$ and $j \in \mathcal{J}$. Assume that the coefficients of these terms go to 0 as $n$ approaches infinity. By
studying the coefficients of $f(x|\theta_i^0)$ when $i \in \mathcal{I}(\lambda_n)$, we find that

$$\left( \sum_{i \in \mathcal{I}(\lambda_n)} (-\bar{p}_i^*) \right)/D(\lambda_n G_n, \lambda^* G_*) \to 0.$$

Given the above result, as the coefficients of $f(x|\theta_i^*)$ and $\frac{\partial^2 f}{\partial \theta^2}(x|\theta_i^*)$ go to 0 when $i \in \mathcal{J}$, we obtain

$$\frac{\mathcal{S}(\mathcal{I}(\lambda_n)) \sum_{j=1}^{s_i} p_{ij}^n - \bar{p}_i^*}{D(\lambda_n G_n, \lambda^* G_*)} = \frac{[\lambda_n - (\sum_{l \in \mathcal{I}(\lambda_n)} \bar{p}_l^*))] \sum_{j=1}^{s_i} p_{ij}^n - \bar{p}_i^*}{D(\lambda_n G_n, \lambda^* G_*)} \to 0,$$

$$\frac{\mathcal{S}(\mathcal{I}(\lambda_n)) \sum_{j=1}^{s_i} p_{ij}^n \left\| \Delta \theta_{ij}^n \right\|_2^2}{D(\lambda_n G_n, \lambda^* G_*)} = \frac{[\lambda_n - (\sum_{l \in \mathcal{I}(\lambda_n)} \bar{p}_l^*))] \sum_{j=1}^{s_i} p_{ij}^n \left\| \Delta \theta_{ij}^n \right\|_2^2}{D(\lambda_n G_n, \lambda^* G_*)} \to 0$$

Putting the above results together, given the expression in equation (77), we obtain $1 =$
$D(\lambda_n G_n, \lambda^* G_*)/D(\lambda_n G_n, \lambda^* G_*) \to 0$ as $n \to \infty$, which is a contradiction. Therefore, not
all the coefficients of $f(x|\theta_i^0)$, $f(x|\theta_j^*)$, $\frac{\partial f}{\partial \theta}(x|\theta_j^*)$, $\frac{\partial^2 f}{\partial \theta^2}(x|\theta_j^*)$ when $i \in \mathcal{I}(\lambda_n)$ and $j \in \mathcal{J}$. From
here, we utilize the Fatou's argument from the previous proofs to obtain the conclusion of claim (73)
under Case 2.2.

## D.9 Proof of Theorem A.5

*Theorem* A.5. Assume that $h_0$ takes the form (7) and $\bar{k} = k_0$. Besides that, $f$ is location-scale Gaussian distribution. Then, for $\tilde{k} := \max\{k_* - k_0, 1\}$, and for any $\lambda \in [0,1]$ and $G \in \mathcal{O}_{K,c_0}(\Theta)$ for some $K \geq k_*$ and $c_0 > 0$, there exist positive constants $C_1$ and $C_2$ depending only on $\lambda^*, G_*, G_0, \Theta$ such that on $\lambda^*, G_*, G_0, \Theta$ such that

(a) If $\mathcal{I}(\lambda)$ is not ratio-independent, then

$$V(p_{\lambda^* G_*}, p_\lambda G) \geq C_1 \left[ 1_{\{\lambda \in \mathcal{B}^c\}} \right.$$
$$\left. + 1_{\{\lambda \in \mathcal{B}\}} W_{\bar{r}(K-\tilde{k})}^{\bar{r}(K-\tilde{k})}(G, \bar{G}_*(\lambda)) \right]. \tag{79}$$

(b) If $\mathcal{I}(\lambda)$ is ratio-independent, then

$$V(p_{\lambda^*, G_*}, p_{\lambda, G}) \geq C_2 \left[ 1_{\{\lambda \in \mathcal{B}^c\}} \left( \sum_{i \in \mathcal{I}(\lambda)} \left[ (\lambda^* - \lambda) p_i^0 - \lambda^* p_i^* \right] \right.\right.$$
$$\left. + \mathcal{S}(\mathcal{I}(\lambda)) W_{\bar{r}(K-\tilde{k})}^{\bar{r}(K-\tilde{k})}(G, \widetilde{G}_*(\lambda)) \right)$$
$$\left. + 1_{\{\lambda \in \mathcal{B}\}} W_{\bar{r}(K-\tilde{k})}^{\bar{r}(K-\tilde{k})}(G, \bar{G}_*(\lambda)) \right]. \tag{80}$$

The proof of Theorem A.5 is similar to what of Theorem A.4 and with the technical details borrowed from Theorem A.3. Therefore we only highlight the main differences. Denote by $D(\lambda G, \lambda^* G_*) = 1_{\{\lambda \in \mathcal{B}^c\}} \left( \sum_{i \in \mathcal{I}(\lambda)} \left[ (\lambda^* - \lambda) p_i^0 - \lambda^* p_i^* \right] + \mathcal{S}(\mathcal{I}(\lambda)) W_{\bar{r}(K-\tilde{k})}^{\bar{r}(K-\tilde{k})}(G, \widetilde{G}_*(\lambda)) \right) + 1_{\{\lambda \in \mathcal{B}\}} W_{\bar{r}(K-\tilde{k})}^{\bar{r}(K-\tilde{k})}(G, \bar{G}_*(\lambda))$ when $\mathcal{I}(\lambda)$ is ratio-independent or $D(\lambda G, \lambda^* G_*) = 1_{\{\lambda \in \mathcal{B}^c\}} + 1_{\{\lambda \in \mathcal{B}\}} W_{\bar{r}(K-\tilde{k})}^{\bar{r}(K-\tilde{k})}(G, \bar{G}_*(\lambda))$ when $\mathcal{I}(\lambda)$ is not ratio-independent.

In order to prove the theorem, it is sufficient to verify the following inequality:

$$\lim_{\epsilon \to 0} \inf_{\lambda \in [0,1], G \in \mathcal{E}_{k_*}(\Theta)} \left\{ \frac{V(p_{\lambda G}, p_{\lambda^* G_*})}{D(\lambda G, \lambda^* G_*)} : D(\lambda G, \lambda^* G_*) \leq \epsilon \right\} > 0. \tag{81}$$

**Proof of claim** (81): Assume that the above claim is not true. It implies that there exist sequences $G_n = \sum_{i=1}^{k_n} p_i^n \delta_{\theta_i^n} \in \mathcal{O}_K(\Theta)$ and $\lambda_n \in [0,1]$ such that $D(\lambda_n G_n, \lambda^* G_*)$ and $V(p_{\lambda_n G_n}, p_{\lambda^* G_*})/D(\lambda_n G_n, \lambda^* G_*)$ go to 0 as $n$ approaches to infinity. Since $\bar{k} = k_0$ and $G_*$ admits the form (10), we find that

$$p_{\lambda_n G_n}(x) - p_{\lambda^* G_*}(x) = \lambda_n \left( \sum_{i=1}^{k_n} p_i^n f(x|\theta_i^n) \right) - \sum_{i=1}^{k_*} \bar{p}_i^* f(x|\theta_i^*), \tag{82}$$

where $\bar{p}_i^* = \lambda^* p_i^* + (\lambda_n - \lambda^*) p_i^0$ when $1 \leq i \leq k_0$ and $\bar{p}_i^* = \lambda^* p_i^*$ otherwise. In addition, $\theta_i^* = \theta_i^0$ for $i \in [k_0]$. One could argue as in Theorem A.4 to get $(\lambda_n)$ being bounded below.

**Case 1:** $\lambda_n \in \mathcal{B}$ for infinitely many $n$. Without loss of generality, we assume that $\lambda_n \in \mathcal{B}$ for all $n \geq 1$. Under this case, every limit of $\sum_{i=1}^{k_*} \bar{p}_i^* f(x|\theta_i^*)$ has a number of atoms ranging from $\tilde{k}$ to $k_*$, which is less than or equal to $K$. So that this is an over-fitted scenario where the number of over-fitted atoms is at most $K - \tilde{k}$. In addition, $D(\lambda_n G_n, \lambda^* G_*) = W_{\bar{r}(K-\tilde{k})}^{\bar{r}(K-\tilde{k})}(G_n, \bar{G}_*(\lambda_n))$. We can further rewrite equation (82) as:

$$p_{\lambda_n G_n}(x) - p_{\lambda^* G_*}(x) = \lambda_n(f(x; G_n) - f(x; \bar{G}_*(\lambda_n))).$$

Now we can argue similarly to the proof Theorem A.3 or Proposition 2.2. in [16] to get $V(p_{\lambda_n G_n}, p_{\lambda^* G_*})/D(\lambda_n G_n, \lambda^* G_*) \not\to 0$, which combines with the fact that $\lambda_n \not\to 0$ gives us a contradiction. Hence, we reach the conclusion of claim (81)

**Case 2:** $\lambda_n \notin \mathcal{B}$ for infinitely many $n$. Without loss of generality, we assume that $\lambda_n \notin \mathcal{B}$ for all $n \geq 1$. Under this setting, $\mathcal{I}(\lambda_n) \neq \emptyset$. In addition, for any $i \in \mathcal{I}(\lambda_n)$, $\bar{p}_i^* < 0$. Given these conditions, we can rewrite equation (74) as follows:

$$
p_{\lambda_n G_n}(x) - p_{\lambda^* G_*}(x) = \sum_{i \in \mathcal{I}(\lambda_n)} (-\bar{p}_i^*) f(x|\theta_i^0) + \left[ \lambda_n \left( \sum_{i=1}^{k_n} p_i^n f(x|\theta_i^n) \right) - \sum_{i \in \mathcal{I}(\lambda_n)^c} \bar{p}_i^* f(x|\theta_i^0) \right.
$$

$$
\left. - \sum_{i=k_0+1}^{k_*} \bar{p}_i^* f(x|\theta_i^*) \right] \tag{83}
$$

We have two separate settings with $\mathcal{I}(\lambda_n)$.

**Case 2.1:** $\mathcal{I}(\lambda_n)$ is not ratio-independent. This is the same as Case 2.1. of Theorem A.4. With a similar argument, we can show that $\mathcal{I}(\lambda_n)$ must be ratio-independent, which is a contradiction. Hence, we get claim (81) under this case.

**Case 2.2:** $\mathcal{I}(\lambda_n)$ is ratio-independent. We can see that the second term of equation (83) is in an over-fitted setting with the number of extra components being at most $K - \tilde{k}$. Arguing similar to Case 2.2. of Theorem A.3 gives us the conclusion of claim (81).