# OpenReview forum: "Beyond black box densities: Parameter learning for the deviated components"
_NeurIPS.cc/2022/Conference — NeurIPS 2022 Accept_

### Official Review · Reviewer_eLfh · 2022-07-11

**Rating:** 5
**Confidence:** 2
**Soundness:** 3 good
**Presentation:** 3 good
**Contribution:** 2 fair

**Summary:**

This paper considers the "deviating mixture model", which is defined by $(1-\lambda^*) h_0 + \lambda^* (\sum_{i=1}^kp_i^* f(x \mid \theta_i^*))$, where $h_0$ is known and the distribution family $f$ is known.
Main Contributions:
- The authors propose a notion of distinguishability of this model.
- The authors provide the converging rate of MLE for this model (in estimating $\lambda*$ and $\theta_i's$).
- The authors provide simulations to verify the bounds they derived.


**Questions:**

See the weaknesses part.

**Limitations:**

NA.

**Strengths And Weaknesses:**

The strengths of the paper lie in the follow two parts:
- The first part is on the novelty: for proposing the "deviating mixture model", and also the definition of the distinguishability.
- The second part is on the rates of convergence part, where assuming the distinguishability conditions, the authors derived inverse bounds to upper bound the Wasserstein distance by the total variation distance (which is known to be upper bounded by Hellinger distance).

The weaknesses of paper are mainly on the significance and importance of this proposed "deviating mixture model". In what scenes/example we can use the "deviating mixture model" to model the data distribution? Some questions in my mind (that need more explanations/arguments) are:
-  The authors mentioned that, from some black-box methods (NN or kernel method) we can get the density estimation $h_0$. But in those cases, $h_0$ itself is just an estimate (a maybe complex deviation) of the underlying true distribution. Why can one further assume that the new data is from a deviation of $h_0$, where the deviation is from a known distribution family?
- The assumption that the distribution family $f$ is known, and the number of components in the mixture ($k^*$) is bounded, I think also needs motivations from practice or explanations.

---

> ### Author Response · Authors · 2022-07-28
> **Practicality of the assumptions in Deviating Mixture Models**
>
> We thank the reviewer for the comments on our work. We address the reviewer's concern below.
>
> **1. [Use $h_0$ as a generating model]** We are considering the scenario where in the beginning, we have a big data set and we use a complex model $h_0$ (e.g., Normalizing Flows) to estimate the density of the data. If the sample size and the model are large enough (a neural network with many nodes and layers), we can estimate the density of data so well (that one can even sample from it and fake the real data). Therefore we can treat $h_0$ as the generating model of the initial dataset. The trade-off here is that $h_0$ is complicated and requires lots of computational resources to train. Now suppose new data come in and the model needs to evolve. The proposed model is useful that it can *adapt* to the new data, *reuse* the old model, and is *interpretable* (detect new data subpopulations via deviating components). It is also much faster to train a mixture model via the E-M algorithm than re-train a complex model like $h_0$ for the new dataset.
>
> Another point of view is to see this model from testing problems: One can test alternative (possibly composite) hypothesis represented by a class of distributions against the null hypothesis represented by $h_0$. In this scenario, scientist can have information about $h_0$ a priori (as a summary statistics from an old data set, or is estimated from another data set of the same kind), and they wish to learn if there is any other information in the new data set that they are working with. The deviating mixture model will work well in that case.
>
> We have added a concise discussion in the Introduction section regarding this question. Thank you.
>
> **2. [Known kernel $f$ and $k^{*}$ is bounded]**  From the practical point of view, even when the mixture of some simple kernels $f$ seems to be misspecified, sometimes people still use it to fit the model, because at least it can capture the subpopulations and their statistics (e.g., means, variances) up to some biases. This is common because "all models are wrong, but some are useful". Statisticians often choose the kernel based on the nature of data and their experience. For example, we can choose Gamma mixtures to model positive data, or Gaussian Mixtures to learn both means and variances of each subpopulation. Similar to Deviating Mixture Models, if we choose $f$ to be Gaussian and have distinguishability between $h_0$ and $f$, then we still can learn about the deviating components (from new data) using this model.
>
> From the theoretical point of view, to the best of our knowledge, the theory for misspecified mixture models (wrong kernel $f$) has still remained open in the literature. We view our theories for the exactly-specified settings of deviating mixture models as the first step toward a deeper understanding of these models.
>
> **Bounded $k^{*}$:** In practice, when the true generative model has finite $k^*$, and we do not know it, we often fit the model with a Deviating Mixture Model with a large $K$ number of components. Theoretically, we prove that in the distinguishable case, no matter how large $K$ is compared to $k^*$, we still have $n^{-1/4}$ convergence rate of $\overline{W}_{2}(\lambda G, \lambda^* G^*)$ when $n\to \infty$ (Theorem 3.4). However, in the partially distinguishable or weakly identifiable cases, we have a slower convergence rate as $n \to \infty$ (Theorem 3.5 and 3.7).
>
> **When $k^{*} = \infty$:** The only available approach for the theoretical study of parameter learning is in [R.1]. When applying Theorem 2 of [1] in our partially distinguishable settings, where $f$ is a location family of distribution and $h_0 = f * G_0$ for some latent mixing measure $G_0$, then we obtain the bound
> $V(p_{\lambda G}, p_{\lambda^* G_*})^{m} \gtrsim W_2^2((1-\lambda) G_0 + \lambda G, (1-\lambda^*)G_0 + \lambda^* G_*) $ when the function $f$ is ordinary smooth, such as Gamma or Laplace distribution,
>
> or
> $(-\log V(p_{\lambda G}, p_{\lambda^* G_*}))^{-2/\beta} \gtrsim W_2^2((1-\lambda) G_0 + \lambda G, (1-\lambda^*)G_0 + \lambda^* G_*),$ when the function $f$ is supersmooth, such as Gaussian distribution or Cauchy distribution (The detailed definitions of ordinary and supersmooth density functions can be found in [R.1]). However, this bound is not sharp in our setting in the sense that it can not distinguish the convergence rate of $\lambda$ and of $G$ separately. We leave this question about sharp bound for infinite components deviating mixture models for the future work.
>
> Reference:
>
> [R.1] Nguyen, XuanLong. "Convergence of latent mixing measures in finite and infinite mixture models." The Annals of Statistics 41.1 (2013): 370-400.

---

> > ### Author Response · Authors · 2022-08-07
> > **Details on the novelty of inverse bounds compared to previous work**
> >
> > Hi reviewer,
> >
> > We want to clarify the novelty of the inverse bounds in our paper in more details. At a high level, the difficulties in our setting arise from the distinguishability of $h_0$ from the mixture of $f$. In particular, if $h_0$ is very different from a mixture of $f$ (Distinguishable setting), then we have the parameter estimation in the model is well behaved in the sense that we can estimate the deviating proportion $\lambda$ at a parametric rate. If $h_0$ is approximated by a mixture of $f(x|\theta)$ or is a mixture of $f(x|\theta)$ itself, then we may lose the identifiability in our model and can not estimate $\lambda$ well. In some cases, it does not even converge. We discuss it in details below.
> >
> > **[Convergence rate of parameter estimation in vanilla models]** Consider a distribution family $f(x|\theta)$ and denote a mixture of them as $f_G(x) = \sum_{i=1}^{k} p_i f(x|\theta_i)$ for $G = \sum_{i=1}^{k} p_i \delta_{\theta_i}$. This family is strongly identifiable in the second order. So that for fixed true $G_*$ having $k_*$ atoms, the inverse bound for them have the form $V(p_G, p_{G_*})\gtrsim W_1(G, G_*)$ for all $G$ having exactly $k_*$ atoms *(Exact-fitted case)*. The inverse bound is $V(p_G, p_{G_*})\gtrsim W_2^2(G, G_*)$ for all $G$ having no more than $K$ atoms, for $K$ is larger than $k_*$ *(Over-fitted case)*. It means that we can estimate $G_*$ at $1/\sqrt{n}$ rate in the exact-fitted case and at $1/n^{1/4}$ rate in the over-fitted case.
> >
> > **[Distinguishable settings]** By Theorem 3.3 and Theorem 3.4, we proved that if $h_0$ is distinguishable from mixtures of $f$, then we have inverse bound $$V(p_{\lambda G}, p_{\lambda^* G_*}) \gtrsim |\lambda - \lambda^*| + W_1(G, G_*)$$ in the exact-fitted case, and
> > $$V(p_{\lambda G}, p_{\lambda^* G_*}) \gtrsim |\lambda - \lambda^*| + W_2^2(G, G_*),$$
> > in the over-fitted case. It means that we can estimate $\lambda^*$ at a parametric rate for both cases, and the estimation rate of $G_*$ is not affected by the signal from $h_0$ (so it stays $1/\sqrt{n}$ in the exact-fitted setting and $1/n^{1/4}$ in the over-fitted setting). We want to clarify that this result is of interest on its own, and it is useful when we combine it with the theory of distinguishability that we developed in Section 2, where we allow $h_0$ to be estimated by a KDE or Neural network. Therefore it is different from the vanilla mixture in its spirit and applications.
> >
> > **[Partially distinguishable settings]** This setting may seem the most like theory of vanilla mixtures. However, many theoretical difficulties may arise when there are interactions between $h_0$ and mixtures of $f$. Recall that here we assume $h_0 = p_{G_0} = \sum_{i=1}^{k_0} p_i \delta_{\theta^0_i}$, and $G^* = \sum_{i=1}^{k_*} p_i^* \delta_{\theta_i^*}$. Hence, the true generating model is
> > $$p_{\lambda^* G_*}(x) = (1-\lambda^*)\sum_{i=1}^{k_0} p_i^0 f(x|\theta_i^0) + \lambda^* \sum_{i=1}^{k_0} p_i^* f(x|\theta_i^*), \quad (1)$$
> >
> > and the working model that we use to estimate the data is
> >
> > $$p_{\lambda G}(x) = (1-\lambda)\sum_{i=1}^{k_0} p_i^0 f(x|\theta_i^0) + \lambda \sum_{i=1}^{K} p_i f(x|\theta_i). \quad (2)$$
> >
> > So the interesting issue here is the interaction between $G_0$ and $G_*$: What if they have common atoms? If yes, then how does it affect the convergence of $\lambda$ and $G$?
> >
> > As a concrete example, we take $k^* = k_0 = 2$, and assume that $G_*$ and $G_0$ share one atom $\theta_1^* = \theta_1^0$, then the model (1) becomes
> >
> > $$p_{\lambda^* G_*}(x) =  ((1-\lambda^*)p_1^0 + \lambda^* p_1^*) f(x|\theta_1^0) + (1-\lambda^*)p_2^0 f(x|\theta_2^0) + \lambda^* p_2^* f(x|\theta_2^*). \quad (3)$$
> >
> > We wonder which part of the working model $(2)$ estimate the first component $((1-\lambda^*)p_1^0 + \lambda^* p_1^*) f(x|\theta_1^0)$? On the one hand, the coefficient $\lambda$ in model $(2)$ can be estimated such that $(1-\lambda) p_1^0 = (1-\lambda^*)p_1^0 + \lambda^* p_1^*$. On the other hand, one component (say, $\hat{\theta}_1$) of $G$ in model $(2)$ and also converge to $\theta_1^0$ and
> > $(1-\lambda) p_1^0 + \lambda \hat{\theta}_1 \to (1-\lambda^*) p_1^* + \lambda^* p_1^0$.
> >
> > It takes several results to completely characterize the convergence behavior in this setting. One can see Theorem 3.6 (or A.1-A.5), which is our treatment of the partially distinguishable setting. At a high level, we show that the convergence of $\lambda$ and $G$ can be characterized from the identifiable equation $p_{\lambda G} = p_{\lambda^* G_*}$. In particular, convergence can have different behavior in different solution regimes of this equation. For illustrating purpose, we can also see the experiment in Appendix B, where $\lambda$ may converge to somewhere different than $\lambda^*$.
> >
> > We hope this discussion can help the readers and reviewers see this setting better. Thank you.

---

> ### Author Response · Authors · 2022-08-09
> **Looking forward to your feedback.**
>
> Dear Reviewer eLfh,
>
> We have addressed your concerns in our responses about the motivation of the deviated mixture models as well as providing a discussion when the kernel $f$ is misspecified or the value of the true number of components $k^\*\$ is unbounded.
>
> Given that the deadline is tomorrow, 3:00 pm (CST time) on August 9th, we would like to hear your feedback. Please feel free to raise questions if you have other concerns.
>
> Best regards,
>
> Authors

---

> ### Comment · Reviewer_eLfh · 2022-08-09
> **Response**
>
> Thanks to the author(s) for the detailed answers. I acknowledged that I've read the authors' responses and other reviewers' comments. I update my score from 4 to 5.

---

> > ### Author Response · Authors · 2022-08-09
> > **Thank you for your response**
> >
> > Thank you for your response and updating the score. Please keep engage with us if you have any further question.
> >
> > Best regards,
> > Authors.

---

### Official Review · Reviewer_UbgM · 2022-07-12

**Rating:** 6
**Confidence:** 2
**Soundness:** 4 excellent
**Presentation:** 3 good
**Contribution:** 3 good

**Summary:**

The paper presents a new model called the deviating mixture model to capture the phenomenon that a known probability density estimate in practice could shift with new data coming in. The shift or deviation is modeled as a mixture distribution in the following manner: $(1-\lambda)h_0 + \lambda \sum_{i=1}^k p_i f(x | \theta_i)$.
$h_0$ is known already by some means, we care about the deviation from $h_0$.
The work studies conditions on $h_0$ and $f(.|\theta)$ under which maximum likelihood estimation can recover the parameters of the deviation.
A key property which is typically assumed in such estimation problems is that of identifiability: that is, two distributions with sufficiently different parameters produce statistically distinguishable distributions.
To identify parameters in the presence of $h_0$ we need a stronger condition which the authors call distinguishability. They follow this up with multiple examples of $h_0$ for which the deviation modeled as a mixture of Gaussians is distinguishable.
The authors then proceed to show bounds for recovering the parameters of the deviation under the assumptions of distinguishability. First they show that for Gaussian mixture models convergence in Wasserstein metric implies convergence in parameters.
Then they show that the total variation distance between two deviation models is lower bounded by a quantity which closely relates to the Wasserstein distance (under the assumption of distinguishability).
They also provide results under partial distinguishability assumptions and supplement theory with some experiments on synthetic data to show that empirically their rate of convergence matches theoretical predictions.


**Questions:**

**Questions:**
1. On the face of it, the deviating mixture setting seems closely related to the well studied setting of learning mixtures of Gaussians. Could you elaborate on what makes this problem harder and where the technical challenges you need to overcome are?

Overall the paper is well written and well presented. Some minor suggestions below.
**Suggestions:**
1. Explicitly state what is meant by location and scale parameters of a Gaussian.
2. Explicitly state that distinguishability implies identifiability.
3. I believe it could help readers to see a high level overview of some of the salient proofs in the paper in the main body rather than a comprehensive set of Theorem statements. This will help the reader understand why this setting is more challenging than the vanilla mixture of Gaussians setting.


**Limitations:**

One limitation of the work which the authors could add a brief discussion on is the lack of strong non-asymptotic rates for high-dimensional settings. i.e. for a really large (but finite) neural network, the distinguishability assumption is still satisfied and hence MLE will give a convergence rate that scales as $1/\sqrt{n}$ but as the neural net becomes larger it could become harder and harder to distinguish it from the deviation. How does the error in parameter recovery scale with the size of the neural network?

The authors have adequately addressed any potential negative societal impact of their work.

**Strengths And Weaknesses:**

**Strengths:**
1. The paper is well written and is clear. It could do with a bit more thorough preliminaries discussion in places but the flow and presentation are done well.
2. As far as I can tell, the results of the paper are original and novel and the direction is one of relevance to NeurIPS audience.
3. Understanding conditions under which we can recover parameters of a distribution from samples is an important problem and the current paper looks at a twist on the classical setting wherein we know one component of a mixture and are trying to learn the remaining components. The paper presents a significant set of results in this direction.

**Weaknesses:**
1. The bounds focus mainly on the convergence rates with respect to $n$ the number of samples and do not focus on the dimension of the distributions being considered. In practice, as dimensions can be huge, the approaches outlined in this paper might be impractical?
2. A possible artifact of the above issue is that we can only run the MLE procedure on low-dimensional data. Is computation the reason why the synthetic experiments were only performed on low-dimensional settings? It would be interesting to see if the approach can be scaled up to settings where the target distribution has at least 10-20 dimensions.

---

> ### Author Response · Authors · 2022-07-28
> **Convergence rate concerning the dimensionalities**
>
> We thank the reviewer for the comments on our work. We address the reviewer's concern below.
>
> **Weakness 1. [Theoretical bounds concerning both sample size $n$ and dimension $d$]**
> Our technique to prove the convergence rate of parameter estimation is divided into 2 steps: **Convergence rate of density estimation in Hellinger distance** (Section 3.1) and **Inverse bound** (Section 3.3 and 3.4). So, to discuss the rate concerning both $n$ and $d$, let us talk about difficulties in both steps.
>
> Firstly, for **density estimation**, the standard technique to prove the convergence rate in Hellinger distance is to use the Empirical Processes theory [1, 2]. However, this theory requires us to bound the "$\epsilon$-bracketing number" of the space of densities $\mathcal{F}$ (Assumption A3), which is the smallest number of pairs $(f_{j}^{L}, f_{j}^{U})_{j=1}^{N}$ such
> that the $l2$ norm $|f_j^U - f_j^L|<\epsilon$ and
> for every $f\in \mathcal{F}$, we have $f_j^L(x) \leq f(x) \leq f_j^U(x) \forall  x$. This is a very strong notion of covering numbers and is difficult to bound. Using the technique in Proposition 3.2, we can only bound this bracketing number by $\log(\epsilon) \times C^{d}$ for a constant $C$ depending on the radius of parameter space $\Theta$, which leads to a convergence rate like $C^{d} \sqrt{\log(n)/n}$. Of course, this result does not mean much when $d$ is large.
>
> Secondly, for the **inverse bounds**, if we look carefully into the proof, for example, of Theorem 3.3, we can see that the only step that can be affected by the dimensionality is Step 3. Basically, the key here is we prove that the ratio
> $$V(p_{\lambda, G}, p_{\lambda^*, G_*}) / \overline{W}(\lambda G, \lambda^* G_*) \geq \int_{x} \left|\alpha_{0}h_0(x) + \sum_{i=1}^{k^*}\alpha_i f(x|\theta_i^*) +  \sum_{i=1}^{k^*}\beta_i \dfrac{\partial f(x|\theta_i^*)}{\partial \theta}\right| dx,$$
> which is greater than 0, because of the distinguishability condition. Therefore to understand this lower bound exactly, we need to study the infimum of the right-hand side of the equation above for all $\alpha_{0}, \alpha_{i}, \beta_{i}$, where at least one of them is non-zero (with respect to $d$). This is non-trivial and may be related to both kernel $f$ and the dimension $d$.
>
> **Weakness 2. [Practicability in high-dimensional setting]** From the discussion above, we can see that when $d = O(\log n)$ then our theory of convergence rate still works well. Therefore, when $d$ is as large as 10 or 20, if $n = exp(10) \approx 20000$, we still have a nice convergence rate to the true parameters as discussed in the paper.
> For the other regime where $d$ can be as large as $n$, it is still an active research topic in the mixture modeling community (with some new results about the minimax rate can be seen in [3]). We are also interested in this direction but we leave it as future work for now.
>
> **Questions [Difference between Deviating Mixture Models and Gaussian Mixtures]** The theory of Deviating Mixture Models is more general than Gaussian Mixture Models.
> In the distinguishable case of Deviating Mixture Models, we need to provide several examples of complex models that satisfy distinguishability conditions. We choose to present results related to Gaussians because it is mostly used in practice. The partially distinguishable case and weakly identifiable is most like Gaussian Mixtures. However, an interesting phenomenon here arises from the fact that the component of $h_0$ can be shared with other components in deviating parts, which makes the convergence rate harder to analyze and we need to consider several smaller cases.
>
> Please also refer to our response to Reviewer iPiq for more explanation about the novelty of our work.
>
>  **Suggestions:** Thank you for your recommendations. We have added those details into the revised version. After every inverse bounds theorem, we have included several comments about techniques used in the proofs.
>
> **Strong non-asymptotic rates in high dimensional settings concerning the size of the neural network:** Thank you for your comment. We also believe that asymptotic theory can help guiding non-asymptotic understanding of parameter estimation in practice. We have added a concise discussion about the effect of the complexity of neural networks on convergence rates in Section 5 of the revised version.
>
> Reference:
>
> [1] Vaart, AW van der, and Jon A. Wellner. "Weak convergence and empirical processes with applications to statistics." Journal of the Royal Statistical Society-Series A Statistics in Society 160.3 (1997): 596-608.
>
> [2] Geer, Sara A. Empirical Processes in M-estimation. Vol. 6. Cambridge university press, 2000.
>
> [3] Doss, Natalie, et al. "Optimal estimation of high-dimensional Gaussian location mixtures." (2021).

---

> > ### Comment · Reviewer_UbgM · 2022-08-06
> > **Thanks for the response**
> >
> > The response confirms some of my questions about the limitations of the current work (weakness 1 and weakness 2).
> > Regarding the challenge of this setting compared to the vanilla mixture of Gaussians setting: can you elaborate more on where the technical difficulties arise? Perhaps with the help of a couple of examples?

---

> > > ### Author Response · Authors · 2022-08-07
> > > **Challenge of Deviating Mixture Model compared to the vanilla mixture of Gaussians**
> > >
> > > Hi reviewer,
> > >
> > > Thank you for your response. At a high level, the difficulties here arise from the distinguishability of $h_0$ from the mixture of Gaussians. In particular, if $h_0$ is very different from a mixture of Gaussians (Distinguishable setting), then we have the parameter estimation in the model is well behaved in the sense that we can estimate the deviating proportion $\lambda$ at a parametric rate. If $h_0$ is approximated by a mixture of Gaussians or is a mixture of Gaussian itself, then we may lose the identifiability in our model and can not estimate $\lambda$ well. In some cases, it does not even converge. For a detailed explanation, you can see three paragraphs below.
> > >
> > > **[Convergence rate of parameter estimation in mixture of Gaussian models]** For simplicity, we consider location Gaussian distribution $$f(x|\theta) = \dfrac{1}{(2\pi)^{d/2}} \exp(-\frac{1}{2}|x-\theta|^2),$$ and denote a mixture of them as $f_G(x) = \sum_{i=1}^{k} p_i f(x|\theta_i)$ for $G = \sum_{i=1}^{k} p_i \delta_{\theta_i}$. This family is strongly identifiable in the second order. So that for fixed true $G_*$ having $k_*$ atoms, the inverse bound for them have the form $V(p_G, p_{G_*})\gtrsim W_1(G, G_*)$ for all $G$ having exactly $k_*$ atoms *(Exact-fitted case)*. The inverse bound is $V(p_G, p_{G_*})\gtrsim W_2^2(G, G_*)$ for all $G$ having no more than $K$ atoms, for $K$ is larger than $k_*$ *(Over-fitted case)*. It means that we can estimate $G_*$ at $1/\sqrt{n}$ rate in the exact-fitted case and at $1/n^{1/4}$ rate in the over-fitted case.
> > >
> > > **[Distinguishable settings]** By Theorem 3.3 and Theorem 3.4, we proved that if $h_0$ is distinguishable from mixtures of $f$, then we have inverse bound $$V(p_{\lambda G}, p_{\lambda^* G_*}) \gtrsim |\lambda - \lambda^*| + W_1(G, G_*)$$ in the exact-fitted case, and
> > > $$V(p_{\lambda G}, p_{\lambda^* G_*}) \gtrsim |\lambda - \lambda^*| + W_2^2(G, G_*),$$
> > > in the over-fitted case. It means that we can estimate $\lambda^*$ at a parametric rate for both cases, and the estimation rate of $G_*$ is not affected by the signal from $h_0$ (so it stays $1/\sqrt{n}$ in the exact-fitted setting and $1/n^{1/4}$ in the over-fitted setting). We want to clarify that this result is of interest on its own, and it is useful when we combine it with the theory of distinguishability that we developed in Section 2, where we allow $h_0$ to be estimated by a KDE or Neural network. Therefore it is different from the vanilla mixture of Gaussians in its spirit and applications.
> > >
> > > **[Partially distinguishable settings]** This setting may seem the most like a vanilla mixture of Gaussians, as the reviewer mentioned. However, many theoretical difficulties may arise when there are interactions between $h_0$ and mixtures of $f$. Recall that here we assume $h_0 = p_{G_0} = \sum_{i=1}^{k_0} p_i \delta_{\theta^0_i}$, and $G^* = \sum_{i=1}^{k_*} p_i^* \delta_{\theta_i^*}$. Hence, the true generating model is
> > > $$p_{\lambda^* G_*}(x) = (1-\lambda^*)\sum_{i=1}^{k_0} p_i^0 f(x|\theta_i^0) + \lambda^* \sum_{i=1}^{k_0} p_i^* f(x|\theta_i^*), \quad (1)$$
> > >
> > > and the working model that we use to estimate the data is
> > >
> > > $$p_{\lambda G}(x) = (1-\lambda)\sum_{i=1}^{k_0} p_i^0 f(x|\theta_i^0) + \lambda \sum_{i=1}^{K} p_i f(x|\theta_i). \quad (2)$$
> > >
> > > So the interesting issue here is the interaction between $G_0$ and $G_*$: What if they have common atoms? If yes, then how does it affect the convergence of $\lambda$ and $G$?
> > >
> > > As a concrete example, we take $k^* = k_0 = 2$, and assume that $G_*$ and $G_0$ share one atom $\theta_1^* = \theta_1^0$, then the model (1) becomes
> > >
> > > $$p_{\lambda^* G_*}(x) =  ((1-\lambda^*)p_1^0 + \lambda^* p_1^*) f(x|\theta_1^0) + (1-\lambda^*)p_2^0 f(x|\theta_2^0) + \lambda^* p_2^* f(x|\theta_2^*). \quad (3)$$
> > >
> > > We wonder which part of the working model $(2)$ estimate the first component $((1-\lambda^*)p_1^0 + \lambda^* p_1^*) f(x|\theta_1^0)$? On the one hand, the coefficient $\lambda$ in model $(2)$ can be estimated such that $(1-\lambda) p_1^0 = (1-\lambda^*)p_1^0 + \lambda^* p_1^*$. On the other hand, one component (say, $\hat{\theta}_1$) of $G$ in model $(2)$ and also converge to $\theta_1^0$ and
> > > $(1-\lambda) p_1^0 + \lambda \hat{\theta}_1 \to (1-\lambda^*) p_1^* + \lambda^* p_1^0$.
> > >
> > > It takes several results to completely characterize the convergence behavior in this setting. One can see Theorem 3.6 (or A.1-A.5), which is our treatment of the partially distinguishable setting. At a high level, we show that the convergence of $\lambda$ and $G$ can be characterized from the identifiable equation $p_{\lambda G} = p_{\lambda^* G_*}$. In particular, convergence can have different behavior in different solution regimes of this equation. For illustrating purpose, we can also see the experiment in Appendix B, where $\lambda$ may converge to somewhere different than $\lambda^*$.
> > >
> > > We hope this discussion can help the readers and reviewers see this setting better. Thank you.

---

> ### Author Response · Authors · 2022-08-05
> **Looking forward to your feedback**
>
> Dear Reviewer UbgM,
>
> We have addressed your concerns in our responses. We would like to hear your feedback. Please feel free to raise questions if you have other concerns.
>
> Best regards,
>
> Authors

---

### Official Review · Reviewer_sdP4 · 2022-07-16

**Rating:** 6
**Confidence:** 2
**Soundness:** 4 excellent
**Presentation:** 3 good
**Contribution:** 3 good

**Summary:**

The paper considers a deviating mixture model in which they try to establish rates for parameter estimation. The deviating mixture model is of the form $(1-\lambda^*) h_0 + \lambda^* \sum_{i=1}^{k} p_i^* f(x|\theta_i^*)$ where $h_0$ is known but $\lambda^*, p^*$ and $theta^*$ are unknown. The authors analyze the maximum likelihood and provide rates under some assumptions on the complexity of the class of allowed distributions (I am not an expert on this but is related to covering numbers). The assumption is justified via a proposition that location scale Gaussian families satisfy the assumption. They provide analysis both when $k$ is known and unknown. Along the way, they define a notion of distinguishability between effectively $h_0$ and $f$.

**Questions:**

Does A.3 work for truncated sub-Gaussian distributions? Does it suffice that the covariance matrix is well-behaved (in terms of eigenvalues)?

**Limitations:**

This is theoretical work.

**Strengths And Weaknesses:**

Pros
The paper is studying an interesting setting. Technically looks non-trivial, though the reviewer is not familiar with the literature to argue about its novelty.

Cons:
I tried to read the overview with the contributions and the section is not well-written, the writing needs to be improved and further discussion on the techniques should be added.

---

> ### Author Response · Authors · 2022-07-27
> **Contributions and Assumptions in the paper**
>
> Thank you for reviewing our paper. We address your questions and concerns as below.
>
> **1. [Overview and contributions/techniques]** Motivation of this paper from an applied point of view is that: suppose we are trying to estimate the density of a large data set, and we take *a long time* to train a *complex model* $h_0$ to fit this data set well (by a kernel or mixture or neural network method). After a period of time, we may collect more data into this dataset and we ask: "$h_0$ still can fit this new data set well or not?"
>
> We do not want to waste time and computing resource to retrain this complex model $h_0$, and we also want to interpret a new heterogeneous cluster in the data if it exists. Therefore, we propose Deviating Mixture Models, where there is a known complex component $h_0$ and a mixture of components of deviating new clusters. As stated, this model has the ability to *adapt, reuse,* and *interpret*.
>
> We are interested in studying the large-sample theory of parameter estimation for this model. When we look at it from mixture modeling literature [1-3], the main contributions are:
>
> (i) We are able to provide *identifiability notion* for mixture models of different types of distributions (that we call Distinguishability --Definition 2.2), including complex structured distributions (such as through a neural network). At a high level, these conditions are required to learn the parameters in the model well. We also prove several interested models satisfying these conditions (Theorem 2.4, Corollary 2.5, and 2.6). Those results are novel and relevant to the mixture models research community.
>
> (ii) The link between *identifiability notion* and *statistical efficiency* of mixture models is through the *inverse bounds*. By carefully considering all the cases of distinguishability conditions, we characterize different inverse bounds for different scenarios of strong and weak distinguishable mixtures (Section 3.3, 3.4, Appendix A). Those results are examined by simulation in Section 4 and Appendix B of the paper.
>
> **2. [Discussion on the techniques]** We have added a proof sketch for the first inverse bound (Theorem 3.3) in the main text so that readers can understand the proof technique easier. All other inverse bounds are proved with similar spirit but different in some details. We commented the differences and challenges after each result.
>
> **3. [Assumption A.3.]** Generally speaking, this assumption is satisfied for most parametric families of distributions. As long as the distributions family of interest $f(x|\theta)$ is regular enough in the sense that we have $\sup_{x} |f(x|\theta) - f(x|\theta')| \lesssim \|\theta-\theta'\|$ and have sub-exponential tail. When we have those conditions, a proof similar to the proof of Proposition 3.2 will go through. The technique here is to use the first condition to show a bound for covering number with respect to $l_{\infty}$ norm (from the proof in the appendix), i.e.
> $$\log N(\epsilon, P_k(\Theta), |\cdot|_{\infty}) \lesssim \log(1/\epsilon).$$
>
> After that, we can combine it with a bound of tails of $f(x|\theta)$ to construct a bracketing net for this space of distributions. Specifically, we start from an $\eta-$net $f_1, \dots, f_N$ of $(P_k(\Theta), |\cdot|_{\infty})$ that we know (from previous step) that $N\lesssim \log(1/\eta)$. We can perturb every element of this set by a small amount $\eta$, and combine with tail bound (such as sub-Gaussian) to construct a bracketing net. The radius of this bracketing net is controlled by $\eta$ and the tail probability of our distribution family. As long as the tail probability is sub-exponential, we can choose $\eta$ such that this radius is $\epsilon$ with $\log(1/\epsilon)\asymp \log(1/\eta)$. The details can be found in the proof of Proposition 3.2.
>
> *So, specifically for your question, the result will hold for sub-gaussian distributions $f(x|\theta)$ with $\sup_{x} |f(x|\theta) - f(x|\theta')| \lesssim \|\theta-\theta'\|$.*
>
> The bounded condition on the covariance matrix of Gaussian distributions is necessary for the proof (to control the tail probability in Step 2 above). It is also needed in practice because the likelihood of mixture models on the data can go to infinity without this condition (See Section 9.2.1 in [4]).
>
> We have added the discussion above after the proof of Prop. 3.2 (in blue color). Thank you.
>
> References:
>
> [1] McLachlan, Geoffrey J., and Kaye E. Basford. Mixture models: Inference and applications to clustering. Vol. 38. New York: M. Dekker, 1988.
>
> [2] Nguyen, XuanLong. "Convergence of latent mixing measures in finite and infinite mixture models." The Annals of Statistics 41.1 (2013): 370-400.
>
> [3] Chen, Jiahua. "Optimal rate of convergence for finite mixture models." The Annals of Statistics (1995): 221-233.
>
> [4] Bishop, Christopher M., and Nasser M. Nasrabadi. Pattern recognition and machine learning. Vol. 4. No. 4. New York: springer, 2006.

---

> > ### Author Response · Authors · 2022-08-05
> > **Looking forwards to your feedback**
> >
> > Dear Reviewer sdP4,
> >
> > We have addressed your concerns in our responses. We would like to hear your feedback. Please feel free to raise questions if you have other concerns.
> >
> > Best regards,
> >
> > Authors

---

> > ### Comment · Reviewer_sdP4 · 2022-08-07
> > **Thank you for your response**
> >
> > Dear authors,
> >
> > I would like to thank you for your response. You addressed my concerns and I feel the paper is above the acceptance bar. I still lack some knowledge of the literature though and my confidence is quite low. I will continue the discussion with fellow reviewers and AC.

---

> > > ### Author Response · Authors · 2022-08-08
> > > **Thanks for your endorsement.**
> > >
> > > Dear Reviewer sdP4,
> > >
> > > Thanks for your support of our paper. We really appreciate it. Please feel free to raise questions if you have other concerns.
> > >
> > > Best,
> > >
> > > Authors

---

### Official Review · Reviewer_iPiq · 2022-07-19

**Rating:** 6
**Confidence:** 3
**Soundness:** 4 excellent
**Presentation:** 2 fair
**Contribution:** 2 fair

**Summary:**


This paper studies a model known as ``deviating mixture model’’ which is a mixture of a single known distribution h0 and an mixture model whose parameters are unknown. The authors modify previous work to give a notion of distinguishability for when one can learn the parameters of the new mixture model. Furthermore, they use this notion, to give convergence estimates under wasserstein metrics.


More concretely,  the authors give a definition of identifiability (heavily based on prior work [15,21] ) for this new deviated model. This notion extends to the case when h0 is a distribution that is not too heavy tailed or light tailed or a distribution captured by Neural Networks. Using an Assumption A.3 on the complexity of the mixture model, they can upper bound the Hellinger distance between MLE estimate and true optimal bounds decay proportional to sqrt( log n/ n) given n samples. The main part of the paper then shows that these bounds also imply the same convergence for the parameters in the wasserstein metric. To do so, they show "inverse bounds" of the form that the total variation distance is bounded below by the Wasserstein distance. Since the hellinger distance upper bounds the TV distance, this completes the proof.

The main proof technique goes as follows:
1)  First we show that if total variation distance is not bounded below by W-distance, one can look at the taylor expansion of the difference of the distributions and express it as a linear combination of the first few derivatives of the functions
2) This linear combination must have non-zero coefficients and must also converge to 0 almost everywhere.
3) However this contradicts the distinguishability assumption which states that the coefficients must be zero if the linear combination of these derivatives is 0 almost everywhere.

The authors also extend this setting to the case where the number of components of the mixture model is not known but only an upper bound is known.




**Questions:**

I noticed that Theorem 3.1 does not have any citation or proof. Could you please provide one?

Line 118, should $i$ range from $1$ to $k$ instead of $k'$?
Line 278, should "limited spaces" should be limited space.


**Limitations:**

I think the authors do a good job of describing the limitations of their work.

**Strengths And Weaknesses:**

Strengths

The work shows a notion of identifiability which is valid deviating mixture model. Using this definition they show that small TV distance implies that the parameters are close in wasserstein distance.

The proofs seem to be complete, rigorous and thorough. The proofs are able to show this even when the known distribution h0 is a black box model such as a Neural network. This seems nice and very useful in practice.


Weakness:

The main notion of identifiability that is used is largely derived from prior work [21,15]. Furthermore the bounds on hellinger distance, which already imply bounds on the TV distance follow from prior work (Thm 3.1). The main novelty is that this also implies bounds on the Wasserstein distance in the parameters, however this is not entirely surprising.

The presentation is a bit terse and notation heavy with less intuition given about the proofs in the main body. One has to read the appendix to even get any intuition of the proofs.

I gave this a weak reject only because I am not entirely convinced of the novelty of work but I am happy to be convinced otherwise.

---

> ### Author Response · Authors · 2022-07-26
> **Novelty of the work**
>
> We thank the reviewer for the comments on our work. We address the reviewer's concern below.
>
> **1. [Notions of identifiability and inverse bounds]** We would like to clarify that the novelty of our work lies in the new notion of distinguishability conditions rather than the strong identifiability condition that the reviewer mentioned. Let us take this chance to explain the difference between the two notions. The notion of distinguishability (Definition 2.2) characterizes the interaction between the function $h_{0}$ and the mixture of density functions $f$, which is important to understand the behaviors of parameter estimation in the deviated mixture model. On the other hand, the notion of strong identifiability (Definition 2.1) is mainly to characterize the interaction of parameters inside the mixture of density functions $f$, which is not sufficient to study the parameter estimation in the deviated mixture model. Furthermore, another contribution of our paper is to provide several examples of distinguishablity condition (including complex structured distributions), which to the best of our knowledge are new and have not been established before in the literature. Finally, we also establish
> a rather comprehensive picture for the lower bound of the total variational distance between the density functions in terms of different degrees of divergences between the corresponding parameters under several settings of distinguishability or partial distinguishability of the models. For detailed explanations, please refer to our points [1a] and [1b] below.
>
> **[1a] (Difference between Strong Identifiability and Distinguishability Conditions)** Because the previous works only work with mixture of densities $\{ f(x|\theta): \theta\in \Theta \}$, they only need to define the strong identifiability with respect to combinations of this family, i.e., if for $k$ different elements $\theta_1, \dots, \theta_k\in \Theta$ if there is coefficients $\alpha_{\eta}^{i}$ such that
> \begin{equation}
> \sum_{l=0}^{r} \sum_{|\eta| = l}\sum_{I=1}^{k} \alpha_{\eta}^{(i)} \dfrac{\partial^{|\eta|} f}{\partial \theta^{\eta}} f(x|\theta_i) = 0,
> \end{equation}
> then all the coefficients  $\alpha_{\eta}^{i} = 0$, and then they prove this condition is satisfied for some classical distributions families such as Gaussians of Students ([Ref. 17 in the main text]). It is very important to show examples of which class of distributions satisfies these conditions, as it allows us to fit corresponding mixture models in practice with confidence.
>
> In this work, because we consider the mixture models in the form of a convex combination between a complex distribution $h_0$ and a mixture of densities $\{f(x|\theta): \theta\in \Theta\}$, we need a (strong) notion of distinguishability between $h_0$ and $f$, i.e., if there is coefficients $\alpha_{\eta}^{i}$ such that
> \begin{equation}
> \alpha^{(0)}h_0(x) + \sum_{l=0}^{r} \sum_{|\eta| = l}\sum_{I=1}^{k} \alpha_{\eta}^{(i)} \dfrac{\partial^{|\eta|} f}{\partial \theta^{\eta}} f(x|\theta_i) = 0.
> \end{equation}
> After that, we can show this condition *satisfies for some complicated (black box) models* $h_0$ and family $f$ (Theorem 2.4, Corollary 2.5 and 2.6), which is novel and non-trivial. We believe that it is the first-time such result is presented in the literature.
>
> **[1b] (Novelty of Inverse Bounds)** Since we use a new notion of distinguishability for the deviated mixture model, the inverse bounds (lower bound TV distances by certain divergences based on Wasserstein distance in latent mixing measures of parameters) are different from those in the literature of traditional mixture model. In particular, for distinguishable cases, by carefully characterizing the inverse bounds, we find that the convergence rate for deviating proportion $\lambda^*$ is of parametric rate $1/\sqrt{n}$ for both exact-fitted and over-fitted and weakly identifiable cases. *This can not be derived if we use the available inverse bounds in the literature*. Indeed, those in the literature can only give us convergence rate for $G$ to $G_*$ in Wasserstein distance (up to some power) but do not precisely give convergence rates for each proportion $p_i$ or parameter $\theta_i$ (or show that they may have different convergence rates).
>
> In the partially distinguishable case, the theory is more challenging in the sense that we can lose the identifiability of our model (see discussion in Section 3.4.1). Therefore we do not hope to have a convergence of estimated $\theta$ to the true $\theta^*$. To the best of our knowledge, there is no literature on mixture models where the identifiability condition is not satisfied. By Theorems 3.6 and A.1-A.5, we provide a rather comprehensive picture of the convergence behavior of $\lambda$ and $G$ in this case. At a high level, it depends on the regime of the solution of the identifiable equation (Eq.(8)). We believe that these results are novel and of interest to the mixture model learning community.

---

> > ### Author Response · Authors · 2022-08-01
> > **Presentation of results**
> >
> > **2. [Presentation of the proofs]** Per your suggestion, we already added a proof sketch for the first inverse bound result (Theorem 3.3) so that the reader can have high-level understanding of the proof technique. All other inverse bounds can be proved with quite similar spirit but different in some details. We also commented on the difference after each theorem.
> >
> > **3. [Proof of Theorem 3.1]** We have added this proof in detail in the Appendix C.4 of the revised version. It is a modifier of the proof of Theorem 7.4 in [Ref. 26] using an explicit upper bound for entropy integral (Assumption A3).
> >
> > **4. [Typos]** Thanks for pointing out the typos. We already corrected them in the revised version of the manuscript.
> >
> > All the changes are marked in blue color. Please let us know if there is something unclear in our answers, or if you have any further questions.

---

> > > ### Author Response · Authors · 2022-08-07
> > > **Details on the novelty of inverse bounds compared to previous work**
> > >
> > > Hi reviewer,
> > >
> > > We want to clarify the novelty of the inverse bounds in our paper in more details. At a high level, the difficulties in our setting arise from the distinguishability of $h_0$ from the mixture of $f$. In particular, if $h_0$ is very different from a mixture of $f$ (Distinguishable setting), then we have the parameter estimation in the model is well behaved in the sense that we can estimate the deviating proportion $\lambda$ at a parametric rate. If $h_0$ is approximated by a mixture of $f(x|\theta)$ or is a mixture of $f(x|\theta)$ itself, then we may lose the identifiability in our model and can not estimate $\lambda$ well. In some cases, it does not even converge. We discuss it in details below.
> > >
> > > **[Convergence rate of parameter estimation in vanilla models]** Consider a distribution family $f(x|\theta)$ and denote a mixture of them as $f_G(x) = \sum_{i=1}^{k} p_i f(x|\theta_i)$ for $G = \sum_{i=1}^{k} p_i \delta_{\theta_i}$. This family is strongly identifiable in the second order. So that for fixed true $G_*$ having $k_*$ atoms, the inverse bound for them have the form $V(p_G, p_{G_*})\gtrsim W_1(G, G_*)$ for all $G$ having exactly $k_*$ atoms *(Exact-fitted case)*. The inverse bound is $V(p_G, p_{G_*})\gtrsim W_2^2(G, G_*)$ for all $G$ having no more than $K$ atoms, for $K$ is larger than $k_*$ *(Over-fitted case)*. It means that we can estimate $G_*$ at $1/\sqrt{n}$ rate in the exact-fitted case and at $1/n^{1/4}$ rate in the over-fitted case.
> > >
> > > **[Distinguishable settings]** By Theorem 3.3 and Theorem 3.4, we proved that if $h_0$ is distinguishable from mixtures of $f$, then we have inverse bound $$V(p_{\lambda G}, p_{\lambda^* G_*}) \gtrsim |\lambda - \lambda^*| + W_1(G, G_*)$$ in the exact-fitted case, and
> > > $$V(p_{\lambda G}, p_{\lambda^* G_*}) \gtrsim |\lambda - \lambda^*| + W_2^2(G, G_*),$$
> > > in the over-fitted case. It means that we can estimate $\lambda^*$ at a parametric rate for both cases, and the estimation rate of $G_*$ is not affected by the signal from $h_0$ (so it stays $1/\sqrt{n}$ in the exact-fitted setting and $1/n^{1/4}$ in the over-fitted setting). We want to clarify that this result is of interest on its own, and it is useful when we combine it with the theory of distinguishability that we developed in Section 2, where we allow $h_0$ to be estimated by a KDE or Neural network. Therefore it is different from the vanilla mixture in its spirit and applications.
> > >
> > > **[Partially distinguishable settings]** This setting may seem the most like theory of vanilla mixtures. However, many theoretical difficulties may arise when there are interactions between $h_0$ and mixtures of $f$. Recall that here we assume $h_0 = p_{G_0} = \sum_{i=1}^{k_0} p_i \delta_{\theta^0_i}$, and $G^* = \sum_{i=1}^{k_*} p_i^* \delta_{\theta_i^*}$. Hence, the true generating model is
> > > $$p_{\lambda^* G_*}(x) = (1-\lambda^*)\sum_{i=1}^{k_0} p_i^0 f(x|\theta_i^0) + \lambda^* \sum_{i=1}^{k_0} p_i^* f(x|\theta_i^*), \quad (1)$$
> > >
> > > and the working model that we use to estimate the data is
> > >
> > > $$p_{\lambda G}(x) = (1-\lambda)\sum_{i=1}^{k_0} p_i^0 f(x|\theta_i^0) + \lambda \sum_{i=1}^{K} p_i f(x|\theta_i). \quad (2)$$
> > >
> > > So the interesting issue here is the interaction between $G_0$ and $G_*$: What if they have common atoms? If yes, then how does it affect the convergence of $\lambda$ and $G$?
> > >
> > > As a concrete example, we take $k^* = k_0 = 2$, and assume that $G_*$ and $G_0$ share one atom $\theta_1^* = \theta_1^0$, then the model (1) becomes
> > >
> > > $$p_{\lambda^* G_*}(x) =  ((1-\lambda^*)p_1^0 + \lambda^* p_1^*) f(x|\theta_1^0) + (1-\lambda^*)p_2^0 f(x|\theta_2^0) + \lambda^* p_2^* f(x|\theta_2^*). \quad (3)$$
> > >
> > > We wonder which part of the working model $(2)$ estimate the first component $((1-\lambda^*)p_1^0 + \lambda^* p_1^*) f(x|\theta_1^0)$? On the one hand, the coefficient $\lambda$ in model $(2)$ can be estimated such that $(1-\lambda) p_1^0 = (1-\lambda^*)p_1^0 + \lambda^* p_1^*$. On the other hand, one component (say, $\hat{\theta}_1$) of $G$ in model $(2)$ and also converge to $\theta_1^0$ and
> > > $(1-\lambda) p_1^0 + \lambda \hat{\theta}_1 \to (1-\lambda^*) p_1^* + \lambda^* p_1^0$.
> > >
> > > It takes several results to completely characterize the convergence behavior in this setting. One can see Theorem 3.6 (or A.1-A.5), which is our treatment of the partially distinguishable setting. At a high level, we show that the convergence of $\lambda$ and $G$ can be characterized from the identifiable equation $p_{\lambda G} = p_{\lambda^* G_*}$. In particular, convergence can have different behavior in different solution regimes of this equation. For illustrating purpose, we can also see the experiment in Appendix B, where $\lambda$ may converge to somewhere different than $\lambda^*$.
> > >
> > > We hope this discussion can help the readers and reviewers see this setting better. Thank you.

---

> ### Author Response · Authors · 2022-08-09
> **Looking forward to your feedback.**
>
> Dear Reviewer iPiq,
>
> We have addressed your concerns in our responses about the difference between the strong identifiability and distinguishability conditions as well as providing an explanation for the novelty of the bounds derived in the paper.
>
> Given that the deadline for the rebuttal discussion is only a few hours from now and you are the only one giving a negative score on the paper, we would like to hear your feedback. Please feel free to raise questions if you have other concerns.
>
> Best regards,
>
> Authors

---

> > ### Comment · Reviewer_iPiq · 2022-08-10
> > **Thank you for your response**
> >
> > Thank you for your detailed response. The new modifications greatly help me (and hopefully the reader) to understand the main ideas of the proof. I also want to say that I appreciate the examples showing that the types of new results that the paper produces, in comparison to the existing literature.  The new results do stem for the novel notion of distinguishability that the authors introduce. In some ways, it seems that this notion is the ``natural extension'' of the previous notion of identifiability, however this might just be hindsight.  I am happy to give the authors the benefit of the doubt and raise my score to a 6.

---

### Author Response · Authors · 2022-08-02
**Summary of the revision**

Dear Reviewers and Chairs,

We would like to thank the reviewers for the valuable feedback. We have answered all the questions in the discussions section. Besides, based on the reviewers' recommendations, we have also included several results, explanations, and discussions in our revision (in blue color):

1. We highlight the novelties of our results and techniques in the Introduction section. Many reviewers comment that we did not explicitly state the paper's novelty. So we added some concise sentences to emphasize our new results. (For more detailed explanations, please refer to the supplementary material, or in the comment section of Reviewer 1.)

2. After each main theoretical result, we include some comments on the proof techniques, so it is easier for readers to approach just by reading the main text. We also added a proof sketch for the first inverse bound in the paper. For other inverse bounds, we commented on the difference in the proofs.

3. We added a discussion on the effect of large neural networks on parameter estimation in the last section and a discussion about known $h_0$ in the first section.

4. The proof of Theorem 3.1 is provided, which is a modifier of a standard result in M-estimation theory.

5. We have fixed typos and revised the writing based on the suggestions of reviewers (in blue color).

We are looking forward to hearing your feedback,

Best regards,

Authors.

---

### Meta-Review · Area_Chair_tbxj · 2022-08-24

**Recommendation:** Accept
**Confidence:** Less certain

**Metareview:**

The authors study a mixture of a single known distribution h0 and an mixture model whose parameters are unknown. They propose new notions of distinguishability and partial distinguishability that they use for characterizing convergence rates.

The reviewers had a mixed opinion about the paper and had several comments about the technical novelty of the work. The authors addressed these suitably in their rebuttal. In particular, their revised introduction is much clearer about the motivation and contributions. I am happy to recommend acceptance of the paper.

**Award:**

No

---

### Decision · Program_Chairs · 2022-09-14

Accept